# Obesity-induced overexpression of *miR-802* impairs insulin transcription and secretion

Fangfang Zhang[1], Dongshen Ma[1,2], Wanli Zhao[3], Danwei Wang[1], Tingsheng Liu[1], Yuhong Liu[1], Yue Yang[1], Yue Liu[1], Jinming Mu[1], Bingbing Li[1], Yanfeng Zhang[1], Yi Pan[1], Changying Guo[1], Hong Du[4], Ling Li[5], Xianghui Fu[6✉], Zhengyu Cao[3✉] & Liang Jin[1✉]

B cell dysfunction due to obesity can be associated with alterations in the levels of micro-RNAs (miRNAs). However, the role of miRNAs in these processes remains elusive. Here, we show that *miR-802* is increased in the pancreatic islets of obese mouse models and demonstrate that inducible transgenic overexpression of *miR-802* in mice causes impaired insulin transcription and secretion. We identify Foxo1 as a transcription factor of *miR-802* promoting its transcription, and NeuroD1 and Fzd5 as targets of *miR-802*-dependent silencing. Repression of NeuroD1 in β cell and primary islets impairs insulin transcription and reduction of Fzd5 in β cell, which, in turn, impairs $Ca^{2+}$ signaling, thereby repressing calcium influx and decreasing insulin secretion. We functionally create a novel network between obesity and β cell dysfunction via *miR-802* regulation. Elucidation of the impact of obesity on microRNA expression can broaden our understanding of pathophysiological development of diabetes.

[1] State Key Laboratory of Natural Medicines, Jiangsu Key Laboratory of Druggability of Biopharmaceuticals, School of life Science and Technology, China Pharmaceutical University. 24 Tongjiaxiang, Jiangsu province, Nanjing, PR China. [2] Department of Pathology, Affiliated Hospital of Xuzhou Medical University, Xuzhou, Jiangsu, China. [3] Jiangsu Key Laboratory of TCM Evaluation and Translational Research, School of Traditional Chinese Pharmacy, China Pharmaceutical University, Nanjing 211198, PR China. [4] Department of Endocrinology, Nanjing Jinling Hospital. 305 Zhongshan East Road, Nanjing, jiangsu, PR China. [5] Department of Endocrinology, School of Medicine, Zhongda Hospital, Southeast University, 87 DingJiaQiao Rd, Nanjing, Nanjing, Jiangsu, PR China. [6] Division of Endocrinology and Metabolism, State Key Laboratory of Biotherapy, West China Hospital, Sichuan University and Collaborative Innovation Center of Biotherapy, Chengdu 610041 Sichuan, China. ✉email: xfu@scu.edu.cn; zycao1999@hotmail.com; ljstemcell@cpu.edu.cn

Obesity is a predisposing factor for the development of type 2 diabetes (T2D). Obesity promotes insulin resistance in target tissues and has detrimental effects on β cells, resulting in reduction of insulin content, abnormally elevated insulin release in the absence of stimuli, diminished capacity to secrete insulin in response to glucose, and increased β cell apoptosis[1,2]. The mechanisms underlying the negative impact of obesity on β cell functions are still incompletely understood. Whole-genome association studies for T2D susceptibility genes revealed that most of the associated variants were located in non-coding regions[3,4]. We hypothesize that failure of β cell functioning due to obesity can be associated with alterations in the levels of micro-RNAs (miRNAs).

MiRNAs are endogenous small (~22 nt), non-coding RNAs that target the 3′ untranslated region (3′UTR) of messenger RNAs by repression of protein translation or by cleavage of mRNAs, resulting in diminished target protein synthesis[5–8]. It has been previously speculated that up to one third of all human genes might be controlled by miRNAs[9]. Moreover, miRNAs have been shown to be involved in multiple biological processes, including glucose homeostasis and lipid metabolism[10–12]. Recent studies have highlighted some nutrient-sensitive miRNAs. For example, palmitate causes a time- and dose-dependent increase in expression of miR-34a and miR-146 in Min6 cells[13]. Obesity-induced overexpression of miR-143 inhibits insulin-stimulated AKT activation and impairs glucose metabolism[14]. miR-33a/b is involved in the regulation of fatty acid metabolism and insulin signaling[15]. However, the role of miRNAs in regulation of β cell functions during obesity is still largely unknown.

In this study, we investigate the potential involvement of miRNAs in obesity-mediated β cell dysfunction. We find that expression of miR-802 is upregulated in the islets of genetic and dietary mouse models of obesity. Detailed analysis of the role of the obesity-sensitive miRNAs reveals that modification of miR-802 levels has an important impact on different β cell functions. Our data suggests that the harmful effects of obesity on insulin secreting cells may be mediated, at least partially, by alterations in the miRNA expression pattern.

## Results

**miR-802 is upregulated in the islets of obese mouse models**. To identify miRNAs that are dysregulated during obesity and that may contribute to β cells dysfunction, we performed "miRNome" expression profiling using RNA-seq analysis on RNA isolated from islets of two mouse models of obesity: high fat diet (HFD)-fed mice compared to normal chow diet (NCD) fed mice and mice homozygous for the diabetes db mutation of the leptin receptor (Lepr[db/db]) compared to wild type controls. The body weight, blood glucose, and insulin levels of these mice were listed in Supplementary Fig. 1a–f. Out of 2612 miRNA-specific probe sets, 1282 (49.1%) and 1330 (50.9%) miRNAs were detected in islets of HFD and Lepr[db/db], respectively (Supplementary Fig. 1g, h). In the islets of HFD-fed mice, expression of 41 miRNAs was significantly altered compared to miRNAs in NCD mice, of which expressions of 20 (49%) miRNAs increased (Fig. 1a, Supplementary Table 4). In Lepr[db/db] islets, expressions of 120 miRNAs were significantly changed, of which expressions of 72 (60%) miRNAs increased (Fig. 1b, Supplementary Table 5). Furthermore, we performed cluster analysis of the top 10 upregulated miRNAs in the islets of HFD and db/db mice, respectively (Supplementary Fig. 1i, j). Intriguingly, miR-802-5p (miR-802) and miR-1945 were consistently upregulated in both obese models. miR-1945 has been identified in the mouse genome, but its human homologue has not yet been reported. Moreover, it has recently shown that hepatic miR-802 can be induced by obesity

and plays a role in insulin resistance and glucose metabolism[16]. However, the role of miR-802 in pancreatic β cells remains unknown. Therefore, we chose miR-802 for further analysis.

Next, increased miR-802 expression in the islets of obese mouse models was further confirmed by qRT–PCR analyses, which revealed a 2-fold and 6-fold upregulation of miR-802 expression in the islets of HFD-fed mice and Lepr[db/db] mice (Fig. 1c, d), respectively. As shown in Fig. 1e, the expression level of miR-802 in the islet was up-regulated along with body weight gain. We next determined its expression in various organs of obese mice compared with control mice. This analysis revealed significant upregulation of miR-802 in liver, kidney, white adipose tissue (WAT), brown adipose tissue (BAT), and skeletal muscle of obese mice (Fig. 1f). As expected, miR-802 expression was highly enriched in the liver and islet, and also abundant in kidney, heart and WAT, while miR-802 almost could not detect in other tissues of wide type mice (Supplementary Fig. 1k). Moreover, miR-802 levels were significantly increased in overweight (body mass index (BMI) > 25) individuals compared with lean individuals (Fig. 1g) and serum miR-802 expression levels were significantly correlated with the BMI of these subjects (Fig. 1h). As shown in Fig. 1i, the expression level of Pri-miR-802 in the islets of HFD mice displayed a similar trend with miR-802. Taken together, miR-802 expression in islets is increased in both dietary and genetic mouse models of obesity.

***Foxo1* improves miR-802 level in the islets of obese mice**. To determine the possible causes of the changes in miR-802 expression detected in the islets of obese mice, we exposed normal mouse islets and Min6 cells to pathophysiological concentrations of palmitate and pro-inflammatory cytokines. The expression of miR-802 only increased in the presence of palmitate (Fig. 2a, b and Supplementary Fig. 2a, b). Next, we explored the molecular mechanisms underlying the upregulation of expression of miR-802 in islets of obese mice. We predicted miR-802 promoter region (Supplementary Fig. 2c) 3 kb upstream of mice miR-802 sequence, and constructed the four sgRNAs corresponding to its promoter in lentiCRISPRv2 puro vector. Supplementary Fig. 2d shows that miR-802 levels were decreased in Min6 cells transfected with sgRNAs. These results suggested that the predicted promoter region could modify miR-802 expression level. In addition, we identified 20 putative miR-802 transcription factors (Supplementary Fig. 2e) through Jaspar and Promo, among which, the nuclear-protein and mRNA levels of Foxo1 were upregulated in response to palmitate treatment in primary islets as well as in obese mice islets (Fig. 2c, d), which possess potential binding site within the promoter of miR-802. Min6 cells transfected with ADA-Foxo1 exhibited significantly the higher binding ability of Foxo1 to miR-802 promoter compared to control via dual luciferase (Supplementary Fig. 2f) and ChIP (Fig. 2e, Supplementary Fig. 2g) assay. And we also detected this binding was further increased in 0.5 mM palmitate treated Min6 cells (Fig. 2f and Supplementary Fig. 2h), as well as in obese mice islets (Fig. 2g and Supplementary Fig. 2i). As shown in Fig. 2h signal from the probe-protein-anti-Foxo1 complex was detected using a miR-802 probe in Min6 cells via EMSA assays. The same result was obtained in the islets (Fig. 2i). Further research showed that palmitate-induced miR-802 upregulation in Min6 cells was partly reversed by knockdown of Foxo1 (Fig. 2j).

Next, to verify whether knockdown of Foxo1 also repressed miR-802 expression level in vivo, $1 \times 10^9$ lentivirus particles encoding Foxo1-shRNA was injected through the tail vein. We observed an 80% reduction of Foxo1 expression in islets that has received the lentivirus-shFoxo1 compared to those receiving lentivirus-LV3 (pGLV-H1-GFP + Puro, Supplementary Fig. 2j

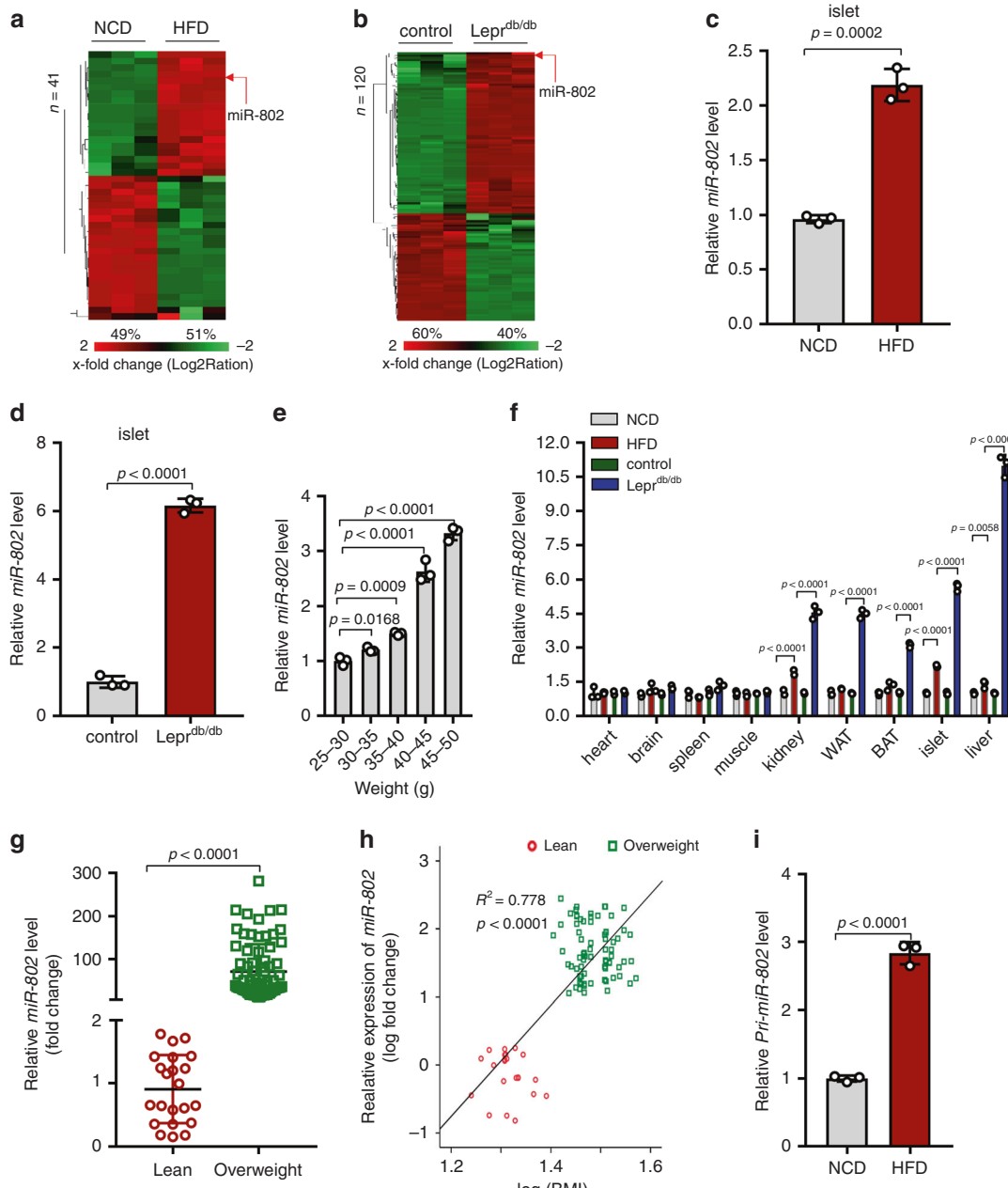

**Fig. 1 *miR-802* expression level in obese mice and obese individuals.** Heat map diagram illustrating the differential expression of miRNAs in islets of HFD compared to normal chow-diet (NCD) mice (**a**), $n = 8$ NCD group and $n = 7$ HFD group), and in Lepr[db/db] compared to wide type control mice (**b**) ($n = 8$ control group and $n = 7$ Lepr[db/db] group). Red and green indicate increased and decreased gene expression levels, respectively. *miR-802* was significantly upregulated in islets of HFD mice (**c**), and Lepr[db/db] mice (**d**) ($n = 7$) compared to NCD and wild type control as measured by qRT-PCT. **e** qRT-PCR was performed to measure the expression level of *miR-802* in the islets at different stages (after 0-week, 4-week, 6-week, 8-week and 16-week feeding HFD) during the development of obesity inducing diabetes ($n = 5$). **f** qRT-PCR was performed to evaluate the expression levels of *miR-802* in different tissues of obese and wild type mice ($n = 7$). **g** The expression levels of *miR-802* in the serum extracted from lean individuals ($n = 22$) and overweight individuals ($n = 72$), using *Ce-miR-39-1* as positive control. *miR-802* expression was set to 1 in SD. Data sets were statistically analyzed using two-tailed unpaired Student's t test and Bonferroni Post-hoc correction. **h** Correlation between *miR-802* levels and BMI. Pearson's correlation coefficients ($r$) and p values are shown. **i** The expression level of *Pri-miR-802* in the islets of HFD and NCD mice were analyzed by qRT-PCR ($n = 3$). All experiments above were performed in triplicates, and each group contained three batches of individual samples. The p-values by Pearson's correlation test h, two-tailed unpaired Student's t test c, d, g, and i, one-way ANOVA e, or two-way ANOVA f are indicated. Data represent the mean ± SD. Source data are provided as a Source Data file.

and Supplementary Fig. 2k). Although *lentivirus-shFoxo1* treatment also efficiently reduced Foxo1 expression in the liver, it only slightly reduced Foxo1 expression in WAT and kidney (Supplementary Fig. 2j). As showed in Fig. 2k, after *lentivirus-shFoxo1* treatment in vivo, the expression level of *miR-802* was dramatically decreased in the islet. Then, *Lentivirus-shFoxo1*

treatment mice were fed with HFD for 8-week weighting 40-45 g. The expression level of *miR-802* in the islets has no significant up-regulation compared with control (Fig. 2l). Taken together, we speculated that increase in the expression of *miR-802* in the islets of obese mouse models was mediated by up-regulation of *Foxo1*.

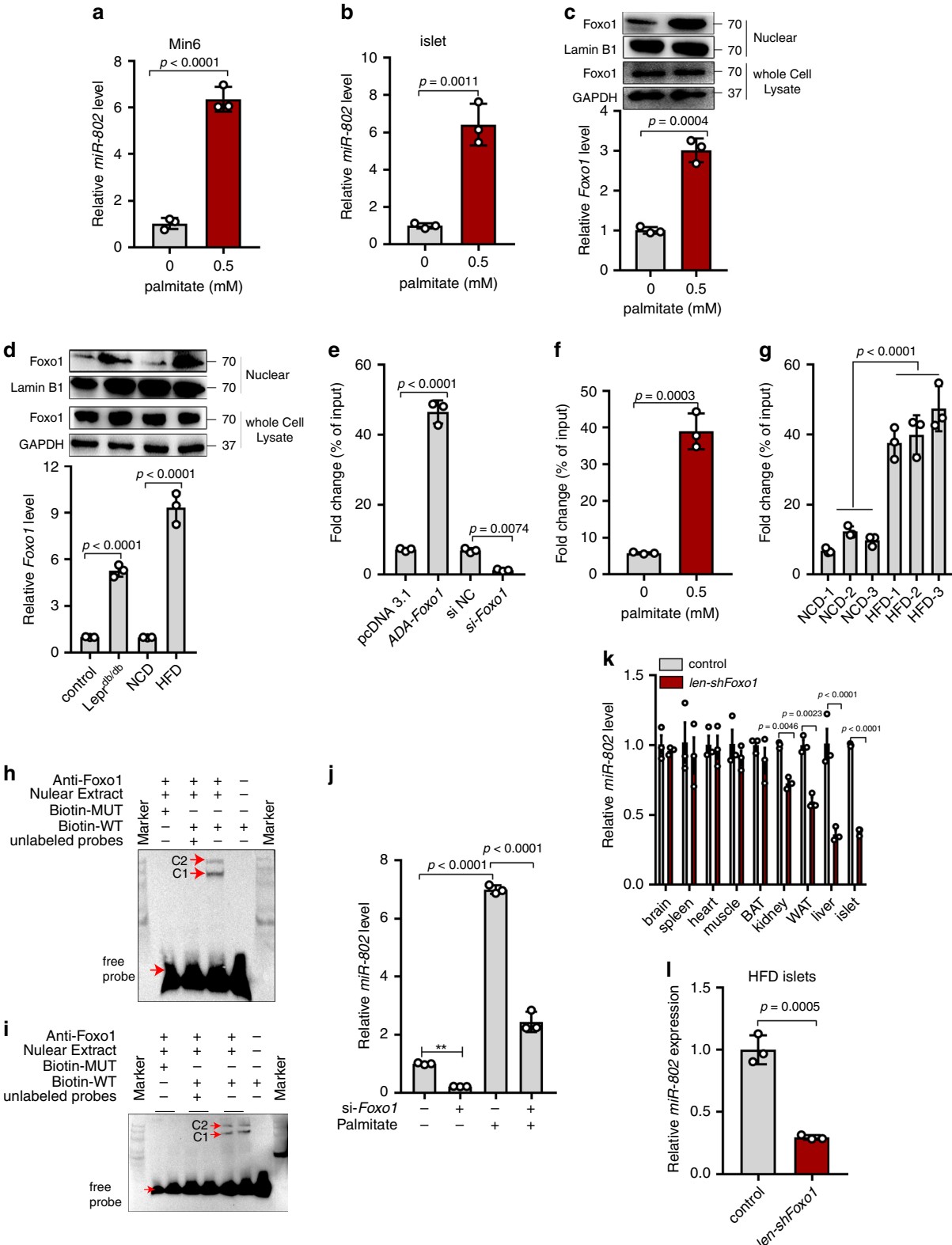

**Overexpression of miR-802 in β cell impairs β-cell functions**. To investigate whether increased *miR-802* expression in islets is involved in regulation of insulin synthesis and secretion, we overexpressed *miR-802* in primary islets and Min6 cells (Supplementary Fig. 3a, b). Overexpression of *miR-802* significantly decreased the expression of insulin at mRNA and protein levels (Fig. 3a, b), as well as insulin content (Fig. 3c, d). Next, we performed glucose

challenge experiment on primary islets and Min6 cells. Overexpression of *miR-802* decreased insulin secretion at high glucose, even in the presence of 35 mM KCl, a general depolarizing agent (Fig. 3e, f). However, we found that *miR-802* exhibited no effects on proliferation and apoptosis of Min6 cells (Supplementary Fig. 3c, d).

To verify whether overexpression of *miR-802* also impairs insulin transcription and secretion in vivo, we mimicked the

**Fig. 2 Foxo1 induces miR-802 expression.** Min6 cells and primary islets were incubated with 0.5 mM primary for 48 h and qRT-PCR was performed to examine the miR-802 levels, in Min6 cells (**a**), and in primary islets (**b**). **c** Primary islets were incubated 0.5 mM palmitate for 48 h, the nuclear and whole cell lysate protein and mRNA levels of Foxo1 were determined by western blot and qRT-PCR, respectively. **d** The nuclear and whole cell lysate protein and mRNA levels of Foxo1 were assessed in the islets of obese model mice using western blot and qRT-PCR, respectively. **e–g** The enrichment of Foxo1 on the miR-802 promoter relative to IgG detected by ChIP-qPCR assays, in Min6 cells transfected with ADA-Foxo1, pcDNA 3.1 vector, si-Foxo1 or si NC (**e**), in Min6 cells treatment with 0.5 mM palmitate or without palmitate (**f**), and in obese mice islets or normal mice islets (**g**), $n = 5$. Foxo1 could directly bind to miR-802 promoter in Min6 cells (**h**) and islets (**i**), $n = 5$ through EMSA assays. C1 and C2 represented nuclear-protein-miR-802 probe complexes, nuclear-protein-miR-802 probe-anti-Foxo1 complexes, respectively. Biotin-WT was a 25 bp fragment probe which included the binding region of Foxo1, while Biotin-MUT was a 25 bp fragment probe and the binding sequence was mutated. **j** Min6 cells were incubated with 0.5 mM palmitate and co-transfected with si-Foxo1, followed by qRT-PCR to examine the expression levels of miR-802. **k** qRT-PCR was performed to measure the miR-802 expression levels in lentivirus-shFoxo1-treated mice compared with control (white adipose tissue (WAT), brown adipose tissue (BAT), $n = 5$). **l** miR-802 expression after intravenous injection of HFD-fed mice with lentivirus-shFoxo1 ($n = 5$). All experiments above were performed in triplicates, and each group contained three batches of individual samples. The p-values by two-tailed unpaired Student's t test (**a–c**, **f** and **l**), one-way ANOVA d, e, g and j or two-way ANOVA k are indicated. Data represent the mean ± SD. Source data are provided as a Source Data file.

obesity-associated increase in miR-802 expression in a transgenic mouse model by generating H11-CAG-LSL-miR-802 knock-in mice using CRISPR/Cas9 system (Supplementary Fig. 3e). Mutant mice were identified by Southern blotting (Supplementary Fig. 3f, $n = 6$). Homozygous miR-802 knock-in mice (miR-802$^{ki/ki}$) were crossed with RIP-Cre transgenic animals to selectively over-express miR-802 in β cells (Supplementary Fig. 3e). Rip-Cre miR-802$^{ki/ki}$ (miR-802 KI) mice were born after mating between Rip-Cre miR-802$^{ki/wt}$ mice and miR-802$^{ki/ki}$ mice (Supplementary Fig. 3g). Assessment of recombination efficiency by the Cre transgene revealed selective overexpression of miR-802 genes in pancreatic islets (Supplementary Fig. 3h, $n = 5$). The sgRNA we used almost has no off-target effect identified by PCR and sequencing of mouse tail genomic DNA (Supplementary Fig. 3i, j, $n = 10$). qRT-PCR revealed an approximately 500-fold increase in total miR-802 levels in miR-802 KI mice relative to miR-802$^{ki/ki}$ mice (Supplementary Fig. 3k, $n = 5$). Moreover, a 12-fold increase of miR-802 expression in the serum of miR-802 KI mice (Supplementary Fig. 3l, $n = 10$). Since Rip-Cre has been previously implicated in hypothalamus and hippocampus[17]. As shown in supplementary Fig. 3 m, Rip-Cre expression was 1400-fold and 400-fold higher in islet versus hypothalamus and hippocampus ($n = 5$). Indeed, a 2-fold and 7-fold upregulation of miR-802 expression in the hypothalamus and hippocampus of miR-802 KI mice (Supplementary Fig. 3n, $n = 5$). Metabolic analysis of mice revealed similar weight and blood glucose level in both male and female miR-802 KI mice and littermate controls (miR-802$^{ki/ki}$ and Rip-Cre miR-802$^{ki/wt}$, Supplementary Fig. 3o, p, $n = 11$ control group, $n = 9$ miR-802 KI group. The actual number of mice for each genotype was listed in Supplementary Table 11). In contrast, miR-802 KI mouse exhibited impaired glucose tolerance when challenged with IPGTT (Fig. 3g, $n = 11$ control group, $n = 9$ miR-802 KI). Importantly, lower levels of insulin were detected in miR-802 KI mice at 5, 15, and 30 min after glucose injection compared with control mice (Fig. 3h, $n = 4$). In addition, insulin content and sensitivity to glucose or 35 mM KCl were decreased in the islets derived from miR-802 KI mice (Fig. 3i, j, $n = 5$). Inspection of islet architecture of miR-802 KI mice revealed intact endocrine cell organization (Fig. 3k, $n = 3$) and morphometric analyses revealed that β cell mass was slightly decreased in miR-802 KI animals compared with miR-802$^{ki/ki}$ mice (Supplementary Fig. 3q, $n = 3$). Glucagon content was not changed in miR-802 KI animals (Supplementary Fig. 3r, $n = 6$). Taken together, these results indicated that overexpression of miR-802 in β cells impairs glucose tolerance by decreasing insulin content and secretion.

To evaluate whether overexpression of miR-802 could reverse the protective effect of β-cell Foxo1 deletion in vivo. We injected lentivirus-miR-802 in lentivirus-shFoxo1 treated mice by an HFD treatment for 8 weeks. The expression level of miR-802 in the

islets was increased approximately 250-fold compared to control mice (Supplementary Fig. 3s, $n = 5$), and lentivirus-miR-802 could also partially restore the pri-miR-802 expression level suppressed by lentivirus-shFoxo1 (Supplementary Fig. 3t, $n = 5$). We observed that lower fasting serum insulin (FINS) levels and homeostasis model assessment of the insulin resistance index (HOMA-IR) were reversed by lentivirus-miR-802 compared to merely lentivirus-shFoxo1 treated mice (Fig. 3l and m, $n = 7$). In addition, compared with the lentivirus-shFoxo1 group, the area under the curve obtained from IPGTT and IPITT assays was increased in lentivirus-miR-802-treated mice, and these effects were markedly ameliorated by lentivirus-shFoxo1 treatment versus control group on HFD treatment (Fig. 3n and o, $n = 7$). Moreover, the ability of insulin transcription (Fig. 3p, $n = 5$) and secretion (Fig. 3q, $n = 5$) were suppressed in mice receiving lentivirus-shFoxo1 and lentivirus-miR-802 treatment compared to merely lentivirus-shFoxo1-treated mice. In summary, these results revealed that overexpression of miR-802 could reverse the protective effect of β-cell Foxo1 deletion in vivo.

**MiR-802-deleted increases insulin content and secretion.** To study the consequence of reduced miR-802 level in pancreatic β cells, we knocked down miR-802 in primary islets and Min6 cells (Supplementary Fig. 4a, b). Knockdown of miR-802 significantly increased the expression of insulin at mRNA, protein levels (Fig. 4a, b), and insulin content (Fig. 4c, d). Knockdown of miR-802 increased insulin secretion both in high glucose and 35 mM KCl even at low glucose exposure (Fig. 4e, f). However, miR-802 knockdown did not affect proliferation and apoptosis of Min6 cells (Supplementary Fig. 4c, d).

To study the effect of reduced miR-802 level in pancreatic β cells in in vivo, we generated miR-802 conditional knockout mice using the Cre/Lox system (Supplementary Fig. 4e). Mutant mice were identified using PCR (Supplementary Fig. 4f, $n = 4$). miR-802$^{fl/fl}$ were crossed with Rip-Cre transgenic animals to select Rip-Cre miR-802$^{fl/wt}$ mice, then Rip-Cre miR-802$^{fl/wt}$ mice were crossed with miR-802$^{fl/fl}$ mice to selectively ablate miR-802 expression in β cells (Supplementary Fig. 4g). Assessment of recombination efficiency by the Cre transgene revealed selective deletion of miR-802 genes in pancreatic islets (Supplementary Fig. 4h, $n = 5$). Rip-Cre miR-802$^{fl/fl}$ (miR-802 KO) mice were born at Mendelian frequencies (Supplementary Fig. 4g, $n = 7$, the actual number of mice for each genotype was listed in Supplementary Table 12) and were normal. These sgRNAs we used have no off-target effect via PCR and sequencing of tail DNA (Supplementary Fig. 4i and j, $n = 10$). Expression analysis revealed an approximately 70% decrease in total miR-802 levels in miR-802 KO versus miR-802$^{fl/fl}$ islets (Supplementary Fig. 4k, $n = 5-6$). Moreover, it only slightly reduced miR-802 expression in

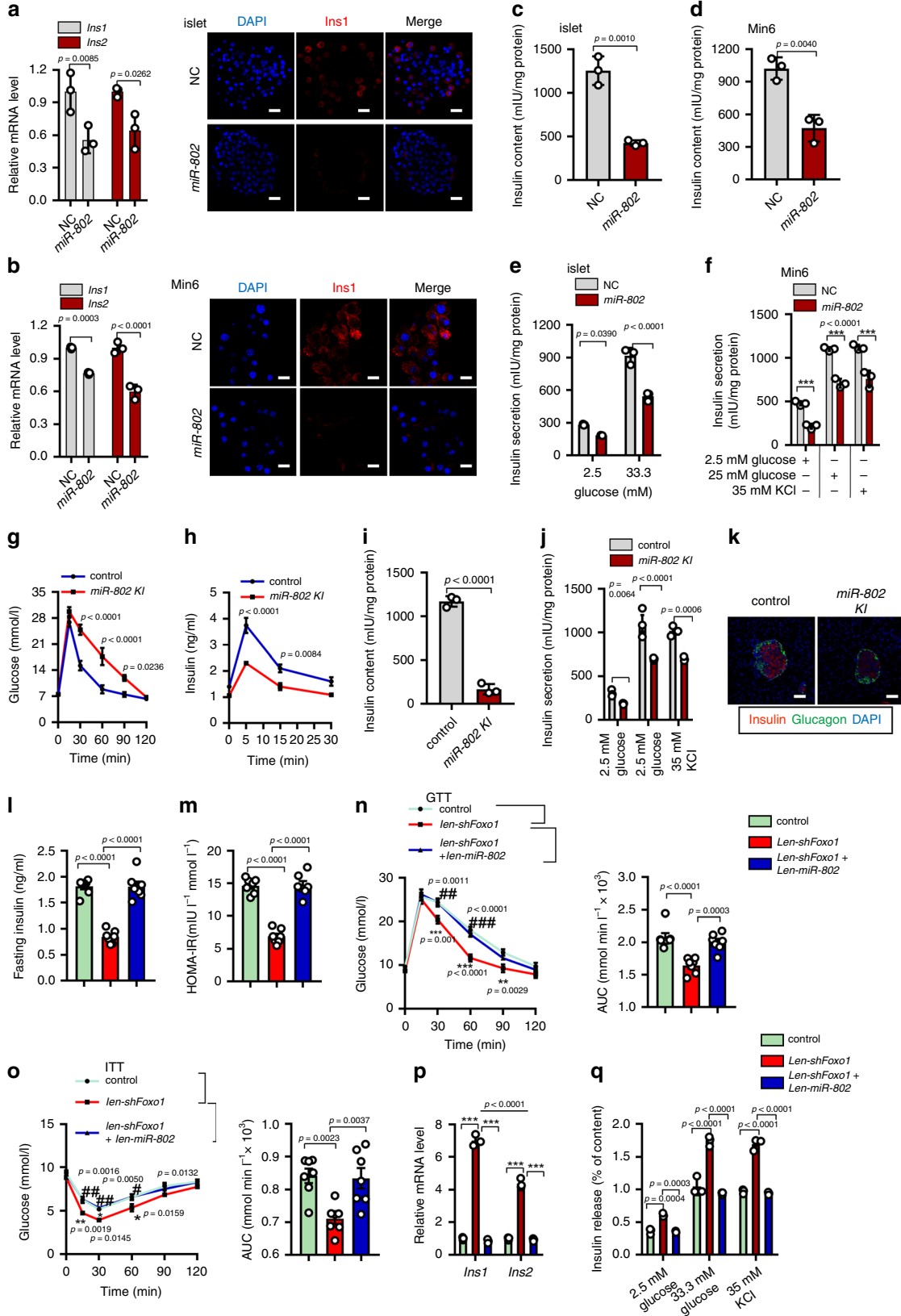

hippocampus (Supplementary Fig. 4l). Although their weight and blood glucose levels were similar to that of littermate controls (*miR-802^{fl/fl}*) in both male and female (Supplementary Fig. 4m, n, *n* = 9), *miR-802 KO* mouse showed improved glucose tolerance when challenged with IPGTT (Fig. 4g, *n* = 4). Importantly, *miR-*

*802 KO* mice displayed higher insulin levels compared with control mice at 5, 15, and 30 minutes after glucose injection (Fig. 4h, *n* = 7). Glucose-induced insulin secretion assays was performed in the islets of 5- and 35-week-old mice (*n* = 5) to further characterize the ability of insulin secretion of *miR-802 KO*

**Fig. 3 Overexpression of *miR-802* impairs β cell function.** qRT-PCR was used to evaluate the *Ins1* and *Ins2* levels, followed by immunostaining for DAPI (blue) and insulin (red) (Magnification: 40 ×, scale bar: 20 μm), (**a**) in islets ($n = 3$, 8 weeks old) and (**b**) in Min6 cells ($n = 3$ cell wells); then, insulin content was measured in islets (**c**), $n = 5$, 8 weeks old) and Min6 cells (**d**), $n = 3$ cell wells); and insulin secretion was analyzed in islets (**e**), $n = 5$, 8 weeks old) and Min6 cells (**f**), ***$p < 0.001$. **g** IPGTT (2 g kg$^{-1}$) in overnight fasted *miR-802 KI* and control mice at 10 weeks of age ($n = 11$ control group, $n = 9$ *miR-802 KI*). **h** In vivo insulin excursions at 15 weeks of age ($n = 4$). **i** Insulin content in the serum of *miR-802 KI* and control mice ($n = 3$, 10 weeks old). **j** Static insulin secretion was evaluated in islets from 15-week-old *miR-802 KI* and control mice ($n = 5$). **k** Islet organization in pancreas of *miR-802 KI* and control mice at 10 weeks of age ($n = 3$). Staining for insulin and glucagon is shown. Scale bar: 50 μm. Fasting insulin levels (FINS, (**l**) and HOMA-IR (**m**) at indicated time points of *lentivirus-shFoxo1*-treated, *lentivirus-shFoxo1* and *lentivirus-miR-802*-treated mice fed an HFD diet compared with control mice ($n = 7$). HOMA-IR was calculated as HOMA-IR = (FBG (mmol L$^{-1}$) x FINS (mIU L$^{-1}$))/22.5. IPGTT (**n**) and IPITT (**o**) assays were performed to evaluate the insulin sensitivity of mice. Area under the curve (AUC) was calculated ($n = 7$). *$p < 0.05$; **$p < 0.01$; ***$p < 0.001$ versus control group; #$p < 0.05$, ##$p < 0.01$, ###$p < 0.001$ versus *lentivirus-shFoxo1* group, insulin synthesis (**p**) and insulin secretion (q, $n = 5$) were analyzed. The *p*-values by two-tailed unpaired Student's *t* test (c, d and l), one-way ANOVA (**l**, **m**) or two-way ANOVA (**a**, **b**, **e**–**h**, **j**, **n**–**q**) are indicated. All data are represented as mean ± SD except (**g**, **h**) and (**l**–**o**) (mean ± SEM). Source data are provided as a Source Data file.

mice. In both age groups, insulin secretion was increased 1.5- to 4-fold when treated with 33.3 mM glucose, and insulin secretion was increased by 2-fold when treated with 35 mM KCl (Fig. 4i, j). Then, endocrine β cell mass and function were analyzed to investigate the cause of miR-802-deletion improved glucose tolerance. The results showed that endocrine cell organization of islet architecture of *miR-802 KO* mice were intact (Supplementary Fig. 4o, $n = 3$) and endocrine cell mass revealed no significant differences via morphometric analyses (Supplementary Fig. 4p, $n = 3$). The mRNA and protein levels of insulin was increased in *miR-802 KO* animals (Fig. 4k, l, $n = 5$). And insulin content also increased (Fig. 4m, $n = 5$), but there was no change in glucagon levels (Supplementary Fig. 4q, $n = 6$). Taken together, these results indicated that deletion of *miR-802* in β cells improves glucose tolerance by increasing insulin content and secretion.

***miR-802* impairs obesity-associated insulin resistance**. Since our data demonstrated that expression of *miR-802* was upregulated in the islets of obese mice, we hypothesized that the improved metabolic control of *miR-802 KO* mice would be enhanced under HFD feeding. Then, we exposed 8-week old male *miR-802 KO* mice to HFD for 12 weeks. We confirmed that mice fed HFD for 12 weeks displayed no effect on body weight, random-fed glycemia, cumulative energy intake or body fat content as well as no effect on adipocyte size between control mice and *miR-802 KO* mice (Supplementary Fig. 5a–e). On the other hand, the serum insulin concentrations was diminished in *miR-802 KO* mice of HFD feeding (Fig. 5a, $n = 10$). In accordance with this, glucose tolerance tests revealed an improvement of glucose tolerance upon *miR-802 KO* mice (Fig. 5b, $n = 10$). Moreover, insulin sensitivity was also improved compared to control mice (Fig. 5c, $n = 10$) and *miR-802 KO* mice lead to a reduction of HOMA-IR indices (Fig. 5d, $n = 10$). Subsequently, we isolated islets of control mice and *miR-802 KO* mice after 6- and 12-week HFD treatment. GSIS results revealed that insulin release was markedly improved in *miR-802 KO* mice when islets exposed to 33.3 mM glucose or 35 mM KCl (Fig. 5e, f, $n = 5$). This finding further indicated that deletion of *miR-802* expression in islets contributes to the compensatory β cell secretory function instigated by insulin resistance and obesity.

**Identification of miR-802 target genes in pancreatic β cells**. To identify the target mRNAs of *miR-802* which serve as its molecular effectors in β cell dysfunction, four computational algorithms, including TargetScan, miRanda, RNAhybrid, and PicTar, were used in combination to search for potential targets of *miR-802*. We identified 10 putative *miR-802* target genes (Supplementary Fig. 6a). Among the candidates, Neurogenic Differentiation-1 (*NeuroD1*) and Frizzed class receptor 5 (*Fzd5*)

were predicted to be *miR-802* target by all the four algorithms, since they harbored an *miR-802* binding site, which were also conserved in humans, mice, and rats (Supplementary Fig. 6b, c). *NeuroD1* is an islet function activator gene that has been reported to play an important role in pancreatic β cell function[18,19], and *Fzd5*, as a receptor for *Wnt5a*[20], is the key molecule involved in progression of T2D by increasing intracellular calcium level[21,22]. Thus, *NeuroD1* and *Fzd5* were selected as viable candidates.

In most cases, miRNAs generally exhibit the expression patterns that are opposite to that of their targets. Therefore, we investigated whether *miR-802* expression is inversely correlated with *NeuroD1* and *Fzd5* expression in the islets of obese mice, which exhibited increased *miR-802* expression compared to controls. Figure 6a, b shows that the mRNA and protein levels of NeuroD1 and Fzd5 were decreased in the islets of HFD and Lepr$^{db/db}$ mice ($n = 5$), the similar results were obtained in the islets incubated with 0.5 mM palmitate (Fig. 6c, d). Subsequently, a luciferase reporter assay was performed to confirm that *miR-802* directly targeted the predicted binding sites in the *NeuroD1* (Supplementary Fig. 6b) and *Fzd5* 3′-UTR (Supplementary Fig. 6c). The putative *miR-802* MREs of the mouse *NeuroD1* and *Fzd5* 3′-UTR were inserted downstream of the firefly luciferase gene in a Pmir-PGLO reporter plasmid. As predicted, the luciferase activity was significantly reduced in Min6 cells co-transfected with luciferase reporter plasmid and *miR-802* mimic. Overexpression of *miR-802* has no effect on the mutated luciferase reporter activity (Fig. 6e, f).

We next investigated the ability of *miR-802* in regulating NeuroD1 and Fzd5 mRNA and protein levels in Min6 cells. When transfected with *miR-802* mimic, the mRNA and protein levels of NeuroD1 and Fzd5 were significantly decreased in the Min6 cells (Supplementary Fig. 6d, e). Similar pattern was observed in the islets of *miR-802 KI* mice and *miR-802 KO* mice compared to control (Fig. 6g–j, $n = 6$). *Lentivirus-shFoxo1* remarkably increased the mRNA and protein levels of *miR-802* targets NeuroD1 and Fzd5, whereas the simultaneous overexpression of *miR-802* nearly abolished these increases (Supplementary Fig. 6f–h, $n = 5$). Taken together, these experiments indicated *NeuroD1* and *Fzd5* to be the targets of *miR-802*-dependent transcriptional silencing in β cells in vitro and in vivo.

***miR-802* affects insulin transcription by targeting *NeuroD1*.** *NeuroD1* was capable of activating both *Ins1* and *Ins2* by binding to their conserved E box elements to function as an important transcription factor in β cell[18,19]. A luciferase reporter assay was performed to confirm that *NeuroD1* could bind to promoter of insulin 2 (Supplementary Fig. 7a). We next investigated whether *miR-802* affected insulin transcription by targeting *NeuroD1* through overexpression and knockdown of *NeuroD1* in Min6 cells (Supplementary Fig. 7b). Functionally, siRNA-mediated

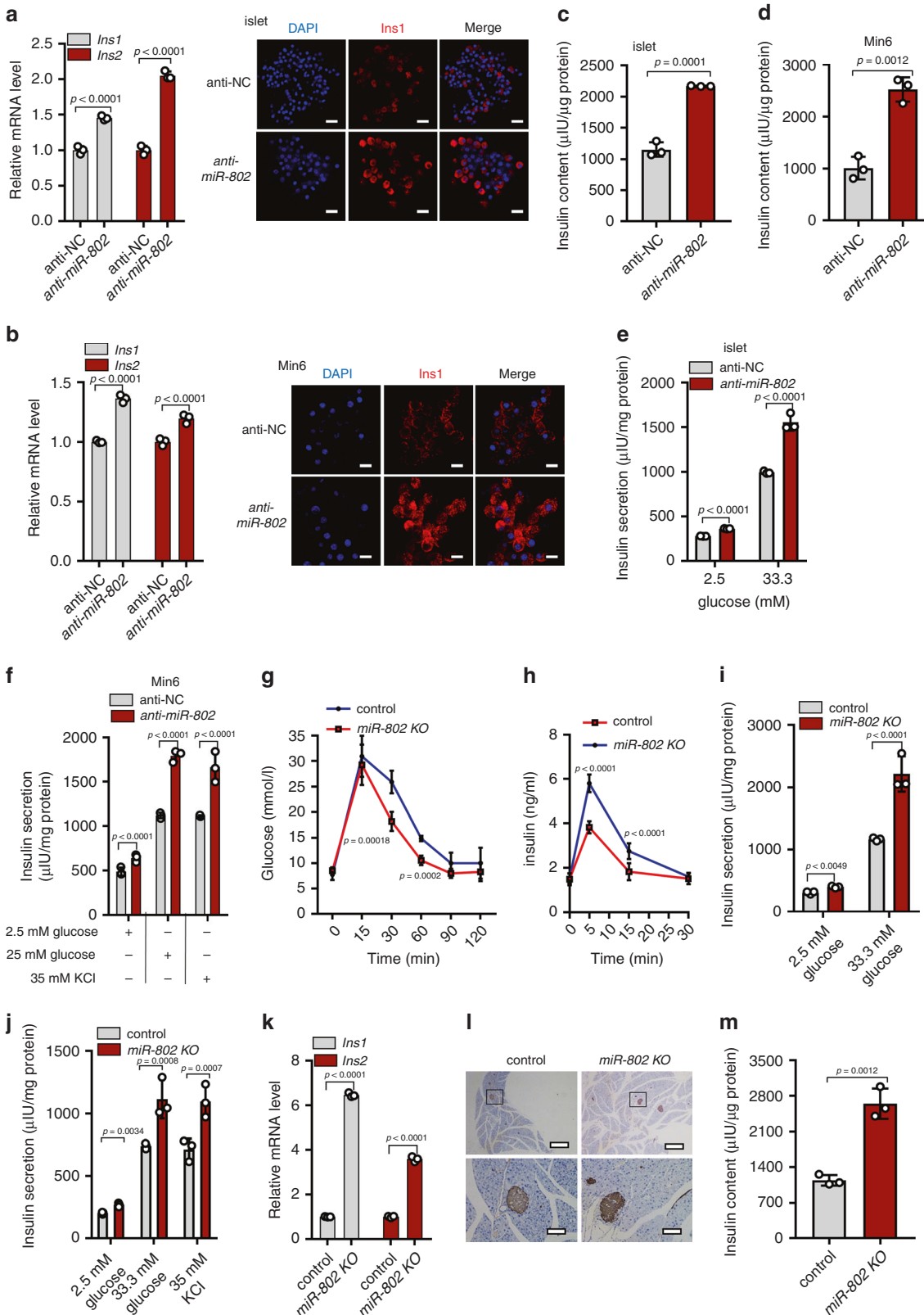

silencing of *NeuroD1* in Min6 cells led to decreased *Ins1* and *Ins2* expression (Fig. 7a). The insulin content was also positively correlated with the expression of *NeuroD1* (Fig. 7b). Rescue experiment result showed that the downregulation of insulin transcription caused by overexpression of *miR-802* was partially reversed by overexpression of *NeuroD1* (Fig. 7c). We next

examined the effect of *miR-802* on the development of β cells in *miR-802 KO* mice and *miR-802 KI* mice at E13.5 and E 17.5 compared them to control mice. Revealing that knockdown *miR-802* could promote β-cell differentiation and pancreatic β-cell numbers were increased in *miR-802 KO* mice (Supplementary Fig. 7c, $n = 3$), while *miR-802 KI* mice achieved the opposite

**Fig. 4 Genetic deletion of *miR-802* results in increased insulin content and secretion.** *anti-miR-802* was transfected into primary islets and Min6 cells for 48 h. Then, qRT-PCR and immunostaining for DAPI (blue) and insulin (red) were performed (Magnification: ×40, scale bar: 20 µm) in islets (**a**), $n = 3$, 8 weeks old) and Min6 cells (**b**), $n = 3$ cell wells), and insulin content was evaluated in islets (**c**), $n = 5$, 8 weeks old) and Min6 cells (**d**), $n = 3$ cell wells). Insulin secretion was analyzed by GSIS assay in islets (**e**), $n = 5$, 8 weeks old) and Min6 cells (**f**), $n = 3$ cell wells). **g** IPGTT (2 g kg$^{-1}$) in overnight fasted *miR-802 KO* and control mice at 10 weeks of age ($n = 4$). **h** In vivo insulin excursions in overnight fasted *miR-802 KO* and controls at 35 weeks of age after IPGTT exposure ($n = 7$). **i, j** Static insulin secretion performed with islets from 5- (**i**) and 35-week-old (**j**) *miR-802 KO* and control mice ($n = 5$) at indicated glucose and 35 mM KCl concentrations. **k** Relative *Ins1* and *Ins2* expression levels in vivo ($n = 5$). **l** Pancreatic sections were immunohistochemically stained for insulin. Magnification: ×4 and ×20, Scale bars: 200 and 20 µm ($n = 3$, 10-week old). **m** Insulin content in in vivo ($n = 5$, 8 weeks old). The *p*-values by two-tailed unpaired Student's *t* test (**c**, **d**, **m**), or two-way ANOVA (**a**, **b**, **e**–**k**) are indicated. All data are represented as mean ± SD, except (**g**, **h**) (mean ± SEM). Source data are provided as a Source Data file.

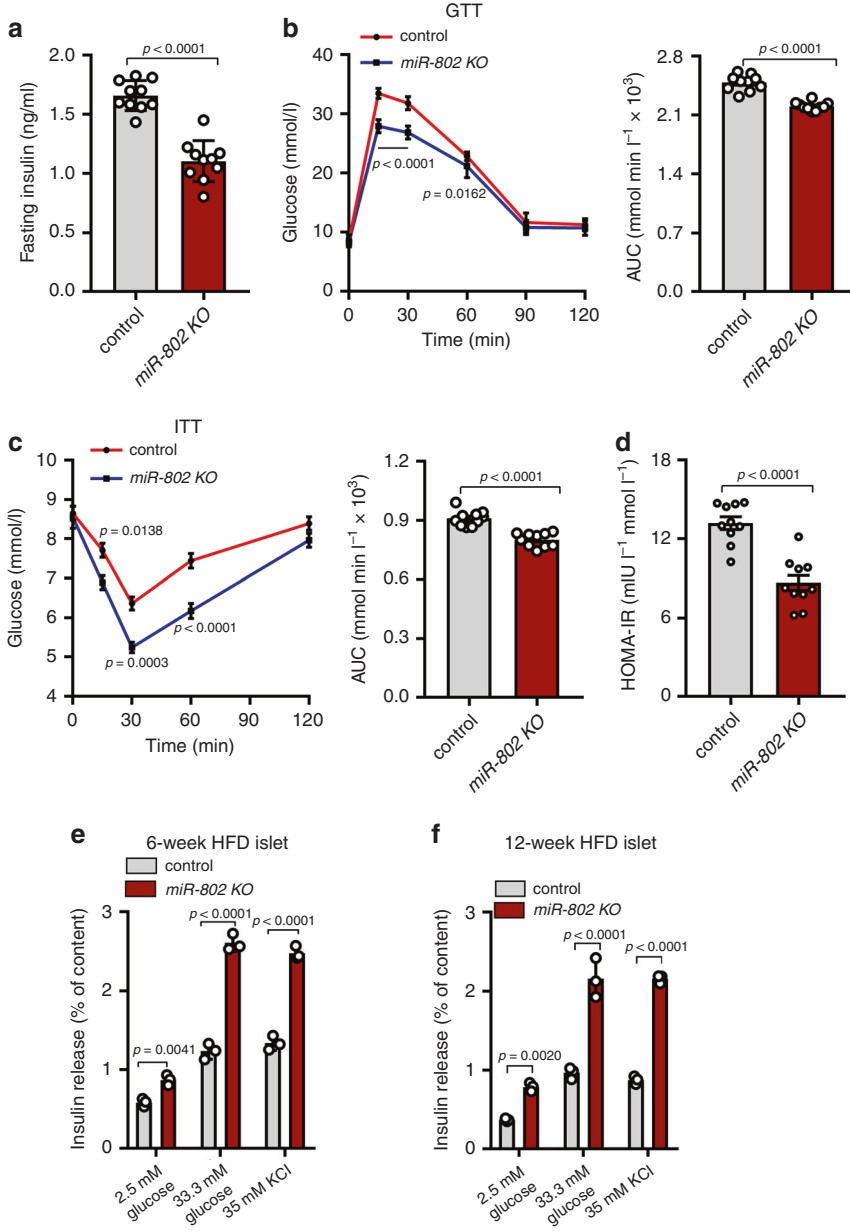

**Fig. 5 *miR-802 KO* mice improves obesity-associated insulin resistance and glucose intolerance when fed with high fat diet (HFD).** 8-week-old male *miR-802 KO* mice and control mice (*miR-802$^{fl/fl}$*) were exposed to HFD for 12 weeks. **a** Then, Fasting insulin levels (FINS) of HFD-fed mice were measured by ELISA after 12-week of feeding HFD ($n = 10$). (**b**, **c**) Intraperitoneal glucose tolerance test (GTT) (1.5 g kg$^{-1}$) (**b**) and intraperitoneal insulin tolerance test (ITT; 0.75 U kg$^{-1}$) (**c**) were performed in *miR-802 KO* mice and control mice at the 11th or 12th week of High fat diet administered, respectively. The corresponding area under the curve (AUC) of blood glucose level was calculated ($n = 10$). **d** Homeostatic model assessment indices of insulin resistance (HOMA-IR) of *miR-802 KO* mice and control mice ($n = 10$). **e**, **f** Insulin release from islets of *miR-802 KO* mice or control mice after 6-week (**e**) or 12-week (**f**) HFD treatment ($n = 5$). The *p*-values by two-tailed unpaired Student's *t* test (**a**, **d**), or two-way ANOVA (**b**, **c**, **e**, **f**) are indicated. In all panels error bars indicate mean ± SEM, except (**e**, **f**) (mean ± SD). Source data are provided as a Source Data file.

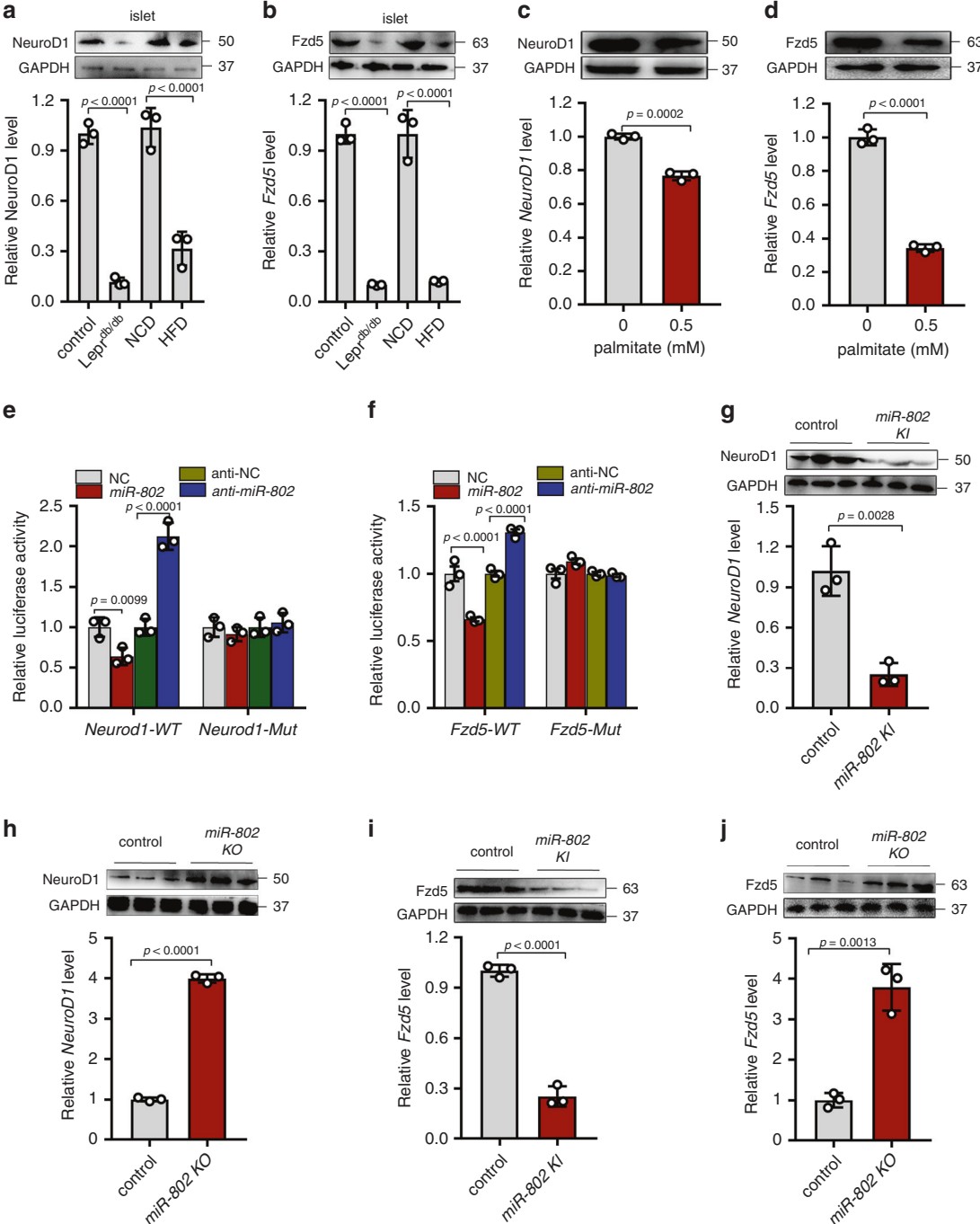

**Fig. 6 Validation of target genes of *miR-802*.** qRT-PCR and western blot were performed to determine the expression of *NeuroD1* (**a**) and *Fzd5* (**b**) in the islets of HFD mice compared to NCD mice and in the islets of Lepr^db/db^ mice at 8-week-old compared to wide type control mice (*n* = 5). Islets were incubated with 0.5 mM palmitate for 48 h, and qRT-PCR and western blot were performed to investigate the expression levels of *NeuroD1* (**c**) and *Fzd5* (**d**). Relative luciferase activity of the firefly reporter constructs containing either the wild type or mutated 3′ UTR of the murine *NeuroD1* (**e**) and *Fzd5* (**f**) gene. Firefly luciferase activity was normalized to the activity of Renilla luciferase. The mRNA and protein levels of NeuroD1 (**g**, **h**) and Fzd5 (**i**, **j**) in the islets of *miR-802* KI and *miR-802* KO mice compared to control (*n* = 6, 8–10 weeks old). All experiments above were performed in triplicates, and each group contained three batches of individual samples. The *p*-values by two-tailed unpaired Student's *t* test (**c**, **d**, **g**–**j**), one-way ANOVA (**a**, **b**), or two-way ANOVA (**e**, **f**) are indicated. Data represent the mean ± SD. Source data are provided as a Source Data file.

effect (Supplementary Fig. 7d, e, *n* = 3). Taken together, these data suggested that *miR-802* affected insulin transcription and β-cell development in a *NeuroD1*-dependent manner.

***Fzd5* restores insulin synthesis and secretion of β cells.** To explore the roles of *Fzd5* on β cell function, we overexpressed or knocked down *Fzd5* in Min6 cells (Supplementary Fig. 8a).

Suppressed *Fzd5* by siRNA decreased insulin transcription and reduced insulin secretion even on stimulation with high glucose and 35 mM KCl (Fig. 8a, b). Dysregulation of *Fzd5* affected the content of insulin too (Fig. 8c). Moreover, the downregulation of insulin transcription caused by knockdown of *Fzd5* was partially reversed by *anti-miR-802* (Supplementary Fig. 8b). The *miR-802*-mediated reduction in insulin secretion was restored by

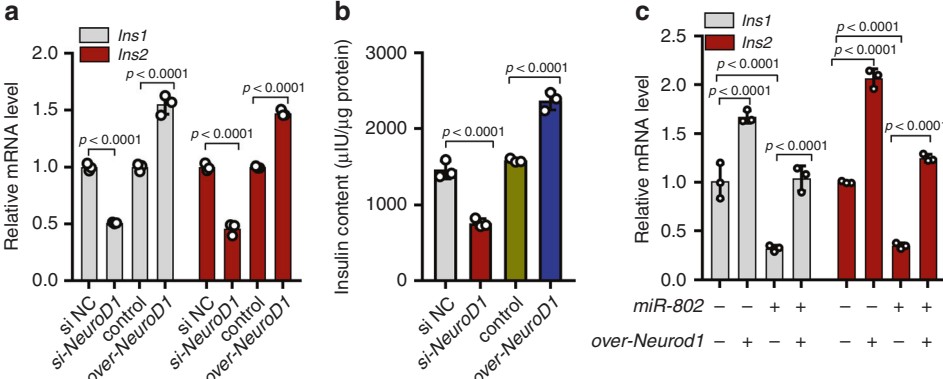

**Fig. 7 miR-802 affects insulin transcription by targeting NeuroD1.** Overexpression and knockdown of *NeuroD1* was conducted in Min6 cells for 48 h. Then, qRT-PCR was performed to analyze the expression levels of *Ins1* and *Ins2* (**a**) and insulin content (**b**). **c** *miR-802*-induced decrease in *Ins1* and *Ins2* levels was partly restored by *NeuroD1* overexpression. All experiments above were performed in triplicates, and each group contained three batches of individual samples. The *p*-values by one-way ANOVA b or two-way ANOVA a and c are indicated. Data represent the mean ± SD. Source data are provided as a Source Data file.

overexpression of *Fzd5* (Fig. 8d). And we observed that the abundance of insulin granules was significantly increased in *miR-802* knockdown group (Fig. 8e). In agreement with these findings, overexpression of *Fzd5* in Min6 cells increased the abundance of insulin granules and *Fzd5* knockdown had opposite effect (Supplementary Fig. 8c).

Based on our findings that *miR-802* regulates insulin secretion and transcription by silencing of *Fzd5*, we first investigated why *miR-802* targeting of *Fzd5* affected insulin secretion. It is well known that *Fzd5* is a receptor of non-canonical Wnt signaling pathway[23,24]. Different independent studies have shown that overexpression of Fzd5 can phosphorylate calcium/calmodulin-dependent kinase II (CamkII) increasing intracellular calcium influx in different kinds of cells[20,25,26]. Corroborating these reports, modified *Fzd5* expression induced changes in CamkII activity in Min6 cells (Fig. 8f). As expected, overexpression of *miR-802* in Min6 cells decreased intracellular $Ca^{2+}$ concentration (Fig. 8g and Supplementary Fig. 8d). In addition, it was observed that overexpression of *miR-802* inhibited $Ca^{2+}$ content in response to 25 mM glucose or extracellular $K^+$ (35 mM, Fig. 8h, i) by FLIPR assays. The above results suggested that *miR-802* induced calcium influx by targeting *Fzd5*, thus causing insulin secretion.

We next investigated why *miR-802* targeting of *Fzd5* altered insulin transcription. It is well known that phosphorylated cAMP response element binding protein (CREB) is an effecter molecule downstream of Fzd5/CamkII pathway[27,28]. After *CamkII* overexpression or treatment with 8 μM *CamkII* inhibitor (KN-93), the phosphorylation level of CREB was upregulated or downregulated in Min6 cell, respectively (Supplementary Fig. 8e). Interestingly, p-CREB levels increased with the increase in p-CamkII expression in *miR-802 KO* mice (Fig. 8j, n = 6), while they decreased with the decrease in p-CamkII expression in *miR-802 KI* mice (Fig. 8k, n = 6). This inhibitory effect of *miR-802* on p-CamkII and p-CREB was recapitulated in the *len-shFoxo1* mice injected with *len-miR-802* (Fig. 8l, n = 6). In addition, we found that after the overexpression or knockdown of *miR-802* in Min6 cells, the expression of various islet-transcription factors was altered and change in *Sox6* expression was the most significant among them (Supplementary Fig. 8f). The dysregulation of Sox6 even occurred in *miR-802 KO* mice and *miR-802 KI* mice (Fig. 8m, n, n = 6). Moreover, we found that the mRNA and protein levels of Sox6 were decreased during overexpression of *CREB* (Supplementary Fig. 8g).

Next, we investigated whether *CREB* could bind to *Sox6* promoter to regulate insulin gene expression. Two binding sites of *CREB* in *Sox6* promoter region were found (Supplementary Fig.

8h top). The overexpression of *CREB* in Min6 cells decreased the transcriptional activity of *Sox6* promoter and the mutation of the Region 1 (R1) binding site ameliorated the positive effect of *Sox6* on gene transcription, while the mutation of Region 2 (R2) showed no such effect as R1 (Supplementary Fig. 8h bottom). These results were confirmed via ChIP strategy (Fig. 8o and Supplementary Fig. 8i). *Sox6* plays an important inhibitory role in insulin biosynthesis through binding to insulin promoter[29,30]. Rescue experiment in Min6 cells showed that *Sox6*-mediated decrease in luciferase activities in Min6 cells was reversed by transfection with over-*CREB* vector (Fig. 8p). These results supported the role of *miR-802* in suppressing insulin gene transcription by promoting the expression of *Sox6* via downregulation of *CREB*.

## Discussion

The relevance between obesity and β cell dysfunction is a fundamental research topic for the development of new therapies increasing insulin secretion and optimizing metabolism of T2D patients. Here, we found that miR-802, overexpressed in the islets of HFD mice and Lepr[db/db] mice, is a key regulator of insulin secretion in pancreatic β cells. We show that overexpressing *miR-802* in β cells impairs insulin transcription and secretion through directly targeting NeuroD1 and Fzd5. Further underlying molecular mechanism research shows that *miR-802* can regulate the phosphorylation of CREB by targeting Fzd5, thereby regulating the expression of Sox6, and ultimately, affecting insulin transcription. Moreover, *miR-802* induces calcium influx by targeting Fzd5, causing insulin secretion (Fig. 9).

To determine the possible causes of the changes in *miR-802* expression detected in the islets of obese mice, we investigate the mechanisms responsible for the induction and activation of *miR-802*. We identify 20 putative *miR-802* transcription factors through Jaspar and Promo, among which *Foxo1* upregulate in response to palmitate treatment in primary islets as well as in obese mice islets[31,32]. Based on previous study, *Foxo1* could involve in insulin resistance and dysregulated glucose metabolism in the liver[33,34]. But in the pancreas, *Foxo1* was suggested as a double-edged sword[35], and the molecular mechanism underlying its ability to regulation obesity in the pancreas remains largely unknown. *Foxo1*, a transcription factor, can activate and repress gene expression through binding to the genes' promoter regions[32,36]. Consistent with that in the islet, *Foxo1* can bind to *miR-802* promoter via luciferase reporter, EMSA and ChIP assays.

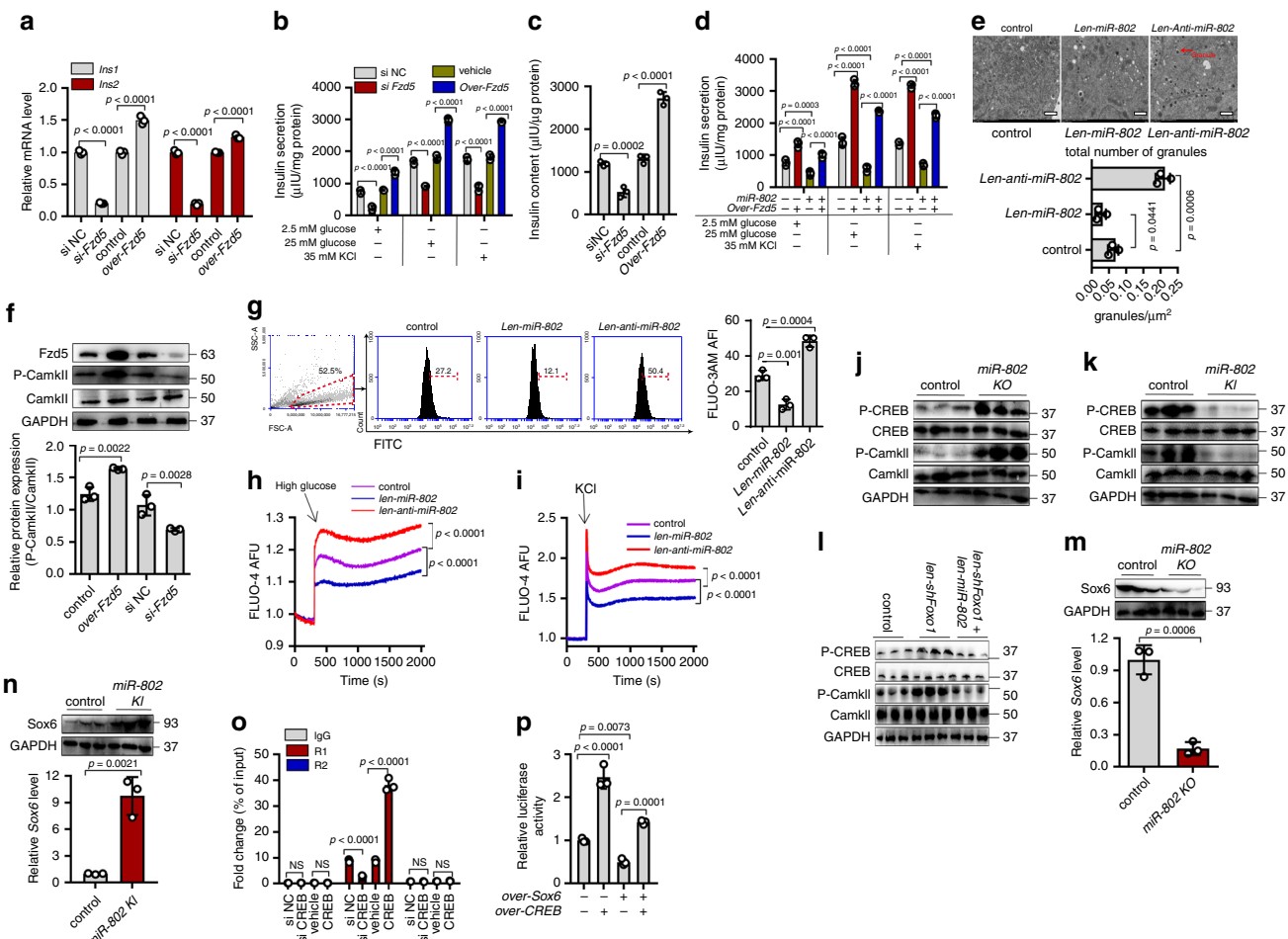

**Fig. 8 Silencing of the *miR-802* target, *Fzd5*, impairs insulin gene expression and insulin secretion. a** *Si-Fzd5* or *over-Fzd5* plasmids was transfected into Min6 cells for 48 h. qRT-PCR was conducted to determine the insulin synthesis, insulin secretion was analyzed by GSIS assay (**b**) and insulin content was evaluated using ELISA in Min6 cells (**c**). **d** *miR-802*-induced decrease in insulin secretion was partly restored by *Fzd5* overexpression. **e** Overexpression and knockdown *miR-802* in Min6 cell by lentivirus, granules were measured by electron microscopy. Scale bars: 20 μm. **f** Phosphorylation level of CamkII was increased when *Fzd5* was overexpressed in Min6 cells. **g** Overexpression and knockdown of *miR-802* in Min6 cells for 48 h; flow cytometry was conducted to investigate the intracellular $Ca^{2+}$ concentration, the numbers represent the percentage of $FITC^+$ cells. $Ca^{2+}$ concentration were examined by FLIPR induced by 25 mM glucose (**h**) and 35 mM KCl (**i**). **j–l** Western blot was performed to determine the phosphorylation level of CamkII and CREB in the islets of *miR-802 KO* mice (**j**, $n = 6$, 8–10 weeks old), *miR-802 KI* mice (**k**, $n = 6$, 8–10 weeks old), and *len-miR-802* and *len-shFoxo1* mice (**l**, $n = 5$, 17–18 weeks old). qRT-PCR and western blot were performed to evaluate the mRNA and protein levels of Sox6 in the islet of *miR-802 KO* mice (**m**, $n = 6$, 8–10 weeks old) and *miR-802 KI* mice (**n**, $n = 6$, 8–10 weeks old). **o** ChIP experiment showed that *CREB* binds to the promoter of *Sox6* by qRT-PCR assays. Only binding site R1 showed positive results. **p** Min6 cells were co-transfected with *Sox6* and insulin promoter luciferase reporter plasmid for 24 h. Relative luciferase activity of the firefly reporter was analyzed. All experiments above were performed in triplicates. Each group contained three batches of individual samples, The *p*-values by two-tailed unpaired Student's *t* test (**e**, **m**, **n**), one-way ANOVA (**c**, **g**, **f**, **h**, **i**, **p**) or two-way ANOVA (**a**, **b**, **d**, **o**) are indicated. Values are represented as mean ± SD. Source data are provided as a Source Data file.

MiRNAs, which are stable in the serum and plasma[37], are possibly transferred from donor cells to recipient cells where they alter the gene expression of recipient cells. In the present study, we find that *miR-802* expression in the serum of *miR-802 KI* mice is increased, suggesting that *miR-802* may play potential roles in tissues' communication, such as liver. For instance, previous study had reported that obesity-induced hepatic *miR-802* overexpression, which caused insulin resistance and impairing glucose metabolism in vivo[16].

To gain further insights into the mechanism of *miR-802*-mediated regulation of β cell function, we use a stringent bioinformatics approach to identify *miR-802* target genes. Bioinformatics analysis reveals that NeuroD1 and Fzd5 may be the target genes of *miR-802*. NeuroD1 is an islet-transcription factor, which increases insulin production by targeting insulin promoter[38–40], and NeuroD1 is also known to be critical for β-cell differentiation

during embryogenesis[41,42]. Moreover, the mRNA expression levels of insulin gene transactivators such as *Pdx1*, *Pax6*, *Nkx6.1* and *Mafa* were decreased by *miR-802*, which indicate that miR-802 plays a critical role in dedifferentiation of pancreatic β cells. Deregulation of a subset of transcriptional regulators in pancreatic β cells is underlined a fundamental cause of β cell dysfunction leading to T2D[43]. Together, these findings suggest that deregulation of *miR-802* represents a central cause of compromised insulin production and reduces expression of transcription regulators maintaining β cell maturity.

CamkII is a key mediator and feedback regulator of $Ca^{2+}$ signaling pathway in β cells and promotes $Ca^{2+}$ influx[44], which plays a critical role in the promotion of insulin secretion[45,46]. Here, we find that Fzd5 is a novel *miR-802* target, which can mediate at least part of the effects of *miR-802* on insulin secretion via increasing intracellular $Ca^{2+}$ content. In this context, *miR-802*

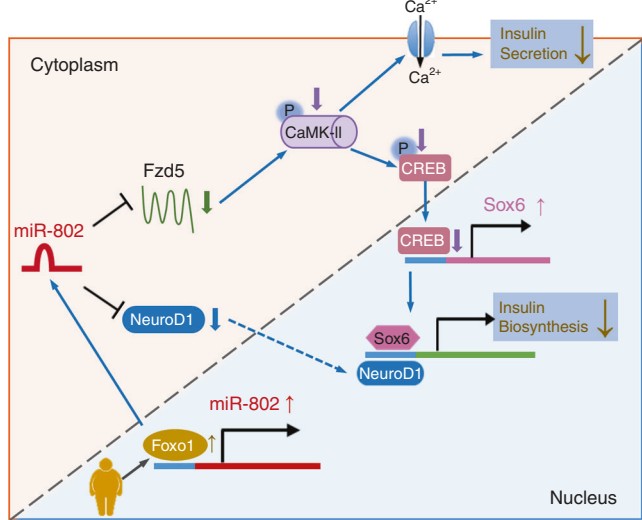

**Fig. 9 A working model that illustrates the mechanism by which silencing of *miR-802* impairs β cell function-targeted *NeuroD1* and *Fzd5*.** Obesity predisposes to type 2 diabetes, but the mechanisms of obesity-associated beta-cell dysfunction are incompletely understood. Here the authors report that obesity increases the levels of miR-802, which impairs insulin transcription and secretion by targeting NeuroD1 and Fzd5.

represents a unique miRNA molecule for its ability to suppress $Ca^{2+}$ influx, which reduces the release competence of secretory granules and impairs secretory robustness to pancreatic β cells.

Interestingly, we also determine that *miR-802* suppresses insulin gene expression by inhibiting $Ca^{2+}$ signaling pathway. Welsh et al.[47] reported that $Ca^{2+}$ signaling pathways could also affect the rates of insulin mRNA translation, but the underlying mechanism was still elusive. CREB[28], which plays an important role in insulin gene transcription[48], is the downstream gene of CamkII. Since the mechanism of *CREB*-mediated regulation of insulin synthesis is not clear, the downstream mechanism by which *CREB* promotes insulin synthesis is investigated. In our study, we show that Sox6 is suppressed by transfection of over-CREB. Dual luciferase analyses and Chip strategy further demonstrate that *CREB* binds to *Sox6* promoter. *Sox6*, an important islet-transcription factor, binds to insulin promoter to inhibit insulin synthesis[29]. These results indicate that knockdown of *miR-802* contributes to insulin synthesis by suppressing $Ca^{2+}$ signaling pathway via suppression of *CREB* binding to *Sox6* promoter.

In conclusion, our study reveals an important role of *miR-802* in the development of obesity-associated β cell dysfunction. It is observed that overexpression of *miR-802* downregulates insulin transcription and secretion and impairs glucose tolerance, clearly indicating a functional role of *miR-802* in the development of obesity-associated β cell dysfunction. Therefore, our present study characterizes *miR-802* and its target gene(s), which unveil novel research avenues for the treatment of obesity-associated T2D.

## Methods

**Animal care**. Lepr$^{db/db}$, Lepr$^{db/-}$ and C57BL/6 J male mice (7–8 weeks old) were obtained from Model Animal Research Center of Nanjing University (Nanjing, China). RIP-Cre mice were obtained from Jackson laboratory (003573-B6.Cg-Tg (Ins2-cre) 25Mgn/J). All animals were of pure C57BL/6 background except Lepr$^{db/db}$ and Lepr$^{db/-}$ mice, which were of BKS background. C57BL/6 J mice were fed high fat diet (HFD) for 8 weeks (D12494, 60% energy from fat) according to the criteria defined by Peyot ML[49], and weighted between 40 g and 45 g. The control groups were fed with normal diet (D12450J, 10% energy from fat), and weighted between 23 g and 25 g. Mice were housed in groups of 3–5 animals/cage on a 12 h light/dark cycle in an SPF facility at 22-24 °C, 40-50% humidity. All care and handling of animals were carried out according to the international laws and

policies (EEC Council Directive 86/609, 1987) and approved by the animal ethics committee of China Pharmaceutical University (Nanjing, China). Permit Number: 2162326. Care of animals was done within institutional animal-care committee guidelines.

**Generation of *miR-802* knockout and knock-in mice**. We generated *miR-802* conditional knockout and knock-in mice via CRISPR/Cas9 system from Model Animal Research Center of Nanjing University (Nanjing, China). For *miR-802* knockout mice, two sgRNAs-targeting the introns on both sides of the floxed region of *miR-802* were constructed and transcribed in vitro (two sgRNA sequences were listed in Supplementary Table 1). The donor vector with the loxp fragment was designed and constructed in vitro. For *miR-802* knock-in mice, one sgRNA-targeting the near sequence of inserted site was constructed and transcribed in vitro (one sgRNA sequence was listed in Supplementary Table 1). The donor vector with the inserted fragment was designed and constructed in vitro. Then, Cas9 mRNA, sgRNA, and donor of each model mice were co-injected into 0.5 day' zygotes of C57BL/6 J mice. Thereafter, 400 zygotes were transferred into the oviduct of pseudopregnant ICR females at 0.5 days post-copulation. 61 *miR-802* knockout F0 mice and 109 *miR-802* knock-in F0 mice were born after 19–21 days of transplantation and all the offspring of ICR females (F0 mice) were identified by PCR and sequencing of tail DNA. The primers are listed in Supplementary Table 2. Finally, 2 positive *miR-802* knockout F0 mice and 9 positive *miR-802* knock-in F0 mice were crossed with C57BL/6 J mice to build up *miR-802* knockout and knock-in heterozygous mice (*miR-802$^{fl/wt}$*, *miR-802$^{ki/wt}$*). Homozygote *miR-802* knockout and knock-in mice (*miR-802$^{fl/fl}$*, *miR-802$^{ki/ki}$*) were obtained from backcross-intercross of *miR-802$^{fl/wt}$* or *miR-802$^{ki/wt}$*. Then *miR-802$^{fl/fl}$* or *miR-802$^{ki/ki}$* mice were crossed with Rip-Cre mice to select Rip-Cre *miR-802$^{fl/wt}$* or Rip-Cre *miR-802$^{ki/wt}$*. Finally, these mice were crossed with *miR-802$^{fl/fl}$* or *miR-802$^{ki/ki}$* mice to delete or overexpress *miR-802* in β cells, respectively. Off-target effects were identified by PCR and sequencing of tail DNA for 10 Rip-Cre *miR-802$^{fl/fl}$* (*miR-802 KO*) and 10 Rip-Cre *miR-802$^{ki/ki}$* (*miR-802 KI*) mice. The primers are listed in Supplementary Table 2.

**Serum samples of lean and overweight individuals**. The serum and clinicopathological data were collected from the Zhongda Hospital, Affiliated to southeast University (Nanjing, China). All the patients enrolled in this study had obesity (BMI > 25). The negative controls were represented by lean individuals ($20 \leq BMI \leq 25$). All human subjects provided informed consent. All human studies were conducted according to the principles of the Declaration of Helsinki, were approved by the Ethics Committees of the Department Zhongda Hospital Southeast University (Nanjing, China, 2018ZDSYLL132-P01). The clinical features of the patients are listed in Supplementary Table 3.

**Southern blotting**. Tail biopsies were processed in Sarkosyl-based buffer supplemented with 50 mg mL$^{-1}$ Proteinase K at 56 °C. The supernatants were extracted once with phenol/chloroform/isoamyl alcohol solution (Sigma) before precipitation with 2 volumes of ethanol. Genomic DNA was digested at 37 °C for 15 h, loaded on 0.8% Tris-Acetate-EDTA agarose gels, and transferred to nylon blotting membrane (BioRad). Membranes were UV-cross-linked and incubated at 85 °C for 1 h. $^{32}P$Radialoabeled PCR probes were prepared by Prime-It Random Primer Labeling kit (Agilent Technologies) and purified by G50 gel filtration (GE Healthcare). Membranes were incubated with specific probes for 16 h at 60 °C, washed, and exposed to screen.

**Islet dispersion and insulin secretion cells**. Mouse pancreatic islets were incubated in $Ca^{2+}$ and $Mg^{2+}$ free PBS for 28 min at 37 °C and centrifuged for 5 min at 200 g. Supernatant was removed and islets were resuspended in KRBH balanced buffer (115 mM NaCl, 4.8 mM KCl, 2.5 mM CaCl$_2$, 1.2 mM MgSO$_4$, 1.2 mM KH$_2$PO$_4$, 20 mM NaHCO$_3$, and 16 mM HEPES; pH 7.4) containing 0.2% BSA supplemented with 2.5 mM glucose, 10% FBS, 100 IU mL$^{-1}$ penicillin, 100 μg mL$^{-1}$ streptomycin, 2 mM L-glutamine and 10 mM nicotinamide, and seeded in plate conditioned medium (96-well plate contains 30 islets for glucose-stimulated insulin secretion (GSIS), 48-well plate contains 100 islets for RNA, and 6-well plate contains 200 islets for western blot). Then, dispersed islet cells were transfected with either *miR-802* mimic or *miR-802* inhibitor using Lipo2000 according to the manufacturer's instructions for further studies.

The mice pancreatic β cell line Min6 (passage 20-30) was maintained in DMEM (Gibco) containing 15% FBS (Gibco, Burlinton, ON, USA), 100 IU mL$^{-1}$ penicillin, 100 μg mL$^{-1}$ streptomycin, and 50 μM β-mercaptoethanol (Sigma Aldrich, St. Louis, MO, USA) at 37 °C in humidified atmosphere containing 5% CO$_2$.

For palmitate treatment, islets and Min6 cells were incubated in 0.5 mM palmitate (Sigma Aldrich). Palmitate (200 mM) was dissolved in ethanol, filter sterilized, diluted 1:22.2 in 10% BSA (9 mM palmitate). Before use, 9 mM palmitate-BSA stock was diluted 1:18 in serum-free DMEM to achieve 0.5 mM palmitate in the presence of 0.5% (wt vol$^{-1}$) BSA.

For GSIS, Min6 cells ($2\times10^5$ cells well$^{-1}$ in 48-well plate) were preincubated overnight in KRBH balanced buffer containing 0.2% BSA supplemented with 2.5 mM glucose and 10% FBS, and were incubated for 2 h in 2.5 mM, 25 mM glucose or 35 mM KCl. The insulin levels were measured by mice insulin ELISA kit

(Crystal Chem, USA), according to the manufacturer's instructions. The amount of insulin secretion was normalized by the total cellular protein content. The protein contents of Min6 cells were quantified by BCA kit.

**Glucose-stimulated insulin secretion (GSIS) in mouse Islets**. Mouse pancreatic islets (30 islets well$^{-1}$ in 48-well plate) were collected under a stereomicroscope at room temperature and cultured in 2.5 mM glucose in the absence or presence of test agents for 48 h. Thereafter the islets were washed and preincubated for 30 min at 37 °C in KRBH buffer, supplemented with 0.2% bovine serum albumin and 2.5 mM glucose. After preincubation, the buffer was changed to a medium containing either 2.5 mM, 33.3 mM glucose or 35 mM KCl. The islets were then incubated for 1 h at 37 °C. Immediately after incubation an aliquot of the medium was removed for analysis of insulin via mice insulin ELISA kit (ExCell Bio, Shanghai, China), and the islets were lysed to extract total protein, the amount of insulin secretion was normalized by the total cellular protein content[50].

**Glucose-tolerance tests in vivo**. Glucose-tolerance tests were carried out in mice after a 12 h overnight fasting. After determination of fasted blood glucose levels, animals received an intraperitoneal (i.p.) bolus of 2 g glucose kg$^{-1}$ body weight. Blood glucose levels were determined after 15, 30, 60, and 120 min. Serum sample was collected from eye canthus blood at 0, 5, 15, and 30 min. Insulin level was evaluated using mice insulin ELISA kit (Crystal Chem, USA), according to the manufacturer's instructions.

**In vivo experiments**. Male C57BL/6 J mice (8 weeks old, about 23–25 g) were randomly divided into 3 groups ($n = 10$). Lentivirus-shFoxo1 (Len-shFoxo1) was constructed from GenePharma (shanghai, China). Then, 1 ×10$^9$ lentivirus-shFoxo1 dissolved in 0.2 mL normal saline (NS) was injected intravenously into each mouse through the tail vein. Lentivirus-LV3 (pGLV-H1-GFP + Puro) at the same concentration was used as a control. At 72 h after injection, islets (100 islets for RNA and 200 islets for protein) were lysed to extract total RNA or protein to measure the knockdown efficacy. And the expression level of miR-802 was analyzed by qRT-PCR. Then mice were fed with high fat diet (HFD, D12494, 60% energy from fat) for 8 weeks.

**Measurement of intracellular Ca$^{2+}$ concentration**. Forty-eight hours after transfection, the culture medium of the cells in the 96-well clear-bottomed black plates was removed and replaced with dye loading buffer (100 μL well$^{-1}$) containing 4 μM fluo-4. After incubation for 45 min, cells were washed with Locke's buffer for five times using an automated cell washer (BioTek Instruments, Winooski, VT), reaching a final volume of 150 μL in each well. Then, the plate was transferred to a fluorescence laser plate reader (Molecular Devices, Sunnyvale, CA) chamber. Fluorescence reading was taken for 5 min to establish the baseline, and then, 25 μL of the test compound solutions (8×) were added to corresponding wells. After that, the fluorescence readings were taken for 30 min. Background fluorescence was automatically subtracted from all fluo-4 fluorescence measurements.

Min6 cells were transfected with lentivirus miR-802 (len-miR-802) or lentivirus anti-miR-802 (len-anti-miR-802) for 48 h. The intracellular calcium contents were measured by Fluo-3M (Abcam, ab145254), according to the manufacturer's instructions, using flow cytometry and immunofluorescence.

**RNA-sequencing and analysis**. Total RNA from islets of NCD, HFD, wide type control mice, and Lepr$^{db/db}$ mice was isolated using the RNeasy mini kit (Qiagen), following the protocol for total RNA isolation. RNA integrity and concentration were assessed using the RNA Nano 6000 Assay Kit of the Bioanalyzer 2100 system (Agilent Technologies, CA, USA), according to the manufacturer's instructions. The expression level of miRNAs was measured by "Transcripts Per Kilobase Million" (TPM). Differentially expressed miRNAs were analyzed using DESeq package (DESeq version 1.26.0) based on the negative binomial distribution test[51], which were considered differentially expressed with an adjusted $p$-value ≤ 0.05 and the absolute value of log 2 (fold change) > 1 was used to identify significantly differentially expressed genes. The raw data is presented in Supplementary Table 4 and 5.

**Plasmid construction**. The coding sequences for Foxo1, Sox6, Fzd5, NeuroD1, CamkII, and CREB were amplified by PCR from full-length cDNA of mice, and then cloned in pcDNA 3.1 vector. The primer sequences for PCR are listed in Supplementary Table 6.

The sgRNAs of miR-802 promoter were constructed in lentiCRISPRv2 puro vector according to the Zhang F libraries[52]. The lentiCRISPRv2 puro vector was digested with BbsI and ligated with annealed oligonucleotides. The sgRNA primer sequences are listed in Supplementary Table 7A.

The mouse Sox6 promoter, upstream 2 kb, and mouse miR-802 promoter, upstream 3 kb (the sequence was obtained from UCSC), were obtained by PCR and the mutation site primers are listed in Supplementary Table 6. The PCR product was cloned into the PGL3-basic vector (Promega, Madison, WI).

**Chromatin immunoprecipitation (ChIP) experiment**. Min6 cells, transfected with ADA-Foxo1, si-Foxo1, CREB, sh-CREB, were fixed with 37% formaldehyde for 10 min, followed by 30 rounds of sonication each for 3 s to fragment the chromatin. The chromatin was incubated with the anti-Foxo1 antibody (ab39670, Abcam) or anti-CREB antibody (ab31387, Abcam) at 4 °C overnight, and then, immunoprecipitated with Proteinase K (Millipore). Purified DNA was amplified by PCR using primer pairs that spanned the predicted Foxo1 or CREB binding sites. Primer sequences are listed in Supplementary Table 6.

**Electrophoretic mobility shift assays (EMSA) analysis**. The probes, an approximately 25 bp fragment included the binding sizes, were biotin end-labeled referring to the instructions of the Biotin 3′ End DNA Labeling Kit (Thermo Pierce, Massachusetts, USA) and then annealed to double-stranded probe DNA. Foxo1-DNA complexes were performed according to the instructions of the LightShift Chemiluminescent EMSA kit (Thermo Fisher Scientific, Massachusetts, USA). Probe sequences are listed in Supplementary Table 6.

**siRNA for Protein Expression Silencing in cells**. Small interfering RNAs (siRNAs) were designed and synthesized by Ribobio (Guangzhou, Guangdong, China). For transient transfection, isolated mouse islets or Min6 cells (~5 ×10$^5$) were seeded in six-well plates cultured in media without antibiotics and transfected with Lipofectamine 2000 reagent (Invitrogen) according to the manufacturer's instructions. Cells were transfected for 6 h with the Fzd5, NeuroD1 and Foxo1 siRNA at a final concentration of 50 nM or with control siRNA (non-targeting siRNA) at the same concentration before changing to fresh media including antibiotics. At 48 h after transfection, cells were lysed to extract total RNA or protein to measure the knockdown efficacy. Sequences of shRNA and siRNA were listed in Supplementary Table 7.

**Plasmid and Transient Transfections**. Min6 cells (~5 ×10$^5$) were seeded in six-well plates in culture medium without antibiotics and transfected with full-length cDNA encoding Fzd5, NeuroD1, CREB, Sox6, ADA-Foxo1 or control plasmid (non-coding) using Lipofectamine 2000 reagent (Invitrogen) according to the manufacturer's instructions. 6 h post-transfection, medium was replaced with fresh medium containing antibiotics. At 48 h post-transfection, the cells were harvested and analyzed by immunoblotting and qPCR for the relative level of various proteins and mRNA. All the plasmid used were listed in Supplementary Table 8.

MiR-802 duplex mimics (50 nM), 2′-O-methylated single-stranded miRNA antisense oligonucleotides (anti-miR-802, 100 nM), and negative controls at the same concentration were obtained from GenePharma (Shanghai, China). For transient transfection, Lipofectamine 2000 reagent (Invitrogen) was mixed with miRNA mimics/inhibitors according to the manufacturer's instructions[53], mimics NC or inhibit NC was transfected at the same concentration as negative control. At 48 h post-transfection, the cells were harvested and analyzed by qPCR for the relative level of miR-802.

**Virus (Lentivirus) infection**. Min6 cells were seeded in six-well plates at a density of about 5 ×10$^5$ cells with Lentivirus encoding either miR-802 (len-miR-802, 5 μL mL$^{-1}$) or miR-802 sponge (len-anti-miR-802, 5 μL mL$^{-1}$) plus Polybrene for 48 h. For comparison, scramble (lentiviral particles without targeting any specific region) served as control.

**Luciferase assays**. Min6 cells were plated at the concentration of 10,000 cells well$^{-1}$ in 24-well plates 24 h before transfection. The cells were then transfected with 0.9 μg DNA well$^{-1}$ (0.4 μg construct promoter and 0.4 μg transcription factors; 0.1 μg constitutive renilla expression plasmid as a control for transfection efficiency). Cells were harvested 24 h after transfection and luciferase activities were measured using a dual-luciferase reporter assay system (Promega).

For the generation of reporter constructs, the complete 3′-UTR of murine Fzd5 and NeuroD1 containing either the wild type or mutated miR-802 binding sites was cloned behind the stop codon of the firefly luciferase open reading frame using specific primers. Min6 cells were plated in 24-well plate with 100 ng pmir-PGLO reporters along with miR-802 mimic or miR-802 inhibitor using Lipofectamine 2000 transfection reagent (Invitrogen), according to manufacturer's instructions. Dual luciferase reporter assays were performed 24 h after transfection using a Luciferase Assay System (Promega). Transfection data represent at least three independent experiments each performed in triplicates.

**Real-time PCR**. Islets (100 islet per group) and Min6 cells were isolated using TRIZOL (Invitrogen), cDNA was generated using HiScript Q RT SuperMix for qPCR (Vazyme, China) and real-time PCR assays were conducted with a LC480 Light Cycler (Roche, Germany) using the applied primer sequences listed in Supplementary Table 9. Relative expression of genes was determined using a comparative method (2$^{-\triangle CT}$). U6 and GAPDH were used as internal standards for miRNAs and mRNAs, respectively. For miR-802 and U6, TaqMan probes (Ambion) were used to confirm our results.

**Cell counting kit-8 (CCK-8) assay**. Min6 cells were seeded in 96-well plates ($4 \times 10^4$ cells well$^{-1}$) in 100 μL culture medium. CCK-8 assay (Vazyme, Nanjing, Jiangsu, China) was performed at 0, 24, 48, and 72 h after transfection, according to the manufacturer's instructions.

**Flow cytometry**. Forty-eight hours after transfection, cells were treated with EDTA, according to the instructions of Annexin V-FITC Apoptosis detection kit (KeyGEN BioTECH, Nanjing, Jiangsu, china). The FITC and PI were examined at 488 nm and 630 nm for the evaluation of cell apoptosis, FACS data was analyzed using FlowJo v10 software.

**Electron microscopy**. Min6 cells, transfected with either lentivirus *miR-802* (*len-miR-802*) or lentivirus *anti-miR-802* (*len-anti-miR-802*) for 48 h, were fixed in Karnovsky's fixative at 4 °C overnight, incubated with 1% osmium tetroxide in 0.1 M cacodylate buffer for 30 min, washed three times, dehydrated, embedded in eponate, and processed for transmission electron microscopy (HITACHI, HT7700). Granule number was determined using Metamorph software and normalized to the area of the footprint (vesicles mm$^{-2}$).

**Western blot**. The whole cell lysate and nuclear-protein fractions were isolated from tissues or cultured cells using a protein extraction kit (Beyotime, Shanghai, China). The total proteins from indicated tissues, primary islets (200 islets per group) and Min6 cells, were extracted in RIPA Lysis Buffer (Beyotime) containing 1% PMSF (Sigma). Western blot analyses were conducted according to standard procedures using specific antibodies. The antibodies are listed in Supplementary Table 10. Densitometric calculations were expressed as fold change in proteins relative to GAPDH expression levels by ImageJ software.

**Immunohistochemistry and immunofluorescence**. Pancreas were fixed in 4% paraformaldehyde and embedded in paraffin, and the antigen from the cut sections were retrieved by boiling them in 10 mM Tris/EDTA (pH 9.0). Sections were permeabilized and blocked in PBS buffer containing 0.1% Triton X-100, 1% BSA, and 5% goat serum. Primary antibody binding was performed overnight at 4 °C, while incubation with secondary antibody was done at room temperature for 1 h. Images were acquired on LSM510 Zeiss confocal microscope (Zeiss) at ×20 or ×40 magnification. The antibodies are listed in Supplementary Table 10.

**Morphometry**. For morphometric analysis of pancreas, 8-μm sections were cut, and five sections, at least 180 μm apart, were taken from each mouse for a different type of analysis; at least 3 mice per group were analyzed in every case. For determining β cell mass, sections were stained with anti-insulin and anti-glucagon antibodies and DAPI, the pancreatic sections were scanned entirely using a ×10 objective of a Zeiss AxioVert 200 microscope. The fraction of the insulin- or glucagon-positive areas were determined using Image J, and the mass was calculated by multiplying this fraction by the initial pancreatic wet weight.

The number of β-cells/pancreas area was determined from counts of β-cells (DAPI$^+$ nuclei in insulin$^+$ area) on sections of 10–20 islets/mouse[54].

**Mouse metabolic assays**. Mouse metabolic assays were performed. Briefly, body weight, fasting blood glucose (FBG) levels and fasting serum insulin (FINS) levels were examined after a 6 h fasting treatment by using a glucometer (OMRON, Japan) and by ELISA (ExCell Bio, Shanghai, China), respectively. And the homeostatic model assessment indices of insulin resistance (HOMA-IR) was calculated with the equation (FBG (mmol L$^{-1}$) x FINS (mIU L$^{-1}$))/22.5. To perform the glucose tolerance tests, 1.5 g kg$^{-1}$ glucose (Sigma-Aldrich, St Louis, MO, USA) was i.p. injected into mice, whereas 0.75 U kg$^{-1}$ insulin (Novolin R, Novo Nordisk, Bagsvaerd, Denmark) was i.p. injected into mice for insulin tolerance tests. Blood glucose levels were examined at 0, 15, 30, 60, and 120 min after injection. The AUC are given as the incremental area under the curve, calculated by the conventional trapezoid rule.

**Statistical analysis**. Two-tailed unpaired Student's *t* test was used to show the significant difference between 2 groups, and ANOVA was used for multigroup difference analysis. Dunn's multiple comparisons for one-way ANOVA and Fisher's least signicant difference (LSD) for two-way ANOVA were used. The level of significance was set at $^*p < 0.05$, $^{**}p < 0.01$, $^{***}p < 0.001$. Correlations between *miR-802* expression and BMI of individuals were performed by linear regression. Data were expressed as ± SD or ± SEM. Graphpad prism 7 was used for all calculation.

**Reporting summary**. Further information on research design is available in the Nature Research Reporting Summary linked to this article.

## Data availability

All relevant data supporting the key findings of this study are available within the article and its Supplementary Information files or from the corresponding author upon reasonable request. The source data underlying Fig. 1c–i, 2a–g, 2j–l, 3a–c, 3l–q, 4a–k, 4m, 5–7, 8a–i, 8m–p and Supplementary Figs. 1a-f, 1k, 2a-b, 2d, 2f, 2j, 3a-c, 3k-l, 3m-t 4a-c,

4k-n, 4p-q, 5a-d, 6d-g, 7a-c, 8a-c, 8e-h are provided as a Source Data file. The RNA-seq raw data that support the findings of this study has been deposited in the NCBI's Sequence Read Archive (SRA) database (PRJNA577478 [https://www.ncbi.nlm.nih.gov/bioproject/PRJNA577478/], and PRJNA577480 [https://www.ncbi.nlm.nih.gov/bioproject/PRJNA577480/]). A reporting summary for this article is available as a Supplementary Information file.

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

## Acknowledgements

This work was supported by National Natural Science Foundation of China (Grant No. 81570696); Supported by sponsored by Qing Lan Project; supported by grants from the "111" project (B16046); Supported by ministry of Science and Technology of China (2018ZX09201018-005 to X.F.); Supported by National Natural Science Foundation of China (81970561, 81570527 and 91540113 to X.F.).

## Author contributions

F.F.Z, D.S.M, W.L.Z, D.W.W, T.S.L, Y.H.L, Y.Y, Y.L, J.M.M, and B.B.L performed the experiments; Y.F.Z, Y.P, C.Y.G, H.D., and L.L analyzed data; F.F.Z, X.H.F, Z.Y.C, and L.J designed the project, interpreted the data and wrote the manuscript.

## Competing Interest

The authors declare no competing interests.
