## [Peer Review File · Nature Communications]

Reviewers' Comments:

Reviewer #1:

Remarks to the Author:

To authors,

The manuscript reported an investigation showing that alternation of islet miR802 partially responsible for obesity induced β cell dysfunction. The manuscript presented a large amount of data from both in vivo and in vitro studies to elucidate the function of miR802 in pancreatic β cells. The findings are interesting, but more experiments should be performed to strengthen their conclusions.

Major concerns,

1. A series of FoxO1 studies had suggested the important role of FoxO1 in pancreas. However, the functions of FoxO1 in pancreas is still controversial, and FoxO1 was suggested as "a double-edged sword" in the pancreas. In my opinion, a major innovation of this study is that they discovered the miRNAs regulated by FoxO1 during obesity-induced β cell dysfunction and had broadened our current knowledge about FoxO1 function in β cells. Therefore, it is important to verify whether obesity induced miR802 upregulation is depend on FoxoO1 activation in vivo. The authors should detect miR802 alternation in inducible β -cell specific FoxO1 KO mice with HFD feeding. Moreover, they can also evaluate whether overexpression of miR802 can reverse the protective effect of β -cell FoxO1 deletion in vivo.
2. A key point of current study is that miR802 plays functional role in the development of obesity-associated β cell dysfunction, and reducing miR-802 expression improves metabolic parameters in vivo. Therefore, the authors should provide data to elucidate that reduced expression of miR802 in β cells could improve whole body homeostasis in certain metabolic stress, such as high fat diet feeding. However, in this study, authors only investigated the effect of miR802 ablation on normal chow diet feeding mice. So the data is not convincingly address that inhibited miR802 expression can exert the protective effect during obesity induced diabetes. Moreover, it remains elusive as of why β cell specific KO of miR802 showing improved metabolic parameters in normal mice. What is the physiological role of miR802 in islets? The authors should explain about this in the discussion.
3. As shown in the result, serum miR802 level was significantly elevated in diabetic mouse models as well as in overweight human subjects. Therefore, it is important to exclude the likelihood that alternation of miR802 in islet is simply the consequence of increased serum miR802 expression. For example, instead of detecting miR802, the authors can detect the pri-miR802 in the islets.
4. More data should be provided to strengthen the regulation of FoxO1 on miR802 expression. ChIP and EMSA should be performed to show the direct binding of FoxO1 to miR802 promoter. Moreover, the authors should also detect whether this binding is further increased in palmitate treated β cells, or in obese mice islets.
5. As shown in the result, increased miR802 expression was detected in the islets of 8-week-db/db mice and 8-week-HFD mice. However, during the early stage, obesity often leads to hyperinsulinemia because the pancreatic β cells are hyperstimulated to release more insulin. Therefore, the authors should also provide some important parameters of the obese mice used here, such as the fast blood glucose and insulin level. Moreover, since the islet β cell failure in obesity is a progressive process, the authors should detect the alternation of miR802 at different stages during the development of obesity induced diabetes.
6. The author detected miR802 expression in different tissues of control and diabetic mice in Fig. 1e. From the result, we can tell that miR802 expression was elevated not only in islets, but also in many other tissues of diabetic mice. Therefore, it is important to provide Q-PCR or Western Blot

result to show the abundance of miR802 in different tissues.

7. As mentioned in the manuscript, RIP-cre mice were used for generation of β cell specific KO and KI mice. It should be noted that the transgene in RIP-cre has been found to be expressed in the hypothalamus. Therefore, the authors should also detect miR802 in the brain, especially in the hypothalamus. The authors have included naive RIP-Cre mice among the controls in experiments, therefore, the data of RIP-cre mice should also be provided in the supplementary data.

8. The authors should detect whether overexpressed FoxO1 is activated after overexpression. Or they can overexpress the constitutively active form of FoxO1 (eg. ADA-FoxO1 mutant).

9. The authors found that miR802 KI could lead to decreased β cell function because of repressed NeuroD expression, therefore, evaluation of pancreatic islet mass of miR802 KI mice should also be provided in the study.

10. As indicated in the result part, KI mice showed an approximately 500-fold increase in total miR-802 in islets. Therefore, it is necessary to detect the miR802 in the serum of KI mice and discuss about the possibility that increased serum miR802 may have effect on the function of other tissues, such as liver. For example, previous study had reported that, obesity induced hepatic miR802 overexpression may cause insulin resistance and impair glucose metabolism in vivo.

11. NeuroD1 has been reported to have an important role on β cell development. Considering that RIP-cre is first produced at embryonic day 13.5, therefore, the authors should also evaluate the effect of miR802 on embryonic β cell development.

Minor concerns,

1. In the result part (Line 338-339), the authors state that: miR-802 expression in islets is increased in dietary and genetic mouse models of obesity as well as in overweight human subjects. This statement is incorrect. They did not detect miR802 expression in islets of overweight human subjects.

2. In Fig.3b and 4b, the images appear to be overexposed, especially the staining of DAPI. The authors should replace them by images with higher quality.

3. In the result part, the authors should use islets with better function to evaluate the effect of miR802 on glucose stimulate insulin secretion, since the glucose stimulation index (eg. Fig.3e) of the primary islets used in the experiment is relative low.

4. The discussion part should be reconstructed to avoid just repeating the results.

5. In Fig.4l, a space should be used to separate the unit from the number, "200 μ M".

Reviewer #2:

Remarks to the Author:

This is an interesting study where the Fangfan Zhang et al hypothesized that the abnormal β -cells function previously reported in obesity could be a consequence of an alteration in cellular micro-RNAs (miRNAs) level. During their study, they found that the miR-802 was upregulated in genetically prone and dietary-induced mouse model of obesity. They show that inducible transgenic overexpression of miR-802 in mice impairs insulin transcription and secretion while miR-802 knockdown in islets improves β -cell function.

The experiments performed with miRNome certainly show that there is some changes in miRNAs, which might have important functional impact on β -cell function. They also showed the existence of an inverse correlation of miR-802 with its upstream targets i.e. NeuroD1 and Fzd5 in the pancreatic islets of db/db and high fat diet mouse.

Some minor concern is that the authors used pancreatic islets in their assay and islets are consisting of at least four different endocrine cells also including endothelia, tissue macrophages. Nevertheless, they also used MIN6 cells, which are poor insulin producing mouse β -cell line for comparisons. The topic of the manuscript is interesting and scientifically, this is a well designed study. The investigated parameters (morphological, biochemical and molecular biology data) are relevant for the study and the generated data support the drawn conclusions.

I have only the following comments for the clarification that might improve the manuscript:

- 1) The data shown in Supplementary figure S1 (a and b) missing Y-labeled notification for what has been measured although this is a comparisons between two variables. Is it log₁₀ (pvalues)?
- 2) The age and number of specific mouse-group when the islets were isolated is not stated in the figures or legends. Taking into account, the declining β -cells in db/db mice and whether the islets are isolated from young aged mice (7-8) or the db/db mice were housed parallel to the HFD and the islets were isolated 8 weeks later. Is there any difference in miRNAs of pancreatic islets in control of young aged (7-8 weeks) islets compared to 8 weeks older control islets (14 weeks of age included 8 weeks treatment)?
- 3) The use of palmitate (0.5 mM) which by itself is very high, while having albumin (10%) or FBS (10%) during incubation or culture also affect the free concentration of palmitate (Olofsson et al Diabetes, 53 (11), pp. 2836-2843; 2004). Based on the literature the authors could easily predict and mention the possible free concentration of palmitate in their solutions.
- 4) For the qPCR and transfection procedure, I couldn't find any normalization calculation! Did the authors normalize the miR-802 CT values to one or more housekeeping gene?
- 5) Concerning the transfection of isolated islets which is a difficult procedure, do the authors know if the transfections of miR-802 also reached the inner β -cells within the islets or only the outer part of β -cells in the islet mantel was transfected?
- 6) The confocal image of islet is not a good quality image since not only insulin but also the DAPI intensity in miR-802 is also affected and reduced. If possible change the figure and keep the confocal setting the same for the both categories (3a). If possible, put a separated set of the image in supplementary to be easier for reader!
- 7) Why insulin content of islets (Figure 3c) is not related to the protein level of islet and shown as per islet which is a weak way of expressing results (compare with Figure 3d)?
- 8) Regarding the measurement of GSIS, why the authors not relating secreted insulin to the islet insulin content or islet protein in figure 3e. The insulin secretory response of islets is very poor and the basal secretion is very high compared to previously published data (cf Zhang et al, Cell Metabolism 2019)! Do the authors have any explanation for these? Could be due to direct incubation of islets after culture period? Why authors did not preincubated the islets for a short period (like 30-45 min) in KRBH buffer? For overnight culture, the RPMI1640 is more suitable! This might cause the islets to leak insulin and therefore basal insulin release is high? Any explanation to this?
- 9) Surprisingly neither islets nor Min6 cells are responding to high concentration of KCl? Did the addition of KCl (35 mM) compensated by the removal of Na?
- 10) In the figure 4a the confocal image is not of good quality where the insulin staining should at least be as the control in Figure 3k.
- 11) Why the response of islets to high glucose is almost equal to basal in control group (anti-NC)? I roughly estimated the % of insulin secretion (insulin secretion in Figure 4e / insulin content in 4c) in control group and in miR-802 KO islets and there was hardly any differences in the values. Any comments from authors? Is there any significant difference between 2.5 and 33.3 in anti-miR-802?
- 12) Line 491 "od" should be "of".

13) Description of the efficiency Results for the Figure S7a (line 506) should according to the appearance in the figure i.e in the figure S7 overexpression is the first and knockdown later shown.

14) db/db is a communally used animal model of diabetes. Authors do not say anything about the background of choosing this model and neither discuss their findings in db/db mouse islets in relation to previously reported findings. They should also add some sentence about their findings in db/db mouse islets, which can be added to earlier, and very recently findings of mitochondrial dysfunction of this diabetic animal model shown by Zhang et al Cell Metabolism 2019.

Reviewer #3:

Remarks to the Author:

In this study, the authors showed an involvement of miR-802 expression in islets in insulin resistance and type II diabetes using cells and mouse models in which miR-802 is overexpressed or silenced/disrupted. The conclusions are clear and seem to be supported by the results. However, there are parts that are difficult to evaluate/interpret the results because of poor and insufficient description of materials and tools they used.

The authors do not provide the detailed information about the mice used in the study. For example, I could not find the gRNA sequences used to generate miR-802 knockout and knock-in mice (lines 87 and 90). In addition, no information about vector sequences for generation of these mice are provided. What is the length of homology arms used? What kind of stopper sequence and polyA are used? It is desirable to provide the complete DNA sequences of these vectors as supplementary materials (and/or deposit the mouse lines to public resource center). In addition, DNA sequences of all the plasmids including reporter expression constructs can be supplied (or submit the plasmids used in their research to a public repository such as Addgene).

How did the authors obtain RIP-Cre mice? If the authors obtained RIP-Cre strain from the public resource center, official number and name of this strain should be provided.

Without these information, it is difficult for readers/researchers to interpret the results properly and to independently replicate the experiments described by the authors.

Line 82: The approved number should be provided.

Line 92: "donor of two model mice" were co-injected at the same time?

Lines 96-97: Why were PCR genotyping performed two times? What was the knock-in efficiency (how many positive pups were obtained from how many eggs injected)?

Lines 99-101: When heterozygous knockout mice are crossed with Cre mice (homo or hetero?), the offspring are not homozygous knockout mice.

Line 384: The authors use "Homozygous" here but they mention "heterozygous" in the method section.

Line 388: How did the authors get "homozygous"-Cre KI animals (Rip-Cre miR-802ki/ki)?

Lines 388 and 422: Which crosses were used to evaluate mendelian frequencies? Actual number of animals should be provided for each genotype.

Lines 418-419: With this cross, we can obtain only heterozygotes regarding flox allele.

Lines 425 and 930: It is better to provide genotype and sex of littermate controls.

Lines 489-490: What is insulin promoter derived from? Ins1 or 2 of mice? How much length of promoter is used?

Lines 492-493, 506: It is better to give a name to each plasmid used and make a list (as Table).

Line 512: "miR-802"  "anti-miR-802" and delete "Figure 7d"

Line 519: "knockout"  "knockdown"

Line 919: What is Cas9/RNA system?

Line 922: Edit to "AflII and BglI"

Lines 927-928: Why wild-type allele gives rise to two bands?

Line 937: "transfection efficiency"  "knockdown efficiency"

Figure 8: This cartoon looks like the cell with nucleus. However, only Sox6 and CREB but not insulin gene and Foxo1 are within the circle (nucleus-like one). In addition, the Fzd5 is not on the external rectangle (cell membrane-like one). I think it is better to edit the Figure.

Figure S2: "c" and "d" must be "d" and "c" according to the legend and text (lines 349 and 906).

Figure S3e: Homology arm regions should be included in the targeting vector. It is better to add the position of primer binding sites in this Figure (name of primers used should be on the Figure legend). What are "H11-P5-I" and "H11-P3-I"? What is the "] " just downstream of CAG promoter in the vector?

Figure S3f: I cannot see the clear band in WT for BglI.

Figure S3g: What are the "BAT" and "WAT"? It is better to explain those in the Figure legend. The allele names should be unified between Figures S3e and S3g. Size of markers should be added.

Figure S4e: Primer positions used in Figures S4f and S4g should be added. It is better to add exon-intron structure and promoter region.

Figure S4g: Allele name should be unified.

Figure S5b: It is hard to understand this Figure (I cannot see the same sequence between red one and binding region to two genes).

Table S1: "PIP-cre"  "RIP-cre"

Table S7: The title is "Primer sequences used for RT-PCR", but sgRNA sequences and siRNA/shRNA sequences are included.

Dear reviewers:

Thank you very much for your comments and advice to our manuscript entitled **“Obesity-induced overexpression of *miR-802* impairs insulin transcription and secretion”**. We completely accept the reviewers’ recommendation and fully agree to revise our manuscript according to your comments. The manuscript has been subjected to revision carefully and accordingly. We present the comments of each reviewer below. The comments are shown in *italics*, our responses are shown in **blue font**. A thorough, point-by-point response to each point was raised and all changes, a word file of the revised manuscript with all changes in **red font** has been uploaded. If you have any further questions about the revision, please do not hesitate to contact me.

Best regards,

Liang jin

Comments:

Reviewer #1 (Remarks to the Author):

To authors,

The manuscript reported an investigation showing that alternation of islet miR-802 partially responsible for obesity induced β cell dysfunction. The manuscript presented a large amount of data from both in vivo and in vitro studies to elucidate the function of miR-802 in pancreatic β cells. The findings are interesting, but more experiments should be performed to strengthen their conclusions.

Response: Thanks for your comments. We followed your suggestion and provided more convinced evidences to strengthen our conclusions. For example, **we demonstrated that obesity induced miR-802 upregulation depended on Foxo1 activation in vivo, and investigated that overexpression of miR-802 could reverse the protective effect of β -cell Foxo1 deletion in vivo. In addition, ChIP and EMSA assay were performed to identify that Foxo1 could directly bind to miR-802 promoter. Moreover, we have provided data to elucidate that miR-802 KO mice could improve insulin resistance and glucose tolerance fed with high fat diet.** We hope the explanations and changes above would make you and other readers much easier to understand our manuscript.

Major concerns,

1. A series of FoxO1 studies had suggested the important role of FoxO1 in pancreas. However, the functions of FoxO1 in pancreas is still controversial, and FoxO1 was suggested as “a double-edged sword” in the pancreas. In my opinion, a major innovation of this study is that they discovered the miRNAs regulated by FoxO1 during obesity-induced β cell dysfunction and had broadened our current knowledge about FoxO1 function in β cells. Therefore, it is important to verify whether obesity induced miR802 upregulation is depend on FoxO1 activation in vivo. The authors should detect miR802 alternation in inducible β -cell specific FoxO1 KO mice with HFD feeding. Moreover, they can also evaluate whether overexpression of miR802 can reverse the protective effect of β -cell FoxO1 deletion in vivo.

Our response to suggestion 1

Thank you for your good advice. Since we don't have Foxo1^{fl/fl} mice and it was hard to obtain inducible β -cell specific Foxo1 KO mice within 3 months. Based on previous study^{1,2}, tail-vein injection could carry gene or drug to pancreas. Thus, we constructed the lentivirus-shFoxo1, which was injected intravenously into 7-8 weeks old male C57BL/6J mice through the tail vein. Seventy-two hours after the injection, the knockdown efficiency of Foxo1 was measured in islets by qRT-PCR (100 islets for each batch) and western blot (200 islets for each batch). The expression level of miR-802 was also tested. The results showed that the expression level of miR-802 in the islet was dramatically suppressed by lentivirus-shFoxo1 *in vivo* (n=5).

Then mice were fed with high fat diet (HFD) for 8 weeks (D12494, 60% energy from fat). And weighted between 40 g and 45 g. The result showed that the expression level of miR-802 in the islet was almost not increased in Foxo1 knockdown mice compared to control group (n=5).

Furthermore, to evaluate whether overexpression of miR-802 could reverse the protective effect of β -cell Foxo1 deletion *in vivo*. Lentivirus-miR-802 was injected intravenously into Foxo1 knockdown mice through the tail vein, which were fed with HFD for 8 weeks. Seventy-two hours after the injection, the overexpression efficiency of *miR-802* was measured in islets (100 islets for

each batch) by qRT-PCR. The result showed that the expression level of miR-802 was up-regulated 250-fold (n=3). Then we performed IPGTT to test the glucose tolerance, and IPITT was carried out to measure the insulin sensitive (n=7). In addition, insulin secretion and synthesis were also detected in the islets derived from lentivirus-miR-802/lentivirus-shFoxo1 and lentivirus-shFoxo1 mice (n=3-5). As expected, overexpression of miR-802 could reverse the protective effect of β -cell Foxo1 deletion *in vivo*. Glucose tolerance and insulin sensitive were impaired when challenged with IPGTT or IPITT. Moreover, insulin secretion and synthesis were decreased compared to lentivirus-shFoxo1 mice.

Foxo1 mainly expression in adipocyte, liver and islet. Though lentivirus-shFoxo1 could not specifically knock out Foxo1 in β -cell, we did verify that obesity induced miR-802 up-regulation was dependent on Foxo1 activation *in vivo*. We hope the explanations and changes above would strengthen our conclusions. If reviewer thinks that the experiment must perform in inducible β -cell specific Foxo1 KO mice, we can repeat our experiments. Since we have no Foxo1^{fl/fl} mice now, and it will take at least 8 months if we perform this experiment in β -cell specific Foxo1 KO mice. We should submit our manuscript in 3 months, thus we injected lentivirus-shFoxo1 via tail-vein referring to previous study.

To facilitate your check, the main correction in revision was listed below.

.....Next, to verify whether knockdown of *Foxo1* also repressed *miR-802* expression level *in vivo*, 1×10^9 lentivirus particles encoding *Foxo1-shRNA* was injected through the tail vein. We observed an 80% reduction of Foxo1 expression in islets that has received the *lentivirus-shFoxo1* compared to those receiving lentivirus-LV3 (pGLV-H1-GFP+Puro, Figure S2j and Figure S2k). Although *lentivirus-shFoxo1* treatment also efficiently reduced Foxo1 expression in the liver, it only slightly reduced Foxo1 expression in WAT and kidney, and did not affect Foxo1 expression in the BAT, skeletal muscle, brain, spleen and heart (Figure S2j). As showed in Figure 2k, after *lentivirus-shFoxo1* treatment *in vivo*, the expression level of *miR-802* was dramatically decreased in the islet. To further examine *miR-802* alternation in *lentivirus-shFoxo1* treatment mice, *Lentivirus-shFoxo1* treatment mice were fed with HFD for 8-week weighting 40-45 g. The expression level of *miR-802* in the islets has no significant up-regulation compared with control (Figure 2l).....

Figure 2

Figure 2 (k) qRT-PCR was performed to measure the *miR-802* expression levels in *lentivirus-shFoxo1*-treated mice compared with control (white adipose tissue (WAT), brown adipose tissue (BAT), n=3). (l) *miR-802* expression after intravenous injection of HFD-fed mice

with *lentivirus-shFoxo1* (n=3). All experiments above were performed in triplicates, and each group contained three batches of individual samples. All data are represented as mean \pm SD, * p < 0.05, ** p < 0.01, *** p < 0.001.

Figure S2

Figure S2 (j-k) The Knockdown efficiency of Foxo1 were analyzed by qRT-PCR (j) and western blot (k, n=3). All experiments above were performed in triplicates, where each group consisted of three samples. All the results above were represented as mean \pm SD; * p < 0.05, ** p < 0.01, *** p < 0.001 compared with the control.

.....Based on previous study, *Foxo1* could repress the expression level of *miR-802*. To evaluate whether overexpression of *miR-802* could reverse the protective effect of β -cell *Foxo1* deletion *in vivo*. We injected *lentivirus-miR-802* (*len-miR-802*) in *lentivirus-shFoxo1* (*len-shFoxo1*) treated mice by an HFD treatment for 8 weeks. At 72 h after injection, islets (100 islets) were lysed to extract total RNA to measure the overexpression efficacy. The result showed that the expression level of *miR-802* in the islets was increased approximately 250-fold compared to control mice (Figure S3s, n=3). We observed that lower fasting serum insulin (FINS) levels and homeostasis model assessment of the insulin resistance index (HOMA-IR) were reversed by *lentivirus-miR-802* compared to merely *lentivirus-shFoxo1* treated mice (Figure 3l and m, n=7). In addition, compared with the *lentivirus-shFoxo1* group, the area under the curve obtained from intraperitoneal glucose tolerance test and intraperitoneal insulin tolerance test assays was increased in *lentivirus-miR-802*-treated mice, and these effects were markedly ameliorated by *lentivirus-shFoxo1* treatment versus control group on HFD treatment (Figure 3n and o, n=7). Moreover, the ability of insulin transcription (Figure 3p, n=3-5) and secretion (Figure 3q, n=3-5) were suppressed in mice receiving *lentivirus-shFoxo1* and *lentivirus-miR-802* treatment compared to merely *lentivirus-shFoxo1*-treated mice. In summary, these results revealed that overexpression of *miR-802* could reverse the protective effect of β -cell Foxo1 deletion *in vivo*.....

Figure 3

Figure 3 Fasting insulin levels (FINS, l) and HOMA-IR (m) at indicated time points of *lentivirus-shFoxo1*-treated, *lentivirus-shFoxo1* and *lentivirus-miR-802*-treated mice fed an HFD diet compared with control mice (n=7). HOMA-IR was calculated as $HOMA-IR = (FBG \text{ (mmol/l)} \times FINS \text{ (mIU/l)})/22.5$. (n, o) The intraperitoneal glucose tolerance test (GTT) (n) and intraperitoneal insulin tolerance test (ITT) (o) assays were performed to evaluate the insulin sensitivity of mice in the indicated groups after *lentivirus-shFoxo1* or *lentivirs-miR-802* treatment. Area under the curve (AUC) was calculated (n= 7), insulin synthesis (p) and insulin secretion (q) in HFD islets with *lentivirus-shFoxo1* or *lentivirus-miR-802* treatment (n=3-5). All data are represented as mean \pm SEM except p-q (mean \pm SD). * $p < 0.05$; ** $p < 0.01$; *** $p < 0.001$ versus control group; # $P < 0.05$, ## $P < 0.01$, ### $P < 0.001$ versus *lentivirus-shFoxo1* group.

Figure S3 s

Figure S3 (s) Overexpression efficiency of *miR-802* after intravenous injection of HFD-fed mice with *lentivirus-miR-802* (n=3). Each group was analyzed in triplicates. All the results above were represented as mean \pm SD, *** $p < 0.001$.

2. A key point of current study is that *miR802* plays functional role in the development of obesity-associated β cell dysfunction, and reducing *miR-802* expression improves metabolic parameters in vivo. Therefore, the authors should provide data to elucidate that reduced expression of *miR802* in β cells could improve whole body homeostasis in certain metabolic stress, such as high fat diet feeding. However, in this study, authors only investigated the effect of *miR802* ablation on normal chow diet feeding mice. So the data is not convincingly address that inhibited *miR802* expression can exert the protective effect during obesity induced diabetes. Moreover, it remains elusive as of why β cell specific KO of *miR802* showing improved metabolic parameters in normal mice. What is the physiological role of *miR802* in islets? The authors should explain about this in the discussion.

Our response to suggestion 2

Thank you for your suggestions. In the previous study, *miR-802* KO mice were fed with high fed diet for 12 week to elucidate that reduced expression of *miR-802* in β cells could improve whole body homeostasis in high fat diet feeding. In the revised draft, we have added these data (see below). Moreover we explained why β cell specific KO of *miR-802* showing improved metabolic parameters in normal mice in the discussion. We hope the explanations and changes above would strengthen our conclusions that inhibited *miR-802* expression could exert the protective effect during obesity induced diabetes.

To facilitate your check, the main correction in revision was listed below.

Methods

Mouse metabolic assays

Mouse metabolic assays were performed as previously described³. Briefly, fasting blood glucose (FBG) levels and fasting serum insulin (FINS) levels were examined after a 6 h fasting treatment by using a glucometer (OMRON, Japan) and by ELISA (ExCell Bio, Shanghai, China), respectively. And the homeostatic model assessment indices of insulin resistance (HOMA-IR) was calculated with the equation (FBG (mmol⁻¹) x FINS (mIU⁻¹))/22.5. To perform the glucose tolerance tests, 1.5 g/kg glucose (Sigma-Aldrich, St Louis, MO, USA) was i.p. injected into mice,

whereas 0.75 U/kg insulin (Novolin R, Novo Nordisk, Bagsvaerd, Denmark) was i.p. injected into mice for insulin tolerance tests. Blood glucose levels were examined at 0, 15, 30, 60 and 120 min after injection. The AUC are given as the incremental area under the curve, calculated by the conventional trapezoid rule.

Results

Genetic deletion of *miR-802* leads to improve obesity-associated insulin resistance and glucose intolerance

Since our data demonstrated that expression of *miR-802* was upregulated in the islets of obese mice, we hypothesized that the improved metabolic control of *miR-802 KO* mice would be enhanced under HFD feeding. Then, we exposed 8-week old male *miR-802 KO* mice to HFD for 12 weeks. We confirmed that mice fed HFD for 12 weeks displayed no effect on body weight, random-fed glycemia, cumulative energy intake or body fat content as well as no effect on adipocyte size between control (*miR-802^{fl/fl}*) mice and *miR-802 KO* mice (Figure S5a-e, n=5-10). On the other hand, the serum insulin concentrations was diminished in *miR-802 KO* mice of HFD-feeding (Figure 5a, n=10). In accordance with this, glucose tolerance tests revealed an improvement of glucose tolerance upon *miR-802 KO* mice (Figure 5b n=10). Moreover, insulin sensitivity was also improved compared to control mice (Figure 5c) and *miR-802 KO* mice lead to a reduction of homeostatic model assessment indices of insulin resistance (HOMA-IR) indices (Figure 5d). Subsequently, we isolated islets of control mice and *miR-802 KO* mice after 6- and 12-week HFD treatment. GSIS results revealed that insulin release was markedly improved in *miR-802 KO* mice when islets exposed to 33.3 mM glucose or 35 mM KCl (Figure 5e-f, n=3-5). This finding further indicated that deletion of *miR-802* expression in pancreatic islets contributes to the compensatory β cell secretory function instigated by insulin resistance and obesity.

Figure 5

Figure 5 *miR-802 KO* mice improves obesity-associated insulin resistance and glucose intolerance when fed with high fat diet (HFD)

8-week-old male *miR-802 KO* mice and control mice (*miR-802^{fl/fl}*) were exposed to HFD for 12 weeks. (a) Then, Fasting insulin levels (FINS) of HFD-fed mice were measured by ELISA after 12-week of feeding HFD (n=10). (b-c) Intraperitoneal glucose tolerance test (GTT; 1.5g/kg) (b) and intraperitoneal insulin tolerance test (ITT; 0.75 units per kg) (c) were performed in *miR-802 KO* mice and control mice at the 11th or 12th week of High fat diet administered, respectively. The corresponding area under the curve (AUC) of blood glucose level was calculated (n=10). (d) Homeostatic model assessment indices of insulin resistance (HOMA-IR) of *miR-802 KO* mice and control mice (n=10), HOMA-IR was calculated as $HOMA-IR = (FBG (mmol/l) \times FINS (mIU/l))/22.5$. (e-f) Insulin release from islets of *miR-802 KO* mice or control mice after 6-week (e)

or 12-week (f) HFD treatment (n=3-5). In all panels error bars indicate mean \pm SEM, except e-f (mean \pm SD); * p < 0.05, ** p < 0.01, *** p < 0.001.

Figure S5

Figure S5 (a) Changes in the body weight of *miR-802 KO* and control mice treated with a HFD for 12 weeks (n=10 per time point), Fasting blood glucose levels (FBG) (b, n=10), cumulative energy intake (c, n=10), white adipose tissue weight per body weight ratio (d, n=10) and representative H&E staining of white adipose tissue in *miR-802 KO* mice and control mice treated with HFD (e, n=5). FBG levels and cumulative energy intake were measured every 1 week. Data shown are mean \pm SEM.

Discussion

Here, we found that *Fzd5* was a novel *miR-802* target, which could mediate at least part of the effects of *miR-802* on insulin secretion via increasing intracellular Ca^{2+} content. Thus, increased expression of Ca^{2+} influx regulators contributes to improving insulin secretion in *miR-802 KO* mice. In this context, *miR-802* represents a unique miRNA molecule for its ability to suppress Ca^{2+} influx reducing the release competence of secretory granules, thereby impairing secretory robustness to pancreatic β cells.

3. As shown in the result, serum *miR802* level was significantly elevated in diabetic mouse models as well as in overweight human subjects. Therefore, it is important to exclude the likelihood that alternation of *miR802* in islet is simply the consequence of increased serum *miR802* expression. For example, instead of detecting *miR802*, the authors can detect the *pri-miR802* in the islets.

Our response to suggestion 3

Thank you for your meaningful comments. In order to exclude the likelihood that alternation of *miR-802* in islet is simply the consequence of increased serum *miR-802* expression. We detected the *Pri-miR-802* in the islets of HFD mice. The result showed that the expression level of *Pri-miR-802* was also enhanced in the islets of HFD mice compared to NCD mice. All of the

above data confirmed that overweight could induce miR-802 expression up-regulation in the islets.

To facilitate your check, the main correction in revision was listed below.

.....To investigate whether alternation of *miR-802* in islet is simply the consequence of increased serum *miR-802* expression. We detected the *Pri-miR-802* in the islets of HFD mice. As shown in Fig.1i, the expression level of *Pri-miR-802* displayed a similar trend with *miR-802*.....

Figure 1i

Figure 1 (i) the expression level of *Pri-miR-802* in the islets of HFD and NCD mice were analyzed by qRT-PCR (n=3-5). All experiments above were performed in triplicates, and each group contained three batches of individual samples. All data are represented as mean \pm SD, *** $p < 0.001$.

4. More data should be provided to strengthen the regulation of *FoxO1* on *miR802* expression. ChIP and EMSA should be performed to show the direct binding of *FoxO1* to *miR802* promoter. Moreover, the authors should also detect whether this binding is further increased in palmitate treated β cells, or in obese mice islets.

Our response to suggestion 4

Thank you for your suggestion. In the revised draft, ChIP and EMSA analysis were performed to show the direct binding of *Foxo1* to *miR-802* promoter. The result revealed that *Foxo1* could directly bind to *miR-802* promoter in Min6 cells and islets via EMSA analysis. And ChIP-qPCR results showed that Min6 cells transfected with ADA-*Foxo1* exhibited significantly the higher binding ability of *Foxo1* to *miR-802* promoter compared to control, while the binding ability exhibited significantly lower by si-*Foxo1*. Moreover, we also detected this binding was further increased in obese mice islets, and in 0.5 mM palmitate treated Min6 cells.

To facilitate your check, the main correction in revision was listed below.

..... To explore the interaction between promoter region of *miR-802* and *Foxo1*, Min6 cells transfected with ADA-*Foxo1* exhibited significantly the higher binding ability of *Foxo1* to *miR-802* promoter compared to control via dual luciferase (Figure S2f) and ChIP (Figure 2e, Figure S2g) assay, while the binding ability exhibited significantly lower in case of si-*Foxo1*. And we also detected this binding was further increased in 0.5 mM palmitate treated Min6 cells (Figure 2f and Figure S2h), as well as in obese mice islets (Figure 2g and Figure S2i). We then performed an EMSA assays to detect whether *Foxo1* could directly bind to the *miR-802* promoter in Min6 cells. As shown in Figure 2h signal from the probe-protein-anti-*Foxo1* complex was detected using a *miR-802* probe. However, when the core sequence in the *miR-802* probe was mutated, the

probe-protein-anti-Foxo1 complex was completely lost. The same result was obtained in the islets (Figure 2i).....

Figure 2

Figure 2 (e-g) The enrichment of Foxo1 on the *miR-802* promoter relative to IgG detected by ChIP-qPCR assays, in MIN6 cells transfected with *ADA-Foxo1*, pcDNA 3.1 vector, *si-Foxo1* or si NC (e), in MIN6 cells treatment with 0.5 mM palmitate or without palmitate (f), and in obese mice islets or normal mice islets (g, n=3-5). Foxo1 could directly bind to *miR-802* promoter in Min6 cells (h) and islets (i, n=3-5) through EMSA assays. C1 and C2 represented nuclear protein-*miR-802* probe complexes, nuclear protein-*miR-802* probe-anti-Foxo1 complexes, respectively. Biotin-WT was a 25 bp fragment probe which included the binding region of Foxo1, while Biotin-MUT was a 25 bp fragment probe and the binding sequence was mutated. All experiments above were performed in triplicates, where each group consisted of three samples. All the results above were represented as mean \pm SD; ** $p < 0.01$, *** $p < 0.001$ compared with the control.

Figure S2

Figure S2 ChIP experiment showed that Foxo1 binds to *miR-802* promoter via RT-PCR analysis, in the Min6 cells (g), in 0.5 mM palmitate-treated Min6 cells (h), and in obese mice islets (i,

n=3-5). All experiments above were performed in triplicates, where each group consisted of three samples. All the results above were represented as mean \pm SD; ** $p < 0.01$, *** $p < 0.001$ compared with the control.

5. As shown in the result, increased miR802 expression was detected in the islets of 8-week-db/db mice and 8-week-HFD mice. However, during the early stage, obesity often leads to hyperinsulinemia because the pancreatic β cells are hyperstimulated to release more insulin. Therefore, the authors should also provide some important parameters of the obese mice used here, such as the fast blood glucose and insulin level. Moreover, since the islet β cell failure in obesity is a progressive process, the authors should detect the alternation of miR802 at different stages during the development of obesity induced diabetes.

Our response to suggestion 5

Thank you for your great suggestions. We have provided blood glucose, body weight and insulin levels of 8-week-db/db mice and 8-week-HFD mice we used.

Moreover, we have detected the alternation of miR-802 at different stages during the development of obesity induced diabetes, and found that the expression level of miR-802 was up-regulated along with body weight increased.

To facilitate your check, the main correction in revision was listed below.

.... The body weight, blood glucose and insulin levels of these mice were listed in supplementary Figure 1a-f....

Figure S1

Figure S1 The body weight, glucose and insulin level of HFD mice compared with NCD mice (a-c, n=7-8, 14-15-week old) and Lepr^{db/db} mice compared with control mice (d-f, n=7-8, 8-week old). All the results above were represented as mean \pm SEM; *** $p < 0.001$ compared with the control.

.... And we detected the alternation of *miR-802* at different stages during the development of obesity induced diabetes. As shown in Figure 1e, the expression level of *miR-802* in the islet was up-regulated along with body weight gain....

Figure 1e

Figure 1(e) qRT-PCR was performed to measure the expression level of *miR-802* in the islets at different stages during the development of obesity induced diabetes (n=3-5). All experiments above were performed in triplicates, where each group consisted of three samples. All the results above were represented as mean \pm SD; * $p < 0.05$, ** $p < 0.01$, *** $p < 0.001$ compared with the mice weighted 25-30 (g).

6. The author detected *miR802* expression in different tissues of control and diabetic mice in Fig. 1e. From the result, we can tell that *miR802* expression was elevated not only in islets, but also in many other tissues of diabetic mice. Therefore, it is important to provide Q-PCR or Western Blot result to show the abundance of *miR802* in different tissues.

Our response to suggestion 6

Thank you for your advice. The result of Figure 1e (revised manuscript was Figure 1f) was observed via qRT-PCR analysis. The result showed that *miR-802* expression was elevated not only in islet but in liver, kidney, white adipose tissue (WAT), brown adipose tissue (BAT), and skeletal muscle of obese mice. According to your suggestion, we have performed qRT-PCR to test the abundance of *miR-802* in different tissues of wide type mice (n=5). Our analysis revealed that murine *miR-802* expression was highly enriched in the liver and islet. *miR-802* was also abundant in kidney and white adipose tissue (WAT) as well as in brown adipose tissue (BAT), while *miR-802* almost could not detect in other tissues.

To facilitate your check, the main correction in revision was listed below.

.....As expected, *miR-802* expression was highly enriched in the liver and islet, and also abundant in kidney, heart and white adipose tissue (WAT), while *miR-802* almost could not detect in other tissues of wide type mice (n=5, Figure S1i).....

Figure S1i

Figure S1 (i) the abundance of *miR-802* in different tissues of wide type mice by qRT-PCR analysis (n=5). All the results above were represented as mean \pm SD; *** $p < 0.001$ compared with islet.

7. As mentioned in the manuscript, RIP-cre mice were used for generation of β cell specific KO and KI mice. It should be noted that the transgene in RIP-cre has been found to be expressed in the hypothalamus. Therefore, the authors should also detect *miR802* in the brain, especially in the hypothalamus. The authors have included naive RIP-Cre mice among the controls in experiments, therefore, the data of RIP-cre mice should also be provided in the supplementary data.

Our response to suggestion 7

Thank you for your advice and we agree with the reviewer's point. RIP-Cre was mainly expressed in islet, also slightly detected in hypothalamus and hippocampus as previously described⁴. Related to our previous result that *miR-802* almost could not be detected in the brain. In this study, we found *miR-802* was slightly reduced in hippocampus of *miR-802* KO mice and increased in hypothalamus and hippocampus of *miR-802* KI mice. These results have been provided in the supplementary data.

To facilitate your check, the main correction in revision was listed below.

.....Rip-Cre has been previously implicated in hypothalamus and hippocampus⁴, we compared *miR-802* expression in primary islet versus hypothalamus and hippocampus from Rip-Cre mice, revealing that Rip-Cre expression was 1400-fold higher in islet versus hypothalamus, and 400-fold higher in islet versus hippocampus (Figure S3m, n=5). Indeed, a 2-fold upregulation of *miR-802* expression in the hypothalamus and a 7-fold increase of *miR-802* expression in the hippocampus of *miR-802* KI mice (Figure S3n, n=5).....

Figure S3

Figure S3 (m) Rip-Cre expression in islet, hypothalamus and hippocampus from Rip-Cre mice (n=5). *** $P < 0.001$ vs islet. (n) Relative *miR-802* expression level in hypothalamus and hippocampus of *miR-802 KI* mice (n=5).

.....Moreover, it only slightly reduced *miR-802* expression in hippocampus and did not affect *miR-802* expression in the hypothalamus (Figure S4I)....

Figure S4I

Figure S4 (I) Relative *miR-802* expression level in hypothalamus and hippocampus of *miR-802 KO* mice (n=5).

8. The authors should detect whether overexpressed *FoxO1* is activated after overexpression. Or they can overexpress the constitutively active form of *FoxO1* (eg. ADA-*FoxO1* mutant).

Our response to suggestion 8

Thank you for your meaningful comments. In this study, we have detected the phosphorylation level of *Foxo1* and verified that over-*Foxo1* phosphorylated *Foxo1*. According to your suggestion, we have overexpressed the constitutively active form of ADA-*Foxo1* mutant (T24A-S253D, S316A mutations)⁵ for ChIP experiment, and we repeated the dual luciferase experiments.

Min6 cells were transfected with over-Foxo1 vector or si-Foxo1 for 48 h, then western blot was performed to detect the phosphorylation level of Foxo1.

Figure S2f

Figure S2 (f) A *Foxo1* binding site were identified in the -3 kb upstream region of the *miR-802* primary transcript. Mutagenesis in the putative binding site abrogated the induction activity of *Foxo1* in the Min6 cells. All experiments above were performed in triplicates, where each group consisted of three samples. All the results above were represented as mean \pm SD; *** $p < 0.001$ compared with the control.

9. The authors found that *miR802* KI could lead to decreased β cell function because of repressed *NeuroD* expression, therefore, evaluation of pancreatic islet mass of *miR802* KI mice should also be provided in the study.

Our response to suggestion 9

Thank you for your kind advice. *NeuroD1* was first identified to control β cell development and differentiation but is now essential for insulin gene transcription. In this study, we mainly focused on researching the role of *miR-802* regulation insulin transcription via directly targeting *NeuroD1*. According to your advice, we have evaluated pancreatic islet mass of *miR802* KI mice. Morphometric analyses revealed that β cell mass was slightly decreased in *miR-802* KI animals compared with *miR-802*^{ki/ki} mice, although this effect did not reach statistical significance. To facilitate your check, the main correction in revision was listed below.

...morphometric analyses reveal that β cell mass was slightly decreased in *miR-802* KI animals compared with *miR-802*^{ki/ki} mice, although this effect did not reach statistical significance (Figure S3q, n=3)....

Figure S3q

Figure S3 (q) Comparison of β cell mass in control ($miR-802^{ki/ki}$) and $miR-802$ KI mice (n = 3).

10. As indicated in the result part, KI mice showed an approximately 500-fold increase in total $miR-802$ in islets. Therefore, it is necessary to detect the $miR802$ in the serum of KI mice and discuss about the possibility that increased serum $miR802$ may have effect on the function of other tissues, such as liver. For example, previous study had reported that, obesity induced hepatic $miR802$ overexpression may cause insulin resistance and impair glucose metabolism in vivo.

Our response to suggestion 10

Thank you for that good question. We have detected the $miR-802$ expression in the serum of $miR-802$ KI mice. The result revealed that a 12-fold increase of $miR-802$ expression in the serum of $miR-802$ KI mice compared to $miR-802^{ki/ki}$ mice (n=10). And we discussed the possibility that increased serum $miR-802$ may have effect on the function of other tissues in the discussion.

To facilitate your check, the main correction in revision was listed below.

.....Moreover, a 12-fold increase of $miR-802$ expression in the serum of $miR-802$ KI mice compared to $miR-802^{ki/ki}$ mice (Figure S3j, n=10).....

Figure S3l

Figure S3 (l) Relative $miR-802$ expression in the serum of $miR-802$ KI mice and $miR-802^{ki/ki}$ mice

(n=10), using *Ce-miR-39-1* as positive control. *miR-802* expression was set to 1 in SD.

Discussion

.....miRNAs, which are stable in the serum and plasma ⁶, are possibly transferred from donor cells to recipient cells where they alter the gene expression of recipient cells. In the present study, we found that *miR-802* expression in the serum of *miR-802 KI* mice was revealed a 12-fold increase compared to control mice, suggesting that *miR-802* might play potential roles in organization communication, such as liver. For instance, previous study had reported that obesity induced hepatic *miR-802* overexpression, which caused insulin resistance and impairing glucose metabolism *in vivo* ⁷.....

11. *NeuroD1* has been reported to have an important role on β cell development. Considering that *RIP-cre* is first produced at embryonic day 13.5, therefore, the authors should also evaluate the effect of *miR802* on embryonic β cell development.

Our response to suggestion 11

Thanks for your kind advice. *NeuroD1* was first identified to control β -cell development but are now known to also maintain mature β -cell function ^{8,9,10}. Here, we mainly focused on researching the role of *NeuroD1* regulation insulin transcription. According to your suggestions, we evaluated the effect of *miR-802* on embryonic β cell development in the revised manuscript.

First, we detected the *miR-802* expression of E9.5, E10.5, E13.5 and E17.5 in the pancreas, and found *miR-802* was increased with pancreas development, indicating that *miR-802* might play critical roles in pancreas development. We next examined the effect of *miR-802* on the development of β cells in *miR-802 KI* mice and *miR-802 KO* mice at E13.5 and E 17.5 compared them to control mice. The result showed that *miR-802* could slightly regulate β -cell development. These result have added in the supplementary Figure 7c-e. In the further study, we will systematically research the effect of *miR-802* on the development of pancreas following your great suggestion.

To facilitate your check, the main correction in revision was listed below.

qRT-PCR was performed to test the expression levels of *miR-802* in E9.5, E10.5, E13.5 and E17.5 pancreas of C57BL/J mice (n=5-7). qRT-PCR was performed in triplicates, where each group consisted of three groups. The results above were represented as mean \pm SEM, ****p* < 0.001 compared with the E9.5.

....We next examined the effect of *miR-802* on the development of β cells in *miR-802 KO* mice and *miR-802 KI* mice at E13.5 and E 17.5 compared them to control mice. Revealing that knockdown *miR-802* could promote β -cell differentiation and pancreatic β -cell numbers were

increased in *miR-802 KO* mice (Figure S7c, n=3-5), while *miR-802 KI* mice achieved the opposite effect (Figure S7d and e, n=3-5). Taken together, these data suggested that *miR-802* affected insulin transcription and β -cell development in a *NeuroDI*-dependent manner...

Figure S7

Figure S7 (c) Pancreatic sections of control mice and *miR-802 KO* mice were immunohistochemically stained for insulin at E13.5 (upper) and E17.5 (lower). Scale bars: 50 μ m (n = 3-5 mice 6–8 slides/animal). The number refers to cell numbers/pancreas area. Immunofluorescence of insulin (green) and DAPI (blue) in E13.5 (d) and E17.5 (e) pancreas of control mice and *miR-802 KI* mice (n=3-5 mice). Data information: Data are presented as means \pm SEM; statistical significance was assessed by ANOVA and 2-tailed unpaired Student's t-test. *P < 0.05; ***P < 0.001. Scale bars: 50 μ m (c); 20 μ m (d, e).

Minor concerns,

1. In the result part (Line 338-339), the authors state that: *miR-802* expression in islets is increased in dietary and genetic mouse models of obesity as well as in overweight human subjects. This statement is incorrect. They did not detect *miR802* expression in islets of overweight human subjects.

Our response to minor concerns 1

Thank you for your kind advice, and we apologize for the ambiguous formulation. In the revised manuscript, we have deleted “as well as in overweight human subjects”.

2. In Fig.3b and 4b, the images appear to be overexposed, especially the staining of DAPI. The authors should replace them by images with higher quality.

Our response to minor concerns 2

Thank you for your meaningful comments. We have replaced Fig.3b and 4b by images with higher quality.

To facilitate your check, the main correction in revision was listed below.

Figure 3b

Figure 3 *miR-802* mimics was transfected into primary islets and Min6 cells for 48 h. Then, qRT-PCR was used to evaluate the *Ins1* and *Ins2* levels, followed by immunostaining for DAPI (blue) and insulin (red) (Magnification: 20× or 40×, scale bar: 50 μm or 20 μm), (a) in islets and (b) in Min6 cells.

Figure 4b

Figure 4 *anti-miR-802* was transfected into primary islets and Min6 cells for 48 h. Then, qRT-PCR and immunostaining for DAPI (blue) and insulin (red) were performed (Magnification: 20× or 40×, scale bar: 50 μm or 20 μm) in islets (a) and Min6 cells (b),

3. In the result part, the authors should use islets with better function to evaluate the effect of *miR802* on glucose stimulate insulin secretion, since the glucose stimulation index (eg. Fig.3e) of the primary islets used in the experiment is relative low.

Our response to minor concerns 3

Thank you for your good advice. We have modified the GSIS method according to the reference¹¹ and used islets with better function to evaluate the effect of *miR-802* on glucose stimulate insulin secretion.

To facilitate your check, the main correction in revision was listed below.

Methods

Glucose-stimulated Insulin Secretion (GSIS) in Mouse Islets

Mouse pancreatic islets (30 islets/well in 48-well plate) were collected under a stereomicroscope at room temperature and cultured in 2.5 mM glucose in the absence or presence of test agents for 48 h. Thereafter the islets were washed and preincubated for 30 min at 37 °C in KRBH buffer, supplemented with 0.2% bovine serum albumin and 2.5 mM glucose. After preincubation, the buffer was changed to a medium containing either 2.5 mM, 33.3 mM glucose or 35 mM KCl. The islets were then incubated for 1 h at 37 °C. Immediately after incubation an aliquot of the medium was removed for analysis of insulin via mice insulin ELISA kit (ExCell Bio, Shanghai, China), and the islets were lysed to extract total protein, the amount of insulin secretion was normalized by the total cellular protein content¹¹....

Figure 3

Figure 3 *miR-802* mimics was transfected into primary islets for 48 h. insulin secretion was analyzed by GSIS assay in islets (e). (j) Static insulin secretion was evaluated in islets from 15-week-old *miR-802* KI and control mice (n = 5-7) at indicated glucose and KCl concentrations. All data are represented as mean ± SD. ***p* < 0.01; ****p* < 0.001.

Figure 4

Figure 4 *anti-miR-802* was transfected into primary islets for 48 h. Insulin secretion was analyzed by GSIS assay in islets (e). (i and j) Static insulin secretion performed with islets from 5- (i) and 35-week-old (j) *miR-802* KO and control mice (n = 5-7) at indicated glucose and KCl concentrations. All data are represented as mean ± SD, **p* < 0.05, ***p* < 0.01; ****p* < 0.001.

4. *The discussion part should be reconstructed to avoid just repeating the results.*

Our response to minor concerns 4

Thank you for your suggestions. We have reconstructed our discussion and avoid repeating the result in the revised draft. All changes have marked red in the test.

5. *In Fig.4l, a space should be used to separate the unit from the number; “200 μ M”.*

Our response to minor concerns 5

Thank you for your advice. A space was added to separate the unit from the number “200 μ M” in the Fig.4l.

Reviewer #2 (Remarks to the Author):

This is an interesting study where the Fangfang Zhang et al hypothesized that the abnormal β -cells function previously reported in obesity could be a consequence of an alteration in cellular micro-RNAs (miRNAs) level. During their study, they found that the miR-802 was upregulated in genetically prone and dietary-induced mouse model of obesity. They show that inducible transgenic overexpression of miR-802 in mice impairs insulin transcription and secretion while miR-802 knockdown in islets improves β -cell function.

The experiments performed with miRNome certainly show that there is some changes in miRNAs, which might have important functional impact on β -cell function. They also showed the existence of an inverse correlation of miR-802 with its upstream targets i.e. NeuoD1 and Fzd5 in the pancreatic islets of db/db and high fat diet mouse.

Some minor concern is that the authors used pancreatic islets in their assay and islets are consisting of at least four different endocrine cells also including endothelia, tissue macrophages. Nevertheless, they also used MIN6 cells, which are poor insulin producing mouse β -cell line for comparisons. The topic of the manuscript is interesting and scientifically, this is a well designed study. The investigated parameters (morphological, biochemical and molecular biology data) are relevant for the study and the generated data support the drawn conclusions.

I have only the following comments for the clarification that might improve the manuscript:

Response: Thanks for your positive comments. By the way, we have replace Figure 3a and 4a with higher quality, and put a separated set of the image in the revised manuscript. Moreover, we used islets with better function to measure the glucose stimulation index according to your suggestions, and the insulin content and insulin secretion of islet were normalized to the total protein. We hope the explanations and changes above would make you and other readers much easier to understand our manuscript.

1) The data shown in Supplementary figure S1 (a and b) missing Y-labeled notification for what has been measured although this is a comparisons between two variables. Is it log 10 (pvalues)?

Our response to issues 1

Thank you for your good advice, and we apologize for the ambiguous formulation. We added the Y-labeled notification (-log 10 (p value)) in supplementary Figure S1 a and b (revised manuscript was Figure S1 g and h).

To facilitate your check, the main correction in revision was listed below.

Figure S1 The volcano of miRNA in HFD (g, red bar: up, green bar: down and blue bar: no difference) versus NCD mice, and $Lepr^{db/db}$ mice (h) versus $Lepr^{db/-}$ mice.

2) The age and number of specific mouse-group when the islets were isolated is not stated in the figures or legends. Taking into account, the declining β -cells in db/db mice and whether the islets are isolated from young aged mice (7-8) or the db/db mice were housed parallel to the HFD and the islets were isolated 8 weeks later: Is there any difference in miRNAs of pancreatic islets in control of young aged (7-8 weeks) islets compared to 8 weeks older control islets (14 weeks of age included 8 weeks treatment)!

Our response to issues 2

Thank you for your suggestions.

First, the age and number of specific mouse-group when the islets were isolated have been stated in the Figure legends, which were marked red in the revised manuscript.

Second, in our study, the islets of experiment groups and control groups were isolated at the same week. The islets of db/db mice weighting 40-45 g and wild type ($db/-$) control mice weighting 23-25 g were isolated at 8-week old, and the islets of high fat diet (HFD) mice weighting 40-45g and normal diet control (NCD) mice weighting 23-25 g were isolated at 14 weeks old. We have provided blood glucose, body weight and insulin levels of 8-week- db/db mice and 8-week-HFD mice.

Moreover, we have detected the miR-802 expression of pancreatic islets in control of young aged islets weighting 23-25 g (n=5, 7-8 weeks) compared to 8 weeks older control islets weighting 25-28 g (n=5). The result revealed that the expression level of miR-802 has no significantly difference (see below).

To facilitate your check, the main correction in revision was listed below.

.... The body weight, blood glucose and insulin levels of these mice were listed in supplementary Figure 1a-f....

Figure S1

Figure S1 the body weight, glucose and insulin level of HFD mice compared with NCD mice (a-c, n=7-8, 14-15-week old) and Lepr^{db/db} mice compared with control mice (d-f, n=7-8, 8-week old). All the results above were represented as mean \pm SEM; *** p < 0.001 compared with the NCD mice or wild type mice.

qRT-PCR was performed to measure the expression level of *miR-802* at 7-8 weeks old mice weighting 23-25 g and at 14-15 weeks old mice weighting 25-28 g (n=3). qRT-PCR experiment

was performed in triplicate, where each group consisted of three samples. The results above were represented as mean \pm SD; NS: no significance compared to the 7-8 week old mice weighting 23-25 g.

3) The use of palmitate (0.5 mM) which by itself is very high, while having albumin (10%) or FBS (10%) during incubation or culture also affect the free concentration of palmitate (Olofsson et al *Diabetes*, 53 (11), pp. 2836-2843; 2004). Based on the literature the authors could easily predict and mention the possible free concentration of palmitate in their solutions.

Our response to issues 3

Thank you for your good comments.

First, based on earlier studies and concentration effects shown later^{12, 13, 14, 15}, most studies used 0.25 mM or 0.5 mM palmitate. In the previous study, palmitate (0 mM, 0.25 mM, 0.5 mM, 1mM) was used to stimulate islets for 48 h, revealing that miR-802 expression was increased along with increasing concentration of palmitate, while the expression level of miR-802 has no significantly difference in 0.5 mM and 1 mM (see below). Thus, in the further research, we chose 0.5 mM palmitate to stimulate β cells.

Moreover, we apologize for the ambiguous formulation. Palmitate (200 mM) was dissolved in ethanol, then diluted 1:22.2 in 10% BSA (9 mM), while before use, 9 mM palmitate was diluted 1:18 in serum-free DMEM to achieve final palmitate concentration (0.5 mM), and the concentration of BSA was 0.5% (wt/vol), which was the same as previously report^{16, 17}. We have rewritten this part in the revised manuscript. We hope the explanations and changes above would make you and other readers much easier to understand our manuscript. To facilitate your check, the main correction in revision was listed below.

Palmitate (0 mM, 0.25 mM, 0.5 mM, 1mM) was used to stimulate islets for 48 h, then qRT-PCR was performed to detect miR-802 expression, revealing that miR-802 expression was increased along with increasing concentration of palmitate increase, while the expression level of *miR-802* has no significantly difference in between 0.5 mM and 1 mM. qRT-PCR experiment was performed in triplicate and the results above were represented as mean \pm SD; * $p < 0.05$, *** $p < 0.001$ compared with the 0 mM palmitate.

Methods

..... For palmitate treatment, islets and Min6 cells were incubated in 0.5 mM palmitate (Sigma

Aldrich). Palmitate (200 mM) was dissolved in ethanol, filter sterilized, diluted 1:22.2 in 10% BSA (9 mM palmitate). Before use, 9 mM palmitate-BSA stock was diluted 1:18 in serum-free DMEM to achieve 0.5 mM palmitate in the presence of 0.5% (wt/vol.) BSA....

4) For the qPCR and transfection procedure, I couldn't find any normalization calculation! Did the authors normalize the miR-802 CT values to one or more housekeeping gene?

Our response to issues 4

Thank you for your advice, and we apologize for the ambiguous formulation. We have rewritten the qPCR and transfection procedure in the revised manuscript. The miR-802 CT values were normalized to one housekeeping gene (U6 snRNA), and TaqMan probes (Ambion) were used to confirm our results.

To facilitate your check, the main correction in revision was listed below.

Plasmid and Transient Transfections

Min6 cells ($\sim 5 \times 10^5$) or isolated islets were seeded in six-well plates in culture medium without antibiotics and transfected with full-length cDNA encoding *Fzd5*, *NeuroD1*, *CREB*, *Sox6*, *ADA-Foxo1* or **control plasmid (non-coding)** using Lipofectamine 2000 reagent (Invitrogen) according to the manufacturer's instructions. 6 h post transfection, medium was replaced with fresh medium containing antibiotics. At 48h post-transfection, the cells were harvested and analyzed by immunoblotting and qPCR for the relative level of various proteins and mRNA.

miR-802 duplex mimics (50 nM), 2'-O-methylated single-stranded miRNA antisense oligonucleotides (*anti-miR-802*, 100 nM), and **negative controls at the same concentration were obtained from GenePharma (Shanghai, China)**. For transient transfection, Lipofectamine 2000 reagent (Invitrogen) was mixed with miRNA mimics/inhibitors as previously described¹³, **mimics NC or inhibit NC was transfected at the same concentration as negative control**. At 48h post-transfection, the cells were harvested and analyzed by qPCR for the relative level of *miR-802*.

Small Interfering RNA (siRNA) for Protein Expression Silencing in cells

Small interfering RNAs (siRNAs) were designed and synthesized by Ribobio (Guangzhou, Guangdong, China). For transient transfection, isolated mouse islets or Min6 cells ($\sim 5 \times 10^5$) were seeded in six-well plates cultured in media without antibiotics and transfected with Lipofectamine 2000 reagent (Invitrogen) according to the manufacturer's instructions. Cells were transfected for 6 h with the *Fzd5*, *NeuroD1* and *Foxo1* siRNA at a final concentration of 50 nM or **with control siRNA (non-targeting siRNA) at the same concentration** before changing to fresh media including antibiotics. At 48 h after transfection, cells were lysed to extract total RNA or protein to measure the knockdown efficacy. Sequences of siRNA and shRNA were listed in table S7.

Virus (Lentivirus) infection

Min6 cells were seeded in six-well plates at a density of about 5×10^5 cells with Lentivirus encoding either *miR-802* (*Len-miR-802*, 5 μ l/ml) or *miR-802* sponge (*Len-anti-miR-802*, 5 μ l/ml) plus Polybrene for 48 h as previously described¹⁸. **For comparison, scramble (lentiviral particles without targeting any specific region) served as control.**

Real-time PCR

Islets (100 islet per group) and Min6 cells were isolated using TRIZOL (Invitrogen), cDNA was generated using HiScript Q RT SuperMix for qPCR (Vazyme, China) and real-time PCR assays

were conducted with a LC480 Light Cycler (Roche, Germany) using the applied primer sequences listed in Table S9. Relative expression of genes was determined using a comparative method ($2^{-\Delta CT}$). *U6* and *GAPDH* were used as internal standards for miRNAs and mRNAs, respectively. For *miR-802* and *U6*, TaqMan probes (Ambion) were used to confirm our results.

5) Concerning the transfection of isolated islets which is a difficult procedure, do the authors know if the transfections of *miR-802* also reached the inner β -cells within the islets or only the outer part of β -cells in the islet mantle was transfected?

Our response to issues 5

Thank you for that good question. To investigate whether *miR-802* also reached the inner β -cells within the islets, *miR-802/miR-802* inhibit visualized by Cy3-conjugated was designed from GenePharma (Shanghai, China). 50 nM *miR-802-CY3* or 100 nM anti-*miR-802-CY3* or 50 nM Cy3-control was transfected into islets for 48 h. then Z-stack images via Cy3 were captured from top to equatorial plane of islet in 15 μ m thickness. Scale bar is 100 μ m, revealing that the transfections of *miR-802/miR-802* inhibit also reached the inner β -cells within the islets (see below).

Z-stack images via Cy3 were captured from top to equatorial plane of islet in 15 μ m thickness. Scale bar is 100 μ m. Transfection procedure was performed in triplicate.

6) The confocal image of islet is not a good quality image since not only insulin but also the DAPI intensity in *miR-802* is also affected and reduced. If possible change the figure and keep the confocal setting the same for the both categories (3a). If possible, put a separated set of the image in supplementary to be easier for reader!

Our response to issues 6

Thank you for your meaningful comments. We have replace Figure 3a by images with higher quality. And put a separated set of the image in the revised manuscript. We hope the changes would make you and other readers much easier to understand our manuscript.

To facilitate your check, the main correction in revision was listed below.

Figure 3 *miR-802* mimics was transfected into primary islets and Min6 cells for 48 h. Then, qRT-PCR was used to evaluate the *Ins1* and *Ins2* levels, followed by immunostaining for DAPI (blue) and insulin (red) (Magnification: 20× or 40×, scale bar: 50 μm or 20 μm), (a) in islets and (b) in Min6 cells.

7) Why insulin content of islets (Figure 3c) is not related to the protein level of islet and shown as per islet which is a weak way of expressing results (compare with Figure 3d)?

Our response to issues 7

Thank you for your advice. In our research, Islet content studies were performed on size-matched islets. And based on the literature¹⁴, insulin levels were measured by ELISA or radioimmunoassay, and Values were normalized to islet. To be easier for reader, we used islets with better function to repeat the experiments and the insulin content and insulin secretion of islet were normalized to the total protein.

To facilitate your check, the main correction in revision was listed below.

miR-802 mimics was transfected into primary islets and Min6 cells for 48 h. then, insulin content was measured using ELISA in islets (c). The experiments above was performed in triplicates, and each group contained three batches of individual samples. The data are represented as mean ± SD, ** $p < 0.01$.

8) Regarding the measurement of GSIS, why the authors not relating secreted insulin to the islet insulin content or islet protein in figure 3e. The insulin secretory response of islets is very poor and the basal secretion is very high compared to previously published data (cf Zhang et al, Cell Metabolism 2019)! Do the authors have any explanation for these? Could be due to direct incubation of islets after culture period? Why authors did not preincubated the islets for a short period (like 30-45 min) in KRBH buffer? For overnight culture, the RPMI1640 is more suitable! This might cause the islets to leak insulin and therefore basal insulin release is high? Any explanation to this?

Our response to issues 8

Thank you for your advice.

First, in our research, Islet secretion studies were performed on size-matched islets. And based on the literature^{11,14}, insulin levels were measured by ELISA or radioimmunoassay, and Values were normalized to islet. To be easier for reader, we used islets with better function to repeat the experiments and the insulin content and insulin secretion of islet were normalized to the total protein.

Second, as previous reports, isolated islets were also cultured in RPMI1640¹⁹. We agree with the reviewer's point, for overnight culture, which might cause the islets to leak insulin. According to your suggestions, we have modified the GSIS method according to the reference^{11,14} and used islets with better function to evaluate the effect of miR-802 on glucose stimulate insulin secretion. The results showed that the glucose stimulation index was increased. We hope the changes would make you and other readers much easier to understand our manuscript.

To facilitate your check, the main correction in revision was listed below.

Methods

Glucose-stimulated Insulin Secretion (GSIS) in Mouse Islets

Mouse pancreatic islets (30 islets/well in 48-well plate) were collected under a stereomicroscope at room temperature and cultured in 2.5 mM glucose in the absence or presence of test agents for 48 h. Thereafter the islets were washed and preincubated for 30 min at 37 °C in KRBH buffer, supplemented with 0.2% bovine serum albumin and 2.5 mM glucose. After preincubation, the buffer was changed to a medium containing either 2.5 mM, 33.3 mM glucose or 35 mM KCl. The islets were then incubated for 1 h at 37 °C. Immediately after incubation an aliquot of the medium was removed for analysis of insulin via mice insulin ELISA kit (ExCell Bio, Shanghai, China), and the islets were lysed to extract total protein, the amount of insulin secretion was normalized by the total cellular protein content¹¹....

Figure 3

Figure 3 *miR-802* mimics was transfected into primary islets for 48 h. insulin secretion was analyzed by GSIS assay in islets (e). (j) Static insulin secretion was evaluated in islets from 15-week-old *miR-802 KI* and control mice (n = 5-7) at indicated glucose and KCl concentrations. All data are represented as mean ± SD. ***p* < 0.01; ****p* < 0.001.

Figure 4

Figure 4 *anti-miR-802* was transfected into primary islets for 48 h. Insulin secretion was analyzed by GSIS assay in islets (e). (i and j) Static insulin secretion performed with islets from 5- (i) and 35-week-old (j) *miR-802 KO* and control mice (n = 5-7) at indicated glucose and KCl concentrations. All data are represented as mean ± SD, ***p* < 0.01; ****p* < 0.001.

9) Surprisingly neither islets nor Min6 cells are responding to high concentration of KCl? Did the addition of KCl (35 mM) compensated by the removal of Na?

Our response to issues 9

Thank you for your good comments and we apologize for the ambiguous formulation.

First, in our study, islets and Min6 cells were responding to high concentration of KCl (see below). And in the revised manuscript, we have rewritten the result of insulin secretion when islets or Min6 cells were stimulated with 35 mM KCl, which marked red in the text.

Second, 35 mM KCl was added in KRBH balanced buffer (KRBH: 115 mM NaCl, 4.8 mM KCl, 2.5 mM CaCl₂, 1.2 mM MgSO₄, 1.2 mM KH₂PO₄, 20 mM NaHCO₃, and 16 mM HEPES; pH 7.4) containing 0.2% BSA to stimulate Min6 cells or islet. The KRBH buffer contained Na.

To facilitate your check, the main correction in revision was listed below.

..... Overexpression of *miR-802* decreased insulin secretion at high glucose, even in the presence of 35 mM KCl, a general depolarizing agent (Figure 3e, f)....

....In addition, insulin content and sensitivity to glucose or 35 mM KCl were decreased in the islets derived from *miR-802* *KI* mice (Figure 3i, j, n=5-7)....

Figure 3 *miR-802* mimics was transfected into primary islets and Min6 cells for 48 h. Then, insulin secretion was analyzed by GSIS assay in islets (e, n=7, 8 weeks old) and Min6 cells (f). (j) Static insulin secretion was evaluated in islets from 15-week-old *miR-802* *KI* and control mice (n = 5-7) at indicated glucose and KCl concentrations. All data are represented as mean \pm SD. ** $p < 0.01$; *** $p < 0.001$.

.....Next, we performed glucose challenge experiment by using primary islets. Knockdown of *miR-802* increased insulin secretion both in high glucose and 35 mM KCl even at low glucose exposure (Figure 4e, f)....

.....To further characterize the increased insulin secretory function of *miR-802* *KO* mice, glucose-induced insulin secretion assays were performed in islets of 5- and 35-week-old mice (n=5-7). In both age groups, when treated with 33.3 mM glucose, insulin secretion was increased 1.5- to 4-fold and insulin secretion was increased by 2-fold when treated with 35 mM KCl (Figure 4i, j).....

Figure 4 *anti-miR-802* was transfected into primary islets and Min6 cells for 48 h. Insulin secretion was analyzed by GSIS assay in islets (e, n=7, 8 weeks old) and Min6 cells (f). (i and j) Static insulin secretion performed with islets from 5- (i) and 35-week-old (j) *miR-802 KO* and control mice (n = 5-7) at indicated glucose and KCl concentrations.

10) In the figure 4a the confocal image is not of good quality where the insulin staining should at least be as the control in Figure 3k.

Our response to issues 10

Thank you for your meaningful comments. We have replace Figure 4a by images with higher quality, and put a separated set of the image in the revised manuscript. We hope the changes would make you and other readers much easier to understand our manuscript. To facilitate your check, the main correction in revision was listed below.

Figure 4 *anti-miR-802* was transfected into primary islets and Min6 cells for 48 h. Then, qRT-PCR and immunostaining for DAPI (blue) and insulin (red) were performed (Magnification: 20× or 40×, scale bar: 50 µm or 20 µm) in islets (a, n=5, 8 weeks old) and Min6 cells (b),

11) Why the response of islets to high glucose is almost equal to basal in control group (*anti-NC*)? I roughly estimated the % of insulin secretion (insulin secretion in Figure 4e / insulin content in 4c) in control group and in *miR-802 KO* islets and there was hardly any differences in the values. Any comments from authors? Is there any significant difference between 2.5 and 33.3 in *anti-miR-802*?

Our response to issues 11

Thank you for your meaningful comments. First, there was no difference between 2.5 and 33.3 in anti-miR-802, but isolated islets of control mice were transfected with anti-NC or anti-miR-802, then cultured in RPMI 1640 media, containing 11 mM glucose supplemented with 10% FBS, 100 IU/mL penicillin, and 100 µg/mL streptomycin for 48 h. As reviewer's concern raised in the previous point, the RPMI1640 might more suitable. Therefore, the glucose stimulus index is low, resulting in no significant difference. In the revised manuscript. We have modified the GSIS methods according to your suggestion, isolated islets were not overnight in 1640 media, while islets were resuspended in KRBH balanced buffer containing 0.2% BSA supplemented with 2.5 mM glucose streptomycin for 30 min, the results showed that the glucose stimulation index was increased and difference was obvious (see below). We hope the changes would make you and other readers much easier to understand our manuscript.

To facilitate your check, the main correction in revision was listed below.

Islet dispersion and insulin secretion cells

Mouse pancreatic islets were incubated in Ca²⁺ and Mg²⁺ free PBS for 28 min at 37 °C and centrifuged for 5 min at 200 g. Supernatant was removed and islets were resuspended in KRBH balanced buffer (KRBH: 115 mM NaCl, 4.8 mM KCl, 2.5 mM CaCl₂, 1.2 mM MgSO₄, 1.2 mM KH₂PO₄, 20 mM NaHCO₃, and 16 mM HEPES; pH 7.4) containing 0.2% BSA supplemented with 2.5 mM glucose, 10% FBS, 100 IU/mL penicillin, 100 µg/mL streptomycin, 2 mM L-glutamine and 10 mM nicotinamide,....

Glucose-stimulated Insulin Secretion (GSIS) in Mouse Islets

Mouse pancreatic islets (30 islets/well in 48-well plate) were collected under a stereomicroscope at room temperature and cultured in 2.5 mM glucose in the absence or presence of test agents for 48 h. Thereafter the islets were washed and preincubated for 30 min at 37 °C in KRBH buffer, supplemented with 0.2% bovine serum albumin and 2.5 mM glucose. After preincubation, the buffer was changed to a medium containing either 2.5 mM, 33.3 mM glucose or 35 mM KCl. The islets were then incubated for 1 h at 37 °C. Immediately after incubation an aliquot of the medium was removed for analysis of insulin via mice insulin ELISA kit (ExCell Bio, shanghai, China), and the islets were lysed to extract total protein, the amount of insulin secretion was normalized by the total cellular protein content¹¹....

Figure 4

Figure 4 *anti-miR-802* was transfected into primary islets and Min6 cells for 48 h. Then, and insulin content was evaluated in islets (c, n=5, 8 weeks old). Insulin secretion was analyzed by GSIS assay in islets (e, n=7, 8 weeks old). All data are represented as mean \pm SD, ** $p < 0.01$; *** $p < 0.001$.

12) Line 491 “od” should be “of”.

Our response to issues 12

Thank you for your suggestion, and we apologize for the ambiguous formulation. We have corrected it.

13) Description of the efficiency Results for the Figure S7a (line 506) should according to the appearance in the figure i.e in the figure S7 overexpression is the first and knockdown later shown.

Our response to issues 13

Thank you for your suggestion and we are sorry for our incorrect writing. We have corrected “The efficiency of knockdown and overexpression was about 70% and 300-fold, respectively (Figure S7a)” as “The efficiency of overexpression and knockdown were about 300-fold and, 70% respectively (Figure S7a)”.

14) *db/db* is a communally used animal model of diabetes. Authors do not say anything about the background of choosing this model and neither discuss their findings in *db/db* mouse islets in relation to previously reported findings. They should also add some sentence about their findings in *db/db* mouse islets, which can be added to earlier, and very recently findings of mitochondrial dysfunction of this diabetic animal model shown by Zhang et al Cell Metabolism 2019.

Our response to issues 14

Thank you for your good advice. In the revised manuscript, we added some sentence to descript about the background of choosing *db/db* model mice and also discussed our findings in the *db/db* mouse islets in relation to previously reported findings.

To facilitate your check, the main correction in revision was listed below.

Result

Expression of *miR-802* is upregulated in the islets of obese mouse models

To identify miRNAs that are dysregulated during obesity and that may contribute to β cell dysfunction, we performed “miRNome” expression profiling using RNA-seq analysis on RNA isolated from islets of two mouse models of obesity: high fat diet (HFD)-fed mice compared to normal chow diet (NCD) fed mice and mice homozygous for the diabetes *db* mutation of the leptin receptor ($Lepr^{db/db}$) compared to wild type controls. *Lepr^{db/db} mouse was a faithful model of human obesity and β cell failure leading to overt T2D²⁰. Thus, we chose the islets from $Lepr^{db/db}$ mice to mimic the islets from human T2D.*

Discussion

The relationship between obesity and β cell dysfunction remains a fundamental research topic for the elaboration of novel therapeutic avenues enhancing insulin secretion and improving metabolic control of T2D patients. Here, we revealed that *miR-802* was a fundamental regulator of insulin secretion in pancreatic β cells, whose expression was increased in the islets of dietary obese mice and in the islets of diabetic models ($Lepr^{db/db}$ mice on BKS background), which developed age-dependent hyperglycemia and reduced plasma insulin levels due to β cell dysfunction. Like islets from T2D donors^{11,21}

Reviewer # 3

In this study, the authors showed an involvement of miR-802 expression in islets in insulin resistance and type II diabetes using cells and mouse models in which miR-802 is overexpressed or silenced/disrupted. The conclusions are clear and seem to be supported by the results. However, there are parts that are difficult to evaluate/interpret the results because of poor and insufficient description of materials and tools they used.

Response: Thanks for your positive comments. Following your suggestion, we have provided sufficient materials and tools to further illustrate our results. We hope the explanations and changes above would make you and other readers much easier to understand our manuscript

1. *The authors do not provide the detailed information about the mice used in the study. For example, I could not find the gRNA sequences used to generate miR-802 knockout and knock-in mice (lines 87 and 90). In addition, no information about vector sequences for generation of these mice are provided. What is the length of homology arms used? What kind of stopper sequence and polyA are used? It is desirable to provide the complete DNA sequences of these vectors as supplementary materials (and/or deposit the mouse lines to public resource center). In addition, DNA sequences of all the plasmids including reporter expression constructs can be supplied (or submit the plasmids used in their research to a public repository such as Addgene).*

Our response to suggestion 1

Thank you for your kindly advice.

First, miR-802 knockout and knock-in mice were constructed from Model Animal Research Center of Nanjing University (Nanjing, China). We have added the sgRNA sequences used to generate miR-802 knockout and knock-in mice, which were listed in table S1.

Second, the length of homology arms were ~2 kb. The complete DNA sequences of these vector, included the stopper sequence and polyA, were provided as supplementary materials and our mouse lines of miR-802 knockout and knock-in mice were deposited in Model Animal Research Center of Nanjing University (Nanjing, China).

In addition, we have provided the sequence map, which included all the information of miR-802 knockout and knock-in mice (listed below) and DNA sequences of the plasmids including reporter expression constructs have been supplied, which could open via software SnapGene or NTI.

To facilitate your check, the correction in revision was listed below.

Table S1 The sgRNA sequences of miR-802 knockout and knock-in mice.

sgRNA name	sgRNA sequence(5'-3')	PAM
miR-802 knockout-5S4	CTTGGCCTCCCGGCC	TGG
miR-802 knockout-3S6	ACTCGGATAGTATGCACACT	CGG
miR-802 knock-in-H11-S2	CTGAGCCAACAGTGGTAGTA	AGG

The complete DNA sequence for generation miR-802 knock-in mice.

Dark yellow: CAG Promoter sequence;

Grey: loxp-stop-loxp sequence;

Yellow: mir802 sequence;

Red: polyA sequence.

CATTCTCCATTTTCATAATATTCTATTGGACTTTGACTGCAGGGGCCTCCAAGTCTTGACA
GTAGATTATAATCCTTCAGCTGCCACTCTACTGGAGGAGGACAAACTGGTCACTTTTC
AGCAAAACCTGGCTGTGGATCAGGGCAGTCTGGTACTTCCAAGCTCATTAGATGCCAT
CATGCTCTCACTGCCTCCTCAGCTTCAAGAGGAATCTGGAAAAGCAGTCCCACTGGT
CAGGAAAGGAACACTAGTGCACCTTATCCTGGGTGTCTGCTGAGCTCGAGAGTGCACCT

TAATTAAGTCGACATTGATTATTGACTAGTTATTAATAGTAATCAATTACGGGGTCAT
TAGTTCATAGCCCATATATGGAGTTCCGCGTTACATAACTTACGGTAAATGGCCCG
CCTGGCTGACCGCCCAACGACCCCGCCATTGACGTCAATAATGACGTATGTTCC
CCATAGTAACGCCAATAGGGACTTTCCATTGACGTCAATGGGTGGAGTATTTACG
GTAAACTGCCCACTTGGCAGTACATCAAGTGTATCATATGCCAAGTACGCCCCCTA
TTGACGTCAATGACGGTAAATGGCCCGCCTGGCATTATGCCCAGTACATGACCTT
ATGGGACTTTCTACTTGGCAGTACATCTACGTATTAGTCATCGCTATTACCATGG
TCGAGGTGAGCCCCACGTTCTGCTTCACTCTCCCCATCTCCCCCCCCTCCCCACC
CCCAATTTTGTATTTATTTATTTTTTAATTTTGTGTCAGCGATGGGGGGCGGGGG
GGGGGGGCGCGGCCAGGGCGGGGCGGGGCGGGGCGAGGGGCGGGGCGGGGCG
GAGGCGGAGAGGTGCGGCGGCAGCCAATCAGAGCGGCGCGCTCCGAAAGTTTC
CTTTTATGGCGAGGCGGCGGGCGGGCGGCCCTATAAAAAGCGAAGCGCGCGGC
GGGCGGGAGTCGCTGCGCGCTGCCTTCGCCCCGTGCCCGCTCCGCCGCCGCC
TCGCGCCCGCCCGCCCGGCTCTGACTGACCGCGTACTCCACAGGTGAGCGGG
CGGGACGGCCCTTCTCCTCCGGGCTGTAATTAGCGCTTGGTTAATGACGGCTTG
TTTCTTTTCTGTGGCTGCGTGAAAGCCTTGAGGGGCTCCGGGAGGGCCCTTTGT
GCGGGGGGAGCGGCTCGGGGGGTGCGTGCGTGTGTGTGTGCGTGGGGAGCGC
CGCGTGCGGCTCCGCGCTGCCCGGCGGCTGTGAGCGCTGCGGGCGCGGCGCGG
GGCTTTGTGCGCTCCGCAGTGTGCGCGAGGGGAGCGCGGCCGGGGGCGGTGCC
CCGCGGTGCGGGGGGGGCTGCGAGGGGAACAAAGGCTGCGTGCGGGGTGTGT
GCGTGGGGGGGTGAGCAGGGGGTGTGGGCGCGTGGTTCGGGCTGCAACCCCC
CTGCACCCCCCTCCCCGAGTTGCTGAGCACGGCCCGGCTTCGGGTGCGGGGCTC
CGTACGGGGCGTGGCGCGGGGCTCGCCGTGCCGGGCGGGGGGTGGCGGCAGG
TGGGGGTGCCGGGCGGGGCGGGGCCGCTCGGGCCGGGAGGGCTCGGGGGA
GGGGCGCGGCGGCCCCCGGAGCGCCGGCGGCTGTGAGGGCGCGGCGAGCCGC
AGCCATTGCCTTTTATGGTAATCGTGCGAGAGGGCGCAGGGACTTCCTTTGTCC
AAATCTGTGCGGAGCCGAAATCTGGGAGGCGCCGCCGACCCCCCTTAGCGGGC
GCGGGGCGAAGCGGTGCGGCGCCGGCAGGAAGGAAATGGGCGGGGAGGGCCT
TCGTGCGTCCGCGCGCCCGCTCCCTTCTCCCTCTCCAGCTCGGGGCTGTCC
GCGGGGGGACGGCTGCCTTCGGGGGGGACGGGGCAGGGCGGGGTTCCGGCTTCT
GGCGTGTGACCGGCGGCTCTAGAGCCTCTGCTAACCATGTTTCATGCCTTCTTCTT
TTTCTACAGCTCCTGGGCAACGTGCTGGTTATTGTGCTGTCTCATCATTTTGGC
AAAATAACTTCGTATAGCATAACATTATACGAAGTTATCTGTAAGTCTGCAGAAATTGATG
ATCTATTAACAATAAAGATGTCCACTAAAATGGAAGTTTTTCTGTACACTTTGTAA
GAAGGTGAGAACAGAGTACCTACATTTTGAATGGAAGGATTGGAGCTACGGGGGTG
GGGGTGGGGTGGGATTAGATAAATGCCTGCTCTTACTGAAGGCTCTTACTATTGCTTT
ATGATAATGTTTCATAGTTGGATATCATAATTTAAACAAGCAAAACCAAATTAAGGGCA
GCTCATTCTCCACTCATGATCTATAGATCTATAGATCTCTCGTGGGATCATTGTTTTTC
TCTTGATTCCCACTTTGTGGTTCTAAGTACTGTGGTTTCCAAATGTGTCAGTTTCATAGC
CTGAAGAACGAGATCAGCAGCCTCTGTTCCACATACACTTCATTCTCAGTATTGTTTTG
CCAAGTTCTAATTCCATCAGAAGCTTGAGATCTGCGACTCTAGAGGATCGACTGTGCC
TTCTAGTTGCCAGCCATCTGTTGTTTGGCCCTCCCCGTGCCTTCTTGACCCTGGAAG
GTGCCACTCCCACTGTCTTTTCTAATAAAAATGAGGAAATTGCATCGCATTGTCTGAGT
AGGTGTCATTCTATTCTGGGGGGTGGGGTGGGGCAGGACAGCAAGGGGGAGGATTGG

GAAGACAATAGCAGGCATGCTGGGGATGCGGTGGGCTCTATGGCTGCGACTCTAGAGG
ATCATAATCAGCCATACCACATTTGTAGAGGTTTACTTGCTTTAAAAACGTTTAAACC
TCCCACACCTCCCCCTGAACCTGAAACATAAAATGAATGCAATTGTTGTTGTTAACTTG
TTTATTGCAGCTTATAATGGTTACAAATAAAGCAATAGCATCACAAATTCACAAATAAA
GCATTTTTTTCCTGCACTTAGTTGTGGTTTGTCCAAACTCATCAATGTATCTTATCATG
TCTGGATCTGCGACTCTAGAGGATCATAATCAGCCATACCACATTTGTAGAGGTTTACT
TGCTTTAAAAACCTCCCACACCTCCCCCTGAACCTGAAACATAAAATGAATGCAATT
GTTGTTGTTAACTTGTTTATTGCAGCTTATAATGGTTACAAATAAAGCAATAGCATCACA
AATTCACAAATAAAGCATTTTTTTCCTGCACTTAGTTGTGGTTTGTCCAAACTCATC
AATGTATCTTATCATGTCTGGATCTGCGACTCTAGAGGATCATAATCAGCCATACCACAT
TTGTAGAGGTTTACTTGCTTTAAAAACCTCCCACACCTCCCCCTGAACCTGAAACAT
AAAATGAATGCAATTGTTGTTGTTAACTTGTTTATTGCAGCTTATAATGGTTACAAATAA
AGCAATAGCATCACAAATTCACAAATAAAGCATTTTTTTCCTGCACTTAGTTGTGGT
TTGTCCAAACTCATCAATGTATCTTATCATGTCTGGATCCCCATCAAGCTGATAACATAC
GCTCTCCATCAAAAACAAAACGAAACAAAACAAACTAGCAAAATAGGCTGTCCCCAGT
GCAAGTGCAGGTGCCAGAACATTTCTCTATAACTTCGTATAGCATAATTATACGAAGTT
ATCAATCATTACAGCTACGACTTTAAAGATGGATCGTTGCCCGAATTTCCGACTATTTTC
CTTCTAAATGACTGTTTCTTTACAGATCTAAACTAATGTAAAGAGACAGGTTGTCCCCG
TGCCAGGAGCGAACAAGCAAAGTAGGACAGAGGCTGTTTTCTAGGGAAGGAATGTGG
CAAGATAATGCCTGCTCTGTCTCCCGCAGCCGCTAGCTCAGGCAGAGACGGAAGAGGA
TACTGCCTGCCACGGCCTGGTTGTGAGCACAGGCTCCCCACCTGACTCTACATAACCT
ACCGACTGCGGTCTATTATTTGCAATCAGTAACAAAGATTCATCCTTGTGTCAATCATA
CAACACGGAGAGTCTTTGTCACTCAGTGTAATTAATAGCCTTCACCTCGAAAGGGAAG
ATGAGAGGACGCTGTTTCGCACGTGCCTGGGGTGTCTCGGAGGGCGTTTCTCGTGGAT
AAGCGCTGCCTTCTGACATAGTAAGCCAATGGAACTCCAGCTGCAAGGGGGAGGTGG
CAGGGGCGGTGGAGGTGAGGTAAGCTGAGTGTGTTTGGGGAGGCTGGGCAGGGAGC
CCAGTCAGCAAGTGCTCATTGAGGCCCTGAGCTCTGTCCCTGGAACCCACATTAAG
ACATACATGTGATCCAAGCACTGGAGAGAGAGGGGACAGATCCCTGGGGGCTGACTG
ACACTCCTCAGGTGCAGGCTGCCTATCAGAAGGTGGTGGCTGGTGTGGCCAATGCCCT
GGCTCACAAATACCACTGAGATCTTTTTCCCTCTGCCAAAAATTATGGGGACATCATGA
AGCCCCTGAGCATCTGACTTCTGGCTAATAAAGGAAATTTATTTTCATTGCAATAGTGT
GTTGGAATTTTTTGTGTCTCTCACTCGGAAGGACATATGGGAGGGCAATCATTAAAA
CATCAGAATGAGTATTTGGTTTAGAGTTTGGCAACATATGCCCATATGCTGGCTGCCATG
AACAAAGGTTGGCTATAAAGAGGTCATCAGTATATGAAACAGCCCCCTGCTGTCCATTC
CTTATTCCATAGAAAAGCCTTGACTTGAGGTTAGATTTTTTTTTATATTTGTTTTGTGTTA
TTTTTTTCTTTAACATCCCTAAAATTTCCCTTACATGTTTTACTAGCCAGATTTTTCTCC
TCTCCTGACTACTCCAGTCATAGCTGTCCCTCTTCTCTTATGGAGATCCCTCGACCTGC
AGGCGGCCGCGTAAGGGCAGGATGTGTCAAACCTGCCAATAGAGAACTACTTACTCTTC
AGGCTGAAGCTGATGGAACAGGTAACAAAGGCAAACTAATCATGATCAGCAAGAT
GAAGCAGAAAGGGAACAAGGGGATATTAATGTGTATAGACACGCTAGAGAGATGGCT
CAGCAGTTAAGAGAACTAGCTGGTCTTTCAGAGGTCCTGAGATCAATTTTAGACACCC
ACATGGTGGCTCATGACCATCTATCTATAAATGGATCTGATTTTCATGTCTGGCAGTGTA
CAGAAGCTAACT

The complete DNA sequence for generation miR-802 knockout mice.

Yellow: mir802 sequence;

Red: loxp sequences.

TTTTACAGCCTCGATTAAAAGATAGAAGTCAGCTATGCCGAGAATGATTGCTGCAGACT
TGGGATAAGGCTTAAATGGAAACCACAGGTTCAAGGGTGCCAGCATCGCATTGTCTGA
GTAGGTGGGATCCGGTACC**ATAACTTCGTATAATGTATGCTATAACGAAGTTAT**ATGTTCTT
GCCAAGGTCAGTTGGGGGCCGGGAGGCCAAGCTCAGACTCTTGTTTTTTTCTAAGGGG
AATGTCGGTAACATATGCTTTCCATTGCTCCACTTTTGACCATTACATACTGGGGGGCT
GTGTGGTTGGCAGCAGGTCCTATTTTAGGAGTCCTTGAAAACCTTCCCTTTTTAACAG
TTCTCATTCTGGCACATACCTGTGTTTTAAGCAAAGGAAAACCCAAATCCAAGATAGG
AAAAGAGAAACATGTGTACAGACACGGGGGATTTTTGCAGCCCTGGAGTCACACAAT
AAGAAAGCAATCATTACAGCTACGACTTTAAAGATGGATCGTTGCCCGAATTTCCGACT
ATTTTCCTTCTAAATGACTGTTTCTTTACAGATCTAAACTAATGTAAAGAGACAGGTTGT
CCCCGTGCCAGGAGCGAACAAGCAAAGTAGGACAGAGGCTGTTTTCTAGGGAAAGGAA
TGTGGCAAGATAATGCCTGCTCTGTCTCCCGCAGCCGCTAGCTCAGGCAGAGACGGAA
GAGGATACTGCCTGCCACGGCCTGGTTGTGAGCACAGGCTCCCCACCTGACTCTACA
TAACCTACCGACTGC**GGTCCTATTATTTGCAATCAGTAACAAAGATTCATCCTTGTGTCA**
ATCATAACAACCGGAGAGTCTTTGTCACTCAGTGTAATTAATAGCCTTCACCTCGAAAG
GGAAGATGAGAGGACGCTGTTTCGCACGTGCCTGGGGTGTCTCGGAGGGCGTTTCTC
GTGGATAAGCGCTGCCTTCTGACATAGTAAGCCAATGGAACTCCAGCTGCAAGGGGGA
GGTGGCAGGGGCGGTGGAGGTGAGGTAAGCTGAGTGTGTTGGGGAGGCTGGGCAG
GGAGCCCAGTCAGCAAGTGCTCATTGAGGCCCTGAGCTCTGTCCCTGGAACCCACATT
AAAAGACATACATGTGATCCAAGCACTGGAGAGAGAGGGGACAGATCCCTGGGGGCT
GACTGACCAGCCTTGCTAAATCAGCAAGCTCTAGACCAGCGAAGACCCCATCTCAAAG
GCAGTGGACAGAGCCTGAGGAATAATACCCAAGGTTGACCTCTAACCTCCACATGCAC
AAATGCATGTGTGCACACACGTACGCTTGCCTGTCCACACACTAACATGCACATATGAG
TACTCATGCACTGACACTGTAGCCTCTACAAGGGCCTACTGCAAACCACGCTACCTATT
CAGTCAGGAACCTGTCAGAGAAAGGTAATAGGCTCAGACTTCAAAAAGACAGCACTG
AACACAGGCTTGAAGGTCCGAGGAGTGGTGTCCCGTATACAGTTGTTGTGCCAGTCAA
GAATTAAGACAATCCCCACAGACATGTCCAAGGCCAGTCTGGGCTGAGCGCTCCCTC
ACTGAGATTTCTTCTGAGGCAGGCAACCGTAGCCTGGGACAACCTTTACAATTTATTCC
AACTTGGGCTCTGTGCTTCGTGACAAATATACCATTTTATTAAAGAGGATTCATTGTGTA
CTTATTTGCAAGTATGACTTTAACTCCTCATCCCGGCACACCATTTTGTGCTGACAGAGT
CCTAAGTGTTTAAAGAGAGTGAGTGTCCGCATCCCTCCACCCCAAGCTCTTCTGACGC
GTACTGTTGCTTACTATCCGTCTGAGGCGGAAAGAACCAGGGATCCGGGCC**ATAACT**
TCGTATAATGTATGCTATAACGAAGTTATGGACTAACAGAAGAACCCGTTGTGAGTGTGC
ATACTATCCGAGTCTCTGTGTATATATAAACATGTTCAATTTGTGTGTGCATGCACACAGG
TGTGCAGGTGTGTGTACACATGGGAGCACACGTGTGTGGAGGCCAGTGCTGGTGTGG
GAGTCTCCTTTGACCTCTCTTTACCTTATTCTTCAAGGCAGGGTCTCTCCATTGACCAC
AGGGCTAATCAGTAGGGCTATCCAGCTTGCTCCAAGGACTCTGTCTTTTCTTTCT

Sequence map of miR-802 knock-in mice.

Sequence map of miR-802 knockout mice.

2. How did the authors obtain RIP-Cre mice? If the authors obtained RIP-Cre strain from the public resource center, official number and name of this strain should be provided.

Without these information, it is difficult for readers/researchers to interpret the results properly and to independently replicate the experiments described by the authors.

Our response to suggestion 2

Thank you for your meaningful comments, and we agree with the reviewer's point. The RIP-cre mice were obtained from Jackson laboratory (003573-B6.Cg-Tg (Ins2-cre) 25Mgn/J). By the way, we have added this information on the materials and methods of animal care, which have marked red in the revised manuscript.

To facilitate your check, the correction in revision was listed below.

Animal care

Lepr^{db/db}, Lepr^{db/-} and C57BL/6J mice (7-8 weeks old) were obtained from Model Animal Research Center of Nanjing University (Nanjing, China). **RIP-Cre mice were obtained from Jackson laboratory (003573-B6.Cg-Tg (Ins2-cre) 25Mgn/J).** All animals were of pure C57BL/6 background except Lepr^{db/db} and Lepr^{db/-} mice, which were of BKS background (Janvier). C57BL/6J mice were fed high fat diet (HFD) for 8 weeks (D12494, 60% energy from fat) according to the criteria defined by Peyot ML²², and weighted between 40 g and 45 g. The control groups were fed with normal diet (D12450J, 10% energy from fat), and weighted between 23 g and 25 g. Mice were housed in groups of 3-5 animals/cage on a 12h light/dark cycle in an SPF facility at 22-24 °C. All care and handling of animals were carried out according to the international laws and policies (EEC Council Directive 86/609, 1987) and approved by the animal ethics committee of China Pharmaceutical University (Nanjing, China). **Permit Number: 2162326.** Care of animals was done within institutional animal-care committee guidelines.

Figure legend: RT-PCR of tail genomic DNA isolated from Rip-cre mice (n=6). Cre allele revealed ~100 bp; Internal Positive control revealed 324 bp.

3. Line 82: *The approved number should be provided.*

Our response to suggestion 3

Thank you for your kind advice. We have provided the approved number in the revised manuscript.

4. Line 92: *“donor of two model mice” were co-injected at the same time?*

Our response to suggestion 4

Thank you for your suggestion, and we apologize for the ambiguous formulation. “Donor of two model mice” was co-injected into 0.5 day' zygotes. We have rewritten this part in revised manuscript.

Materials and methods

Generation of a *miR-802* knockout and knock-in mice via CRISPR/Cas9 system

.....Then, Cas9 mRNA, sgRNA, and donor of each model mice were co-injected into 0.5 day' zygotes.....

5. Lines 96-97: Why were PCR genotyping performed two times? What was the knock-in efficiency (how many positive pups were obtained from how many eggs injected)?

Our response to suggestion 5

Thank you for your suggestion, and we apologize for the ambiguous formulation.

Firstly, we identified all the offspring of ICR females (F0 mice) by PCR and sequencing of tail DNA to pick up positive F0 mice. In order to ensure the accuracy of genotype-positive F0 generation mice, before positive F0 mice were crossed with C57BL/6J mice, PCR genotyping was performed once again of tail tip DNA. To make you and other readers much easier to understand our manuscript, we have deleted “Then, positive F0 mice were genotyped using polymerase chain reaction (PCR) of tail tip DNA.”

Second, for miR-802 knock-in mice, we obtained 109 F0 mice from 400 eggs injected, 9 of them were positive F0 mice. And for miR-802 knockout mice, we obtained 61 F0 mice from 400 eggs injected, 2 of them were positive F0 mice.

To facilitate your check, the correction in revision was listed below.

..... Thereafter, 400 zygotes were transferred into the oviduct of pseudopregnant ICR females at 0.5 days post-copulation. 61 *miR-802* knockout F0 mice and 109 *miR-802* knock-in F0 mice were born after 19–21 days of transplantation and all the offspring of ICR females (F0 mice) were identified by PCR and sequencing of tail DNA. The primers are listed in Table S2. Finally, 2 positive *miR-802* knockout F0 mice and 9 positive *miR-802* knock-in F0 mice were crossed with C57BL/6J mice to build up *miR-802* knockout and knock-in heterozygous mice (*miR-802^{fl/wt}*, *miR-802^{ki/wt}*),.....

6. Lines 99-101: When heterozygous knockout mice are crossed with Cre mice (homo or hetero?), the offspring are not homozygous knockout mice.

Our response to suggestion 6

Thank you for your comments and we agree with the reviewer’s point. When heterozygous knockout mice were crossed with Rip-Cre (hetero) mice, the offspring included RIP-Cre *miR-802^{fl/wt}* mice, RIP-Cre *miR-802^{wt/wt}* mice, *miR-802^{fl/wt}* mice and *miR-802^{wt/wt}* mice. Then RIP-Cre *miR-802^{fl/wt}* mice were crossed with *miR-802^{fl/fl}* mice to obtain RIP-Cre *miR-802^{fl/fl}* mice (*miR-802* KO). We have rewritten this part, which marked red in revised manuscript.

To facilitate your check, the correction in revision was listed below.

.....Homozygote *miR-802* knockout and knock-in mice (*miR-802^{fl/fl}*, *miR-802^{ki/wt}*) were obtained from backcross-selfcross of *miR-802^{fl/wt}* or *miR-802^{ki/wt}*. Then *miR-802^{fl/fl}* or *miR-802^{ki/ki}* mice were crossed with Rip-Cre mice to select Rip-Cre *miR-802^{fl/wt}* or Rip-Cre *miR-802^{ki/wt}*. Finally, these mice were crossed with *miR-802^{fl/fl}* or *miR-802^{ki/ki}* mice to delete or overexpress *miR-802* in β cells, respectively.....

7. Line 384: The authors use “Homozygous” here but they mention “heterozygous” in the method section.

Our response to suggestion 7

Thank you for your suggestion, and we apologize for the ambiguous formulation. We have unified it both in result and method section.

To facilitate your check, the correction in revision was listed below.

Method

Generation of a *miR-802* knockout and knock-in mice via CRISPR/Cas9 system

..... Then $miR-802^{fl/fl}$ or $miR-802^{ki/ki}$ mice were crossed with Rip-Cre mice to select Rip-Cre $miR-802^{fl/wt}$ or Rip-Cre $miR-802^{ki/wt}$. Finally, these mice were crossed with $miR-802^{fl/fl}$ or $miR-802^{ki/ki}$ mice to delete or overexpress $miR-802$ in β cells, respectively.....

8. Line 388: How did the authors get “homozygous”-Cre KI animals (Rip-Cre $miR-802^{ki/ki}$)?

Our response to suggestion 8

Thank you for your comment. First, $miR-802^{ki/ki}$ mice were crossed with Rip-Cre mice (heterozygote) to select Rip-Cre $miR-802^{ki/wt}$, then Rip-Cre $miR-802^{ki/wt}$ mice were crossed with $miR-802^{ki/ki}$ mice to obtain RIP-Cre $miR-802^{ki/ki}$ mice ($miR-802$ KI). We have corrected it in the revised manuscript.

To facilitate your check, the correction in revision was listed below.

..... Rip-Cre $miR-802^{ki/ki}$ ($miR-802$ KI) mice were born after mating between Rip-Cre $miR-802^{ki/wt}$ mice and $miR-802^{ki/ki}$ mice (Figure S3g).....

Figure S3 (g) The cross flowchart to obtain $miR-802$ KI mice according to Mendelian inheritance.

9. Lines 388 and 422: Which crosses were used to evaluate mendelian frequencies? Actual number of animals should be provided for each genotype.

Our response to suggestion 9

Thank you for your suggestions. The mating flowchart to evaluate mendelian frequencies was shown in Figure S3g and Figure S4g, the exact amount of each genotype mice was listed in table S11 and table S12. The cross experiment result obeys Mendelian inheritance. Based on previous reports, $miR-802$ are functional components of the GH regulatory network that shapes sex-differential gene expression in mouse liver²³. In our study, we found that most offspring of $miR-802$ knockout or knock-in mice were male, which indicated that $miR-802$ might regulate mouse sex.

To facilitate your check, the correction in revision was listed below.

Figure S3 (g) The cross flowchart to obtain *miR-802 KI* mice according to Mendelian inheritance.

Figure S4 (g) The cross flowchart to obtain *miR-802 KO* mice according to Mendelian inheritance.

Table S11 A the actual number of *miR-802 KI* mice (F0 to F1) for each genotype

Genotype	number	female	male
Rip-Cre miR-802 ^{ki/wt}	28	8	20
miR-802 ^{ki/wt}	24	7	17
Total	52	15	37

Table S11 B the actual number of *miR-802 KI* mice (F1 to F2) for each genotype

Genotype	number	female	male
Rip-Cre miR-802 ^{ki/ki}	48	16	32
Rip-Cre miR-802 ^{ki/wt}	49	16	33
miR-802 ^{ki/ki}	46	15	31
miR-802 ^{ki/wt}	41	14	27
Total	184	61	123

Table S12 A the actual number of *miR-802 KO* (F0 to F1) mice for each genotype

Genotype	number	female	male
Rip-Cre miR-802 ^{fl/wt}	39	12	27
miR-802 ^{fl/wt}	37	10	27
Total	76	24	52

Table S12 B the actual number of *miR-802 KO* (F1 to F2) mice for each genotype

Genotype	number	female	male
Rip-Cre miR-802 ^{fl/fl}	73	21	52
Rip-Cre miR-802 ^{fl/wt}	74	24	50
miR-802 ^{fl/fl}	69	19	50
miR-802 ^{fl/wt}	68	21	47
Total	284	85	199

10. Lines 418-419: With this cross, we can obtain only heterozygotes regarding flox allele.

Our response to suggestion 10

Thank you for your suggestion, and we agree with your points. With this cross, we can obtain only heterozygotes regarding flox allele. In order to obtain Rip-Cre miR-802^{fl/fl} mice, the mice were mated for two generations. First, homozygous miR-802 floxed mice (miR-802^{fl/fl}) were crossed with Rip-Cre transgenic animals to select Rip-Cre miR-802^{fl/wt} mice, then Rip-Cre miR-802^{fl/wt} mice were crossed with miR-802^{fl/fl} mice to selectively ablate miR-802 expression in β cells. We apologize for the ambiguous formulation and we have rewritten this sentence.

To facilitate your check, the correction in revision was listed below.

.....miR-802^{fl/fl} were crossed with Rip-Cre transgenic animals to select Rip-Cre miR-802^{fl/wt} mice, then Rip-Cre miR-802^{fl/wt} mice were crossed with miR-802^{fl/fl} mice to selectively ablate miR-802 expression in β cells (Figure S4g).....

Figure S4 (g) The crosses flowchart to obtain miR-802 KO mice according to Mendelian inheritance.

11. Lines 425 and 930: It is better to provide genotype and sex of littermate controls.

Our response to suggestion 11

Thank you for your suggestion, and we apologize for the ambiguous formulation. We have provided the genotype and sex of littermate controls, which marked red in the text.

12. Lines 489-490: What is insulin promoter derived from? Ins1 or 2 of mice? How much length of promoter is used?

Our response to suggestion 12

Thank you for your comments. The insulin promoter was derived from Ins2 of Rat. The length of promoter was 704 bp. We have provided the Rat Ins2 promoter sequence as supplementary material.

To facilitate your check, the correction in revision was listed below.

The sequence of Rat insulin2 promoter (704 bp)

GGATCCCCCAACCACTCCAAGTGGAGGCTGAGAAAGGTTTTGTAGCTGGGTAGAGTAT
 GTACTAAGAGATGGAGACAGCTGGCTCTGAGCTCTGAAGCAAGCACCTCTTATGGAGA
 GTTGCTGACCTTCAGGTGCAAATCTAAGATACTACAGGAGAATACACCATGGGCTTCA
 GCCCAGTTGACTCCCGAGTGGGCTATGGGTTTTGTGGAAGGAGAGATAGAAGAGAAGG
 GACCTTTCTTCTGAATTCTGCTTTCTTCTACCTCTGAGGGTGAGCTGGGGTCTCAGC
 TGAGGTGAGGACACAGCTATCAGTGGGAAGTGTGAAACAACAGTTCAAGGGACAAAG

TTACTAGGTCCCCAACAACTGCAGCCTCCTGGGGAATGATGTGGAAAAATGCTCAGC
CAAGGACAAAGAAGGCCTCACCTCTCTGAGACAATGTCCCCTGCTGTGAACTGGTTC
ATCAGGCCACCCAGGAGCCCCTCTTAAGACTCTAATTACCCTAAGGCTAAGTAGAGGT
GTTGTTGTCCAATGAGCACTTTCTGCAGACCTAGCACCAGGCAAGTGTGGAACTG
CAGCTTCAGCCCCTCTGGCCATCTGCTGATCCACCCTAATGGGACAAACAGCAAAGT
CCAGGGGTCAGGGGGGGGGTGTCTTGGACTATAAAGCTAGTGGGGATTGAGTAACCCC
CAGCCCTAA

13. Lines 492-493, 506: *It is better to give a name to each plasmid used and make a list (as Table).*

Our response to suggestion 13

Thank you for your meaningful comments. We have given a name to each plasmid used and make a table list (table S8).

14. Line 512: *“miR-802”  “anti-miR-802” and delete “Figure 7d”*

Our response to suggestion 14

Thank you for your suggestions and we are sorry for our incorrect writing. In the revised draft, we have modified miR-802 to anti-miR-802 and delete Figure 7d according to your advice, which marked red in the text.

15. Line 519: *“knockout”  “knockdown”*

Our response to suggestion 15

Thank you for your suggestion. We have corrected “knockout” to “knockdown” in the revised manuscript.

16. Line 919: *“miR-802 knockin mice by cas9/RNA system.” What is Cas9/RNA system?*

Our response to suggestion 16

Thank you for your suggestions and we are sorry for our incorrect writing. We wanted to express CRISPR/Cas9 system, but we mistook it as Cas9/RNA system. We have modified it to CRISPR/Cas9 system in revised draft.

17. Line 922: *Edit to “AflII and BgII”*

Our response to suggestion 17

Thank you for your kind advice. We have corrected “AflII and BgLI” to “AflII and BgII”.

18. Lines 927-928: *Why wild-type allele gives rise to two bands?*

Our response to suggestion 18

Thank you for your comment. We used two pair primers to identify miR-802 overexpression in β cells, and position of primer binding sites were added in this Figure S3e. Primers 6163-CAG-tF1 and 708328-miR802-tR1 were used to identify whether stopper sequence was deleted by Cre/LoxP system or not. When miR-802 KI mice post-Cre, we could observe 321 bp band, otherwise we could observe 1921 bp band when miR-802 KI pre-Cre. Primers 6163-CAG-tF1 and 708328-PGK-tR1 were used to test whether complete stop sequence was deleted. If it was deleted, we could not find any bands, otherwise we could find a 317 bp band. We have added the primer positions used on Figure S3e. And we have provided the location and sequence of primers as

supplemental materials, which could open via software SnapGene or NTI.

19. Line 937: “transfection efficiency”  “knockdown efficiency”

Our response to suggestion 19

Thank you for kind advice. We have corrected “transfection efficiency” to “knockdown efficiency”.

20. Figure 8: This cartoon looks like the cell with nucleus. However, only Sox6 and CREB but not insulin gene and Foxo1 are within the circle (nucleus-like one). In addition, the Fzd5 is not on the external rectangle (cell membrane-like one). I think it is better to edit the Figure.

Our response to suggestion 20

Thank you for your great suggestion. We have edited the Figure 8 (Figure 9 in the revised draft). We hope the changes would make you and other readers much easier to understand our manuscript.

To facilitate your check, the correction in revision was listed below.

Figure 9

21. Figure S2: “c” and “d” must be “d” and “c” according to the legend and text (lines 349 and 906).

Our response to suggestion 21

Thank you for your kind advice. We have corrected it in the text, which marked red in the text. The main correction in revision was listed below.

.....Using a stringent bioinformatics approach, we predicted *miR-802* promoter region (Figure S2c) 3 kb upstream of mice *miR-802* sequence, and constructed the four sgRNAs corresponding to its promoter in lentiCRISPRv2 puro vector. Figure S2d shows that *miR-802* levels were decreased in Min6 cells transfected with sgRNAs.....

Figure S2

Figure S2 (c) The sequence of *miR-802* promoter; the red marker represents the binding site of *Foxo1* and blue marker represents the mutation sequences of binding site. (d) Four sgRNAs for predicted *miR-802* promoter region were constructed in lentiCRISPRv2 puro vector, qRT-PCR was performed to determine the *miR-802* promoter region.

22. Figure S3e: Homology arm regions should be included in the targeting vector. It is better to add the position of primer binding sites in this Figure (name of primers used should be on the Figure legend). What are “H11-P5-I” and “H11-P3-I”? What is the “] ” just downstream of CAG promoter in the vector?

Our response to suggestion 22

Thank you for your suggestions. We have provided the sequence of targeting vector including homology arm regions, which could open via software SnapGene or NTI. We also have added the position of primer binding sites and name of primers used in the Figure S3e.

H11-P5-I was the homology arm of 5’ region and H11-P3-I was the homology arm of 3’ region. We added “] ” at just downstream of CAG promoter in the vector, which meant the CAG promoter regions were finished.

To facilitate your check, the correction in revision was listed below.

Figure S3e

Figure S3 (e) Strategy used to generate *miR-802* knock-in mice by CRISPR/Cas9 system. miRNA sequences were flanked with loxP sites and recombination was induced by crossing these mice with Rip-Cre transgenics. The primer of H11-P5F1, H11-P5R1 and H11-P3F1, H11-P3R1 were used to southern blotting analyze. The primer of 6163-CAG-tF1, 708328-PGK-tR1 and 708328-miR802-tR1 were used to identify *miR-802* KI allele mice.

23. Figure S3f: I cannot see the clear band in WT for BglI.

Our response to suggestion 23

Thank you for your comment. We have replace it by image with higher quality. We hope this change would make you and other readers much clearer to see the clear band in WT for BglI.

To facilitate your check, the correction in revision was listed below.

24. Figure S3g: What are the “BAT” and “WAT”? It is better to explain those in the Figure legend. The allele names should be unified between Figures S3e and S3g. Size of markers should be added.

Our response to suggestion 24

Thank you for your advices. WAT was white adipose tissue and BAT was brown adipose tissue. We have explained those in the figure legend. The allele names were unified between Figures S3e and S3g, and the size of markers were added.

To facilitate your check, the correction in revision was listed below.

Figure S3

Figure S3 (f) Southern blotting of tail genomic DNA isolated from wild type (wt) and heterozygote ($miR-802^{ki/wt}$) animals (n=6). DNA was digested with **AflIII** and **BglI**. Wild type allele: 5.6 kb for AflIII and 5.5 kb for BglI; heterozygote targeted allele: 10.2 kb for AflIII and 8.4 kb for BglI. (h) Selective overexpression of the $miR-802$ gene in islets of $miR-802$ KI mice (n=5). Genomic DNA isolated from indicated mouse tissues was subjected to PCR with $miR-802$ primers spanning LoxP sites (Top) and using GAPDH as control (Bottom). Wild type allele revealed two bands, 317 bp and 1921 bp; mutant allele revealed only one band, 321 bp, **white adipose tissue (WAT)**, **brown adipose tissue (BAT)**.

25. Figure S4e: Primer positions used in Figures S4f and S4g should be added. It is better to add exon-intron structure and promoter region.

Our response to suggestion 25

Thank you for your suggestions. We have marked the position of primers used in figures S4f and S4g on Figure S4e. And we have provided the location and sequence of primers as supplemental materials, which could open via software SnapGene or NTL.

To facilitate your check, the correction in revision was listed below.

Figure S4e

Figure S4 (e) Strategy used to generate $miR-802$ mutant mice by homologous recombination. miRNA sequences were flanked with loxP sites and recombination was induced by crossing these mice with Rip-Cre transgenics. The primer of **100276- $miR-802$ -wt-tF1** and **100276- $miR-802$ -wt-tR1** were used to RT-PCR analyze. The primer of **Zmk-2F4**, and **100276- $miR-802$ -F0-3tR1** were used to selective deletion of the $miR-802$ gene in islets of $miR-802$

KO mice.

26. Figure S4g: Allele name should be unified.

Our response to suggestion 26

Thank you for your suggestion. We have unified the allele name.

27. FigureS5b: It is hard to understand this Figure (I cannot see the same sequence between red one and biding region to two genes).

Our response to suggestion 27

Thank you for your comment. We have rewritten this part. We hope these changes would make you and other readers much easier to understand our manuscript.

To facilitate your check, the correction in revision was listed below.

.....Among the candidates, Neurogenic Differentiation-1 (*NeuroD1*) and Frizzled class receptor 5 (*Fzd5*) were predicted to be *miR-802* target by all the four algorithms, since they harbored an *miR-802* binding site, which were also conserved in humans, mice, and rats (Figure S6b and c)....

Figure legend

Figure S6

Fig.S6 (b-c) Graphic representation of the conserved *miR-802* binding motif in the *NeuroD1* and *Fzd5* 3'UTR of three mammalian species. The consensus mature *miR-802* sequence was depicted on top. And schematic description of the wildtype (top) and mutated (bottom) *miR-802* seed binding motif located in the murine *NeuroD1*- and *Fzd5*-3'UTR used for transient reporter gene transfection experiments.

28. Table S1: “PIP-cre”  “RIP-cre”

Our response to suggestion 28

Thank you for your kind advice. We have corrected “PIP-cre” to “Rip-Cre”.

29. Table S7: The title is “Primer sequences used for RT-PCR”, but sgRNA sequences and siRNA/shRNA sequences are included.

Our response to suggestion 29

Thank you for your suggestions and we are sorry for our incorrect writing. In the revised draft, we have put the sgRNA, siRNA and shRNA sequences in the table S7.

1. Kono TM, *et al.* Human adipose-derived stromal/stem cells protect against STZ-induced hyperglycemia: Analysis of hASC-derived paracrine effectors(Article). *Stem Cells* 1831-1842 (2014).
2. Song X, 2, *et al.* JTC801 Induces pH-dependent Death Specifically in Cancer Cells and

- Slows Growth of Tumors in Mice. *Gastroenterology* 1480-1493 (2018).
3. Wang PX, *et al.* Hepatocyte TRAF3 promotes liver steatosis and systemic insulin resistance through targeting TAK1-dependent signalling. *Nat Commun* **7**, 10592 (2016).
 4. Li L, *et al.* Knockin of Cre Gene at Ins2 Locus Reveals No Cre Activity in Mouse Hypothalamic Neurons. *Sci Rep* **6**, 20438 (2016).
 5. Langlet F, *et al.* Selective Inhibition of FOXO1 Activator/Repressor Balance Modulates Hepatic Glucose Handling. *Cell* **171**, 824-835 e818 (2017).
 6. Higuchi C, *et al.* Identification of circulating miR-101, miR-375 and miR-802 as biomarkers for type 2 diabetes. *Metabolism* **64**, 489-497 (2015).
 7. Kornfeld JW, *et al.* Obesity-induced overexpression of miR-802 impairs glucose metabolism through silencing of Hnf1b. *Nature* **494**, 111-115 (2013).
 8. FJ N. Diabetes, defective pancreatic morphogenesis, and abnormal enteroendocrine differentiation in BETA2/neuroD-deficient mice. *Genes Dev* 2323-2334 (1997).
 9. Olbrot M, Rud J, Moss LG, Sharma A. Identification of beta-cell-specific insulin gene transcription factor RIPE3b1 as mammalian MafA. *Proc Natl Acad Sci U S A* **99**, 6737-6742 (2002).
 10. Jia S, *et al.* Insm1 cooperates with Neurod1 and Foxa2 to maintain mature pancreatic beta-cell function. *EMBO J* **34**, 1417-1433 (2015).
 11. Zhang E, *et al.* Preserving Insulin Secretion in Diabetes by Inhibiting VDAC1 Overexpression and Surface Translocation in beta Cells. *Cell Metab* **29**, 64-77 e66 (2019).
 12. Tang C, *et al.* Evidence for a role of superoxide generation in glucose-induced beta-cell dysfunction in vivo. *Diabetes* **56**, 2722-2731 (2007).
 13. Zhu Y, *et al.* MicroRNA-24/MODY gene regulatory pathway mediates pancreatic beta-cell dysfunction. *Diabetes* **62**, 3194-3206 (2013).
 14. Huang Q, *et al.* Erratum. Glucolipotoxicity-Inhibited miR-299-5p Regulates Pancreatic beta-Cell Function and Survival. *Diabetes* 2018;67:2280-2292. *Diabetes* **68**, 676 (2019).
 15. Motterle A, *et al.* Identification of islet-enriched long non-coding RNAs contributing to beta-cell failure in type 2 diabetes. *Molecular metabolism* **6**, 1407-1418 (2017).
 16. Olofsson CS, Salehi A, Holm C, Rorsman P. Palmitate increases L-type Ca²⁺ currents and the size of the readily releasable granule pool in mouse pancreatic beta-cells. *J Physiol* **557**, 935-948 (2004).
 17. Olofsson CS, Salehi A, Gopel SO, Holm C, Rorsman P. Palmitate stimulation of glucagon secretion in mouse pancreatic alpha-cells results from activation of L-type calcium channels and elevation of cytoplasmic calcium. *Diabetes* **53**, 2836-2843 (2004).
 18. Pan Y, *et al.* Slug-upregulated miR-221 promotes breast cancer progression through suppressing E-cadherin expression. *Scientific reports* **6**, 25798 (2016).
 19. Latreille M, *et al.* MicroRNA-7a regulates pancreatic beta cell function. *J Clin Invest* **124**, 2722-2735 (2014).
 20. Wang YW, *et al.* Spontaneous type 2 diabetic rodent models. *Journal of diabetes research* **2013**, 401723 (2013).
 21. Guo S, *et al.* Inactivation of specific beta cell transcription factors in type 2 diabetes. *J Clin Invest* **123**, 3305-3316 (2013).
 22. Peyot ML, *et al.* Beta-cell failure in diet-induced obese mice stratified according to body

weight gain: secretory dysfunction and altered islet lipid metabolism without steatosis or reduced beta-cell mass. *Diabetes* **59**, 2178-2187 (2010).

23. Hao P, Waxman DJ. Functional Roles of Sex-Biased, Growth Hormone-Regulated MicroRNAs miR-1948 and miR-802 in Young Adult Mouse Liver. *Endocrinology* **159**, 1377-1392 (2018).

Reviewers' Comments:

Reviewer #1:

Remarks to the Author:

To authors,

In the revised manuscript, the authors have answered all my queries and added significant new information. More experiments were designed and performed to strengthen the main points of the study. It is now more convincing that obesity induced miR802 upregulation is FoxO1 dependent and miR802 has an important role during obesity induced β cell dysfunction. Moreover, the authors also further detected and discussed the effect of miR802 on embryonic β cell development in the revised manuscript.

Taken together, nearly all of my concerns were now fully addressed. The overall significance of the paper has been broadened by the revisions and new data.

Reviewer #2:

Remarks to the Author:

Due to the additional effort and explanations as well as the added new data, the quality of manuscript has greatly improved. I am satisfied with the authors' response and the changes.

Reviewer #3:

Remarks to the Author:

Manuscript has been revised with attention to reviewer comments and improved markedly.

I still have minor comments:

- 1) It would be better to include strain name of zygotes (C57BL/6J ?) in the sentence "Then, Cas9 mRNA, sgRNA, and donor of each model mice were co-injected into 0.5 day' zygotes".
- 2) I am not familiar with the term "selfcross". I would say "intercross" is better.
- 3) In Table S1, length of sgRNA target for "miR-802 knockout-5S4" seems to be short (16 bases?). It must be 20 bases.
- 4) In Figure S3e, homology arms should be added (blue bar for 5' arm and red one for 3' arm) in the donor cassette (upper most construct).
- 5) In figure legend for Figure S3e, English editing may be required for "The primer of ...". Also for the title of Table S8
- 6) Table S7 title: "Primer sequences.. "  "Oligo sequences.. "
- 7) In Table S7, I think that siRNA sequence should be provided as RNA sequence (not as DNA sequence).

Reviewer #4:

Remarks to the Author:

General comments:

The revised manuscript and additional experiments strongly support the authors proposal. The authors' response to comments is acceptable. (Just minor points need editing.)

1. Based on the miRNAs expression screening you have reported in figure 1 a and b, it would be helpful for the readers to know the relationship between the other majorly upregulated miRNAs and islets. Perhaps, cluster analysis of the reported top ten upregulated miRNAs may be necessary to understand why miR-802 was selected for the understanding of the relationship between

obesity and T2D.

2. In figure 3, the authors should detect pri-miR-802 in the islets in control, len-shFoxO1 and len-shFoxO1+lent-miR-802 groups.

3. In figure 3, the authors should check the expression of target proteins (e.g. NeuroD1, Fzd5, CamKII, CREB) in the islets in control, len-shFoxO1 and len-shFoxO1+lent-miR-802 groups.

4. 23. Figure S3f: I cannot see the clear band in WT for BglI. I think that band quality is still insufficient. I recommend that the authors should reloading for the band image with higher quality and optimize brightness in the original data to show convincing result.

5. Authors should improve the data quality of figure S2 .h – Hard to see the difference between control and palmitate treatment.

6. The referee suggests that the authors need to state the weeks of HFD feeding to clarify different stages– Figure 1.e

Dear reviewers:

Thank you very much for your comments and advice to our manuscript entitled “**Obesity-induced overexpression of *miR-802* impairs insulin transcription and secretion**”. We completely accept your recommendation and fully agree that these recommendation can further strength the quality of the manuscript. We have revised the manuscript very carefully and according to the suggestion. To clearly present the response, the comments are shown in *italics* and our responses are shown in **blue font**. A thorough, point-by-point response to each point was raised and all changes, a word file of the revised manuscript with all changes labelled in **red font** has been uploaded. If you have any further questions about the revision, please do not hesitate to contact us.

Best regards,

Liang Jin

Comments:

Reviewer #1 (Remarks to the Author)

To authors

In the revised manuscript, the authors have answered all my queries and added significant new information. More experiments were designed and performed to strengthen the main points of the study. It is now more convincing that obesity induced miR802 upregulation is FoxO1 dependent and miR802 has an important role during obesity induced β cell dysfunction. Moreover, the authors also further detected and discussed the effect of miR802 on embryonic β cell development in the revised manuscript.

Taken together, nearly all of my concerns were now fully addressed. The overall significance of the paper has been broadened by the revisions and new data.

Response: Thanks for your favorable comments.

Reviewer #2 (Remarks to the Author):

Due to the additional effort and explanations as well as the added new data, the quality of manuscript has greatly improved. I am satisfied with the authors' response and the changes.

Response: Thanks for your positive comments.

Reviewer #3:

Manuscript has been revised with attention to reviewer comments and improved markedly. I still have minor comments.

Response: Thanks for your encouraging comments. Following your suggestion, we have provided sufficient materials and tools to further illustrate our results.

1) It would be better to include strain name of zygotes (C57BL/6J ?) in the sentence "Then, Cas9 mRNA, sgRNA, and donor of each model mice were co-injected into 0.5 day' zygotes".

Response: Thank you for this important point. Following your suggestion, we have improved the sentence in the revised manuscript as the follows

"Then, Cas9 mRNA, sgRNA, and donor of each model mice were co-injected into 0.5 day' zygotes of C57BL/6J mice...."

2) I am not familiar with the term "selfcross". I would say "intercross" is better.

Response: Thank you for your careful reviewing and we are sorry for our incorrect writing. As suggested, we have replaced "selfcross" with "intercross" in the revised manuscript.

3) In Table S1, length of sgRNA target for "miR-802 knockout-5S4" seems to be short (16 bases?). It must be 20 bases.

Response: Thank you for your critical reviewing and we are sorry for our carelessness. Indeed, the length of miR-802 knockout-5S4 sgRNA is 20 bases, and its sequence is TGAGCTTGGCCTCCCGGCC. We have corrected it in the Supplementary Table S1.

To facilitate your check, the main correction in revision was listed below.

Supplementary Table 1 The sgRNA sequences of miR-802 knockout and knock-in mice.

sgRNA name	sgRNA sequence(5'-3')	PAM
miR-802 knockout-5S4	TGAGCTTGGCCTCCCGGCC	TGG
miR-802 knockout-3S6	ACTCGGATAGTATGCACACT	CGG
miR-802 knock-in-H11-S2	CTGAGCCAACAGTGGTAGTA	AGG

4) In Figure S3e, homology arms should be added (blue bar for 5' arm and red one for 3' arm) in the donor cassette (upper most construct).

Response: Thank you for this important suggestion. As suggested, we have added the homology arms in the donor cassette (upper most construct).

To facilitate your check, the main correction in revision was listed below.

5) In figure legend for Figure S3e, English editing may be required for “The primer of ...”. Also for the title of Table S8.

Response: Thank you for these suggestions. According to your advice, we have edited the “The primer of ...” and the title of Table S8, which were marked red in the revised draft.

To facilitate your check, the main correction in revision was listed below.

Figure S3e legend

.....Primers (H11-P5F1, H11-P5R1, H11-P3F1 and H11-P3R1) were used for southern blotting analyze. Primers (6163-CAG-tF1, 708328-PGK-tR1, 708328-miR802-tR1) were used for selective overexpression of the *miR-802* gene in islets of *miR-802 KI* mice...

The title of Table S8:

Supplementary Table 8 Recombinant plasmids used in this paper

6) Table S7 title: “Primer sequences.. “  “Oligo sequences.. “

Our response to suggestion 6

Response: Thank you for your kind advice. We have corrected “Primer sequences” to “Oligo sequences”.

7) In Table S7, I think that siRNA sequence should be provided as RNA sequence (not as DNA sequence).

Response: Thank you for this critical suggestion. As suggested, siRNA sequences have been provided as RNA sequence in the revised manuscript.

Reviewer #4 (Remarks to the Author):

General comments:

The revised manuscript and additional experiments strongly support the authors proposal. The authors' response to comments is acceptable. (Just minor points need editing.)

Response: We appreciate the reviewer for her/his encouraging comment. We followed your suggestion and provided more convinced evidences to strengthen our conclusions. For example,

we have analyzed the expression level of top ten upregulated miRNAs, and we have detected the expression levels of pri-miR-802 and target proteins (e.g. NeuroD1, Fzd5, CamKII, CREB) in the islets of control, len-shFoxO1 and len-shFoxO1+lent-miR-802 groups. We hope the explanations and changes above would make you and other readers much easier to understand our manuscript.

1. Based on the miRNAs expression screening you have reported in figure 1 a and b, it would be helpful for the readers to know the relationship between the other majorly upregulated miRNAs and islets. Perhaps, cluster analysis of the reported top 10 upregulated miRNAs may be necessary to understand why miR-802 was selected for the understanding of the relationship between obesity and T2D.

Our response to issues 1

Response: We appreciate the reviewer for this insightful comment. As suggested, we have performed cluster analysis of the reported top 10 upregulated miRNAs in the islets of both obese models, respectively, and further clarified the rationale for selecting miR-802 for investigation. Based on these analyses, we revised the manuscript as the follows.

“Furthermore, we performed cluster analysis of the top 10 upregulated miRNAs in the islets of HFD and db/db mice, respectively (Figure S1i and j). Intriguingly, miR-802-5p (miR-802) and miR-1945 were consistently upregulated in both obese models. miR-1945 has been identified in the mouse genome, but its human homologue has not yet been reported. Moreover, it has recently shown that hepatic miR-802 can be induced by obesity and plays a role in insulin resistance and glucose metabolism¹. However, the role of miR-802 in pancreatic β cells remains unknown. Therefore, we chose miR-802 for further analysis.”

Figure S1 Heat map diagram illustrating the expression levels of top 10 up-regulated miRNAs in islets of HFD mice (i), and in Lepr^{db/db} mice (j) (n = 7-8).

2. In figure 3, the authors should detect pri-miR-802 in the islets in control, len-shFoxO1 and len-shFoxO1+lent-miR-802 groups.

Our response to issues 2

Response: Thank you for this insightful comment. As suggested, we have measured pri-miR-802 expression in the islets of control, len-shFoxo1 and len-shFoxO1+lent-miR-802 groups. QRT-PCR analysis showed that lentivirus-shFoxo1 treatment led to a significant decrease in pri-miR-802 expression. Since len-miR-802 was mature miRNA, as shown in Figure S3t, len-miR-802 could slightly restore the pri-miR-802 expression level suppressed by len-shFoxo1.

To facilitate your check, the main correction in revision was listed below.

..... and *lentivirus-miR-802* could also partially restore the *pri-miR-802* expression level suppressed by *len-shFoxo1* (Figure S3t, n=3)....

Figure S3 (t) *Pri-miR-802* expression levels of HFD-fed mice with *lentivirus-shFoxo1* and *lentivirus-miR-802* were analyzed by qRT-PCR (n = 3). Each group was analyzed in triplicates. All the results above were represented as mean \pm SD, ** p < 0.01, *** p < 0.001.

3. In figure 3, the authors should check the expression of target proteins (e.g. *NeuroD1*, *Fzd5*, *CamKII*, *CREB*) in the islets in control, *len-shFoxO1* and *len-shFoxO1+lent-miR-802* groups.

Response: Thanks for this critical suggestion. We have performed the suggested experiment and found that *Foxo1* knockdown markedly increased the expression of *miR-802* targets protein (*NeuroD1* and *Fzd5*) and up-regulated the levels of p-*CamkII* and p-*CREB*. However, the restoration of *miR-802* expression in *Foxo1*-knockdowning cells abolished these increases. To maintain the logic and integrity of manuscript, these results are now described in Figure S6f-h and Figure 8l, respectively. The related description in the main text were as the follows.

“*Lentivirus-shFoxo1* remarkably increased the mRNA and protein levels of *miR-802* targets *NeuroD1* and *Fzd5*, whereas the simultaneous overexpression of *miR-802* nearly abolished these increases (Figure S6 f-h, n = 5-6).”

Figure S6 The mRNA (f-g) and protein (i) levels of *NeuroD1* and *Fzd5* in the islet of HFD-fed mice injected with *lentivirus-shFoxo1* and *lentivirus-miR-802*(n = 3-5, 17-18 weeks old). All the results above were represented as mean \pm SD, ** p < 0.001 compared with the control; ### P <0.01 versus *lentivirus-shFoxo1* group.

“This inhibitory effect of *miR-802* on p-CamkII and p-CREB was recapitulated in the *len-shFoxo1* mice injected with *len-miR-802* (Figure 8l, n=3-5).”

Figure 8 (j-l) Western blot was performed to determine the phosphorylation level of CamkII and CREB in the islets of *miR-802 KO* mice (j, n=5-6, 8-10 weeks old), *miR-802 KI* mice (k, n=5-6, 8-10 weeks old), *len-miR-802* and *len-shFoxo1* mice (l, n = 3-5, 17-18 weeks old). All experiments above were performed in triplicates, where each group consisted of three samples.

4. 23. Figure S3f: I cannot see the clear band in WT for BglI. I think that band quality is still insufficient. I recommend that the authors should reloading for the band image with higher quality and optimize brightness in the original data to show convincing result.

Our response to issues 4

Response: Thank you for your advice and we apologize for our error. In the previous version of our manuscript, Figure S3f was vertically flipped by mistake. We have corrected it in the revised manuscript, and also provided the original image for your reference (Figure below).

Figure S3(f) Southern blotting of tail genomic DNA isolated from wild type (wt) and heterozygote (*miR-802^{ki/wt}*) animals (n=6). DNA was digested with AflIII and BglII, respectively Wild type allele: 5.6 kb for AflIII and 5.5 kb for BglII; heterozygote targeted allele: 10.2 kb for AflIII and 8.4 kb for BglII.

5. Authors should improve the data quality of figure S2 .h – Hard to see the difference between

control and palmitate treatment.

Our response to issues 5

Response: Thank you for this important comment. We have repeated the experiment and provided the new Figure S2h, which showed obvious difference between control and palmitate treatment (Figure below).

6. The referee suggests that the authors need to state the weeks of HFD feeding to clarify different stages— Figure 1.e.

Our response to issues 6

Response: Thank you for this important advice. As suggested, we have clearly stated the weeks of HFD feeding in the Figure 1e legend in the revised manuscript.

To facilitate your check, the main correction in revision was listed below.

Figure 1 (e) qRT-PCR was performed to measure the expression level of *miR-802* in the islets at different stages (after 0, 4-week, 6-week, 8-week and 16-week feeding HFD) during the development of obesity inducing diabetes (n=3-5).

1. Kornfeld JW, *et al.* Obesity-induced overexpression of miR-802 impairs glucose metabolism through silencing of Hnf1b. *Nature* **494**, 111-115 (2013).

Reviewers' Comments:

Reviewer #4:

Remarks to the Author:

In the revised manuscripts, the quality of the manuscript has improved with data that we asked for it and additional new information. The author's response to comments is acceptable.